

# Replacing mechanical protection with colorful faces–twice: parallel evolution of the non-operculate marine worm-snail genera *Thylacodes* (Guettard, 1770) and *Cayo* n. gen. (Gastropoda: Vermetidae)

Rüdiger Bieler[1], Timothy M. Collins[1,2], Rosemary Golding[1], Camila Granados-Cifuentes[2], John M. Healy[1,3], Timothy A. Rawlings[1,4] and Petra Sierwald[1]

[1] Negaunee Integrative Research Center, Field Museum of Natural History, Chicago, IL, United States of America
[2] Department of Biological Sciences, Florida International University, Miami, FL, United States of America
[3] Biodiversity and Geosciences Program, Queensland Museum, South Brisbane, Queensland, Australia
[4] Department of Biology, Cape Breton University, Sydney, Nova Scotia, Canada

Corresponding author
Rüdiger Bieler,
rbieler@fieldmuseum.org

## ABSTRACT

Vermetid worm-snails are sessile and irregularly coiled marine mollusks common in warmer nearshore and coral reef environments that are subject to high predation pressures by fish. Often cryptic, some have evolved sturdy shells or long columellar muscles allowing quick withdrawal into better protected parts of the shell tube, and most have variously developed opercula that protect and seal the shell aperture trapdoor-like. Members of *Thylacodes* (previously: *Serpulorbis*) lack such opercular protection. Its species often show polychromatic head-foot coloration, and some have aposematic coloration likely directed at fish predators. A new polychromatic species, *Thylacodes bermudensis* n. sp., is described from Bermuda and compared morphologically and by DNA barcode markers to the likewise polychromatic western Atlantic species *T. decussatus* (Gmelin, 1791). Operculum loss, previously assumed to be an autapomorphy of *Thylacodes*, is shown to have occurred convergently in a second clade of the family, for which a new genus *Cayo* n. gen. and four new western Atlantic species are introduced: *C. margarita* n. sp. (type species; with type locality in the Florida Keys), *C. galbinus* n. sp., *C. refulgens* n. sp., and *C. brunneimaculatus* n. sp. (the last three with type locality in the Belizean reef) (all new taxa authored by Bieler, Collins, Golding & Rawlings). *Cayo* n. gen. differs from *Thylacodes* in morphology (*e.g.*, a protoconch that is wider than tall), behavior (including deep shell entrenchment into the substratum), reproductive biology (fewer egg capsules and eggs per female; an obliquely attached egg capsule stalk), and in some species, a luminous, "neon-like", head-foot coloration. Comparative investigation of the eusperm and parasperm ultrastructure also revealed differences, with a laterally flattened eusperm acrosome observed in two species of *Cayo* n. gen. and a spiral keel on the eusperm nucleus in one, the latter feature currently unique within the family. A molecular phylogenetic analysis based on mitochondrial and nuclear rRNA gene sequences (12SrRNA, trnV, 16SrRNA, 28SrRNA) strongly supports the independent evolution of the two non-operculate lineages of vermetids. *Thylacodes* forms a sister grouping to a clade comprising *Petaloconchus*, *Eualetes*, and

*Cupolaconcha,* whereas *Cayo* n. gen is strongly allied with the small-operculate species *Vermetus triquetrus* and *V. bieleri.* COI barcode markers provide support for the species-level status of the new taxa. Aspects of predator avoidance/deterrence are discussed for these non-operculate vermetids, which appear to involve warning coloration, aggressive behavior when approached by fish, and deployment of mucous feeding nets that have been shown, for one vermetid in a prior study, to contain bioactive metabolites avoided by fish. As such, non-operculate vermetids show characteristics similar to nudibranch slugs for which the evolution of warning coloration and chemical defenses has been explored previously.

## INTRODUCTION

Members of the caenogastropod family Vermetidae differ from other snails in that, after a motile phase as planktonic larvae or crawl-away juveniles, they permanently cement their shells to the substratum and assume a permanently sessile mode of life, not unlike that of tube-building serpulid worms. Cementing takes place on hard substrata (*e.g.,* dead coral surfaces, conspecific or other molluscan shells). Attachment to the substratum may be superficial or embedded (in the latter case, by active entrenching using the rasping organ, the radula, and/or partial overgrowth by neighboring organisms such as corals or coralline algae). Food is acquired by suspension feeding, involving ciliary action of the large gill and/or by trapping particles in a mucous web (*Morton, 1955*). Their irregularly coiled shells, often influenced by the substratum, have made identification and classification by traditional shell characters difficult, even at the genus-group level (*Bieler & Petit, 2011*). Among the few vermetid groups that have been readily identified from gross morphology is the genus *Thylacodes* (formerly known as *Serpulorbis*; *Bieler & Petit, 2010*), as it is the only known group whose members lack an operculum, the protective chitinous (and occasionally calcified) lid that other vermetids use for closing the aperture when the animals withdraw into the shell tube.

Like other sessile organisms, vermetids employ a wide range of predator avoidance strategies. Studies of vermetid predation are sparse, but main predators are known to include fish (such as pufferfish, triggerfish, and parrotfish), sea stars, crab, octopus, and neogastropods (*e.g., Menge et al., 1986*; *Osman, 1987*; *Calvo & Templado, 2005*; *Ramírez et al., 2013*; *Shima, Phillips & Osenberg, 2016*; *Brown et al., 2014*; *Shlesinger, Akkaynak & Loya, 2021*). In intertidal and reef environments with strong predation pressure by grazing and scraping fish, the defense strategy by members of some genera (*Dendropoma, Novastoa*) includes building strong low-lying shells, reducing the profile further by actively entrenching into the substratum by radula action, and closing the aperture with a large operculum that is held shut by the animal's strong columellar muscle (*Golding et al., 2014*; *Schiaparelli et al., 2017*; Bieler, 1983–2023, pers. obs.). Other vermetid groups (*Vermetus,*

*Petaloconchus, Thylacodes*) build long and comparatively thin-shelled "feeding tube" extensions that have been interpreted as exploratory tubes to receive better water flow and to avoid obstacles (*Schiaparelli & Cattaneo-Vietti, 1999*). Such "extension tubes" can be rapidly vacated in case of a predator attack by deeply withdrawing into the shell tube. In *Thylacodes* this behavior is enabled by both the extreme compressibility of the soft-bodied foot that allows comparatively deep withdrawal into the shell tube and the lack of an operculum (Bieler, 1983–2023, pers. obs.).

In *Thylacodes*, yellows, reds, and blacks appear to dominate the exposed foot coloration, which was first suggested to be aposematic by *Morton (1965)*. These non-operculate species seem to rely mostly on the effect such coloration has on potential visual predators during daytime and in shallow-water settings, combined with the deterrent provided by their copiously produced feeding net mucus, which is avoided by fish (*Coles & Strathmann, 1973* (for unspecified Pacific vermetid taxa); Bieler, 1983–2023, pers. obs.). Detailed investigations of vermetid mucus to date exist only for a species of the operculate genus *Ceraesignum*, where it has been demonstrated to contain bioactive metabolites generated by the snails (*Klöppel et al., 2013*). Several of these non-operculate species are now known to be polychromatic, with individuals varying in both overall head-foot coloration and mantle margin color pattern (*e.g.*, *Bieler et al., 2017a*), often forming distinctive color morphs (*e.g.*, orange, dark, beige) and exhibiting substantial differences in conspicuousness. With effects apparently ranging from crypsis to aposematism, there is a strong similarity to color variation in certain nudibranch slugs (*e.g.*, *Tullrot, 1994*).

As part of a long-term biodiversity study of Vermetidae, with special focus on the western Atlantic (*e.g.*, *Bieler, 1996*; *Bieler & Petit, 2011*; *Golding et al., 2014*; *Bieler et al., 2017a*; *Bieler et al., 2019*), this publication focuses on non-operculate taxa. Upon closer examination of previously undescribed non-operculate species in the western Atlantic that traditionally would have been classified in *Thylacodes*, we noticed fundamental morphological differences in soft-body anatomy, shell entrenching behavior, larval shell morphology, and sperm ultrastructure, which together indicated the presence of a previously unrecognized clade of vermetids. A molecular investigation followed to ascertain the position of what herein is recognized as a new genus.

Below we describe a new polychromatic, and partially reef-building, *Thylacodes* species from Bermuda and compare it to previously described members of the genus, especially wide-ranging western Atlantic *T. decussatus* (*Gmelin, 1791*), with which it was confused previously. This is followed by the description of *Cayo* Bieler, Collins, Golding & Rawlings n. gen. with four newly named species from western Atlantic reefs. The –now two –non-operculate groups are placed in phylogenetic context within the Vermetidae based on molecular markers, and aspects of their morphology (including sperm ultrastructure) and body coloration are discussed.

## MATERIALS AND METHODS

The specimens examined during this study were sourced primarily from the collections of the Field Museum of Natural History (FMNH), based on numerous vermetid-focused

expeditions spanning more than 40 years, see below. Further specimens for this study were donated to the Field Museum collections by P.M. Mikkelsen (material from Bahamas, Barbados, and Florida) and J. Worsfold (Bahamas). The material and observations (including regional "absence" data) stem from our vermetid-focused field work in many locales, including: Bahamas (Grand Bahama, New Providence; April 1993, August 1994 [RB]; Andros, March 2004 [TR]; Abaco, August 2012 [RB, TC, PS]); Barbados, August 2010 [RG, TR]; Belize, March 1986 [RB, PS], April 2011 [RB, TC, RG, TR, PS]; Bermuda, July 1983 [RB, PS], May 1988 [RB, PS], July 2013 [RB, TC, TR, PS]); British Virgin Islands (Guana; August 2010 [RB, TC]); Cayman Islands (Grand Cayman; January 2011 [Lauren Dombowsky for this project]); Dominica, August 1978 [RB, PS]; East Florida and the Florida Keys (1985–2022 [RB, TC, TR, PS]); Netherlands Antilles, now Dutch Caribbean and special municipality of the Netherlands (Saba; August 1996, October 2010 [RB, PS]); St. Martin, August 1996 [RB]); St. Lucia, August 1978 [RB, PS]; U.S. Virgin Islands (St. Croix; February 1987 [RB, Michael Hadfield]); and Venezuela, 1986 [RB]. Specimen collecting in the protected waters of the Florida Keys was conducted under Florida Keys National Marine Sanctuary Research Permit FKNMS-2017-069 and U.S. Fish and Wildlife Service Special Use Permit 41580-FWS-17-1 (and earlier issues of these permits) for work in the National Wildlife Refuges. Collecting in Bermuda in 1983 was conducted under the auspices of the Bermuda Biological Station, in 2013 under a Department of Environmental Protection Special Permit (dated 28 June 2013) from the Bermuda Government. Collecting in the Bahamas was conducted under a Scientific Research Collection Permit (dated 17 August 1994) and a Marine Scientific Research Permit (dated 24 July 2012) from the Department of Fisheries/Marine Resources of the Bahamian Government. Field work in Belize was conducted under the auspices of Smithsonian's Caribbean Coral Reef Ecosystems Program, with specimens exported under permission granted by the Belize Fisheries Department (dated 27 April 2011). Vermetid surveys on the Caribbean Island of Saba (now part of The Netherlands) were supported by the Saba Conservation Foundation and permitted by the Executive Council of the Public Entity Saba (1079/10). These collecting events were part of a qualitative investigation of vermetid species-level diversity, and no formal quantitative sampling was undertaken.

Type deposition: Primary types of newly described species collected under the auspices of the Smithsonian's Caribbean Coral Reef Ecosystems Program have been deposited at the National Museum of Natural History in Washington, DC (USNM). Those resulting from collecting in Bermuda have been lodged with the Bermuda Museum of the Bermuda Aquarium, Museum & Zoo in Flatts Village, Bermuda (BAMZ). The other type material is at the Field Museum of Natural History in Chicago (FMNH). Individual catalog numbers are specified below.

Other acronyms and abbreviations: FMNH number designations refer to specimen series lodged as voucher material and accessible in the Field Museum of Natural History (FMNH; database access under "invertebrates" at http://collections-zoology.fieldmuseum.org/). Station designations beginning with "FK" refer to the ongoing Florida Keys Molluscan Biodiversity Survey project in the Florida Keys National Marine Sanctuary (FKNMS; *e.g.*, *Bieler & Mikkelsen, 2004*), see Table 1 for collecting events. Comparative historic material

was studied in the collections of the Zoological Museum of the University of Copenhagen (ZMUC), Denmark.

Anatomical and Morphological Studies: We used the protocols outlined in *Golding et al. (2014)* and the treatment of *Thylacodes* species mirrors that for the species described in *Bieler et al. (2017a)*. Author/date references, acceptance of valid species, and higher taxonomic units follow the treatment in *MolluscaBase eds (2023)*, except where specifically noted. Where possible, animals were collected alive and photographed to record coloration of the head-foot and mantle. The systematic anatomical descriptions below are based on both live-observed and wet-preserved (ethanol and formalin/Bouins) fixed specimens. Shells attached to or freshly removed from the substratum were photographed using various digital cameras in the laboratory or alive in the field, when necessary in underwater housings. Shell measurements were made using digital calipers or calibrated microscope eye pieces to record the maximum aperture diameter, the greatest whorl width (often somewhat wider than the aperture), and the length of the shell mass (*i.e.*, greatest length in any direction). Protoconchs were obtained directly from adult or juvenile shells, but overgrowth by the postlarval shell sometimes hindered obtaining a clean specimen. Intracapsular larval shells were also examined when found, and although they may not have reached settlement size, they were clean and suited for observing sculpture. Members of the new genus described herein entrench deeply into the coral matrix and the shell whorls often are solidly bonded with the coral skeleton. Removal for study (and, without the observation of living animals, even their species-level recognition) necessitated destructive sampling. This explains why the shells of positively identified material, including type specimens, are usually fragmented. Three of the newly described *Cayo* species were initially interpreted as belonging to a single polychromatic taxon and their distinction resulted from subsequent detailed laboratory work. Our notes on field/live observations thus had to be parsed between the three new taxa and details are not available in equal detail.

Scanning electron microscopy (SEM, FMNH) followed the procedures outlined by *Bieler et al. (2019)*. Shells and separated protoconchs were cleaned in an ultrasonic water bath. Length of radula ribbons and the maximum number of rows were recorded from two specimens per species, when available. Images were obtained using Leo EVO 60 and AMRAY 1810 microscopes at FMNH.

Transmission electron microscopy (TEM, Australian Microscopy and Microanalysis Research Facility, University of Queensland): Small blocks of testicular tissue and/or sperm duct were fixed in 3% glutaraldehyde in 0.1M phosphate buffer with 10% sucrose and processed following the procedure described in *Healy et al. (2017)*. Thin sections of 70–80 nm obtained with a Leica EM UC6 Ultracut ultramicrotome were stained with lead citrate and uranyl acetate and imaged under a Jeol 1011 transmission electron microscope, operating at 80 kV.

Molecular Systematics: Our molecular sampling focused on 11 non-operculate vermetid specimens from the western Atlantic. Molecular procedures largely followed those outlined in *Golding et al. (2014)*, with modifications highlighted below. In short, specimens for DNA extraction were preserved in 70–95% ethanol or RNA*later*. Whenever possible, tissue was sampled from the same lots as those examined for morphological and anatomical

**Table 1** **Listing of samples.** Samples incorporated in the molecular phylogenetic analysis, including newly collected material and DNA sequences available on GenBank. Museum registration numbers for voucher specimens, as well as locality information, collectors and year of collection, and GenBank accession numbers are provided. New sequences submitted as part of this study are shown in bold.

| Species | Reg # | Location | Collector*/ Year | GenBank Accession # | | | |
|---------|-------|----------|------------------|-----------------------------|----------|-----|-------------------|
| | | | | mtDNA 12S - 16SrRNA | Gene order | COI | Nuclear 28SrRNA |
| *Cupolaconcha guana* Golding et al., 2014 | FMNH 327154 | Belize: Little Cat Cay, 1–2 m; 16.151°N, 88.200°W | RB, RG, TR, TC, PS; 2011 | KC583378 | 12S-16S | – | KC583410 |
| *Petaloconchus varians* (d'Orbigny, 1839) | FMNH 327246 | U.S.A., Florida: Hobe Sound, Palm Beach County, seawall; 26.956°N, 80.078°W | RB, PMM; 1994 | KC583381 | 12S-V-16S | – | KC583416 |
| *Eualetes tulipa* (Chenu 1843) | FMNH 318223 | U.S.A., Florida: Peanut Island, Palm Beach County, seawall; 26.7719°N, 80.0437°W | TR; 1997 | HM174254 | 12S-V-16S | – | KC583415 |
| *Thylacodes bermudensis n. sp.* | FMNH 337201 | Bermuda: Hamilton Parish, SW corner of Harrington Sound, near Flatt's Inlet; immediately north of Bermuda Aquarium, Museum and Zoo, rocky shore with sand patches, 1-2.5 m; 32.32393°N, 64.73827°W | RB, TR, TC, PS; 2013 | OQ728805 | 12S-V-16S | OQ720922 | OQ725639 |
| *Thylacodes bermudensis n. sp.* | FMNH 337208 | Bermuda: Hamilton Parish, SW corner of Harrington Sound, near Flatt's Inlet; immediately north of Bermuda Aquarium, Museum and Zoo, rocky shore with sand patches, 1-2.5 m; 32.32393°N, 64.73827°W | RB, TR, TC, PS; 2013 | OQ728806 | 12S-V-16S | OQ720923 | OQ725640 |
| *Thylacodes bermudensis n. sp.* | FMNH 337746 | Bermuda: Sandys Parish, N of Grey's bridge, intertidal rocks; 32.31362°N, 64.84862°W | RB, TR, TC, PS; 2013 | OQ728807 | 12S-V-16S | OQ720924 | OQ725641 |
| *Thylacodes decussatus* (Gmelin, 1791) | FMNH 327164 | Belize: Stann Creek District, about 24 km off Dangriga; Carrie Bow Cay and vicinity, 1–3; m16.8065°N, 88.083°W | RB, RG, TR, TC, PS; 2011 | OQ728808 | 12S-V-16S | KY586141 | OQ725642 |
| *Thylacodes decussatus* (Gmelin, 1791) | FMNH 327089 | U.S.A., Florida: Monroe County, Middle Keys; patch reef off Marathon, Key Vaca, 7m; 24.659°N, 81.016667°W | RB, PMM; 1997 | AF338154 | 12S-V-16S | OQ720925 | KC583414 |
| *Thylacodes squamigerus* (Carpenter, 1857) | FMNH 318997 | U.S.A., California: Corona del Mar; 33.59°N, 117.87°W | JN; 2002 | HM174255 | 12S-V-16S | HM174255 | KC583413 |
| *Thylacodes vandyensis* Bieler, Rawlings & Collins 2017 | FMNH 328529 | U.S.A., Florida: Monroe County, Lower Keys; between Western Sambo Reef and Sand Key, about 11 km off Key West, 29m; 24.45273°N, 81.72657°W | RB, PS; 2014 | OQ728809 | 12S-V-16S | KY586138 | OQ725643 |
| *Cayo brunneimaculatus n. sp.* | USNM 1688732 ex FMNH 327187 | Belize: Stann Creek District, about 24 km off Dangriga;patch reef SSW of Carrie Bow Cay, 6m; 16.80116°N, 88.08229°W | RB, PS; 2011 | OQ728810 (12S) OQ732731 (16S) | 12S-V-16S | – | OQ725644 |
| *Cayo galbinus n. sp.* | FMNH 327151 | Belize: Stann Creek District, about 24 km off Dangriga; SW from boat launch at Carrie Bow Cay, on upper surface sides of coral boulder in shallow intertidal zone, 0–1 m; 16.80218°N, 88.08252°W | RB, RG, TR, TC, PS; 2011 | OQ728811 | 12S-V-16S | OQ720926 | OQ725645 |
| *Cayo galbinus n. sp.* | FMNH 327173 | Belize: Stann Creek District, about 24 km off Dangriga; in lagoon east of Carrie Bow Cay, on upper surfaces of dead coral boulders, 0–1 m; 16.80275°N, 88.08163°W | RB, RG, TR, TC, PS; 2011 | OQ728812 | 12S-V-16S | OQ720927 | OQ725646 |

**Table 1** (*continued*)

| Species | Reg # | Location | Collector*/ Year | mtDNA 12S - 16SrRNA | Gene order | COI | Nuclear 28SrRNA |
|---|---|---|---|---|---|---|---|
| | | | | GenBank Accession # | | | |
| *Cayo margarita n. sp.* | FMNH 326730 | U.S.A., Florida: Monroe County, Lower Keys; Looe Key reef, about 9 km north off the island of Ramrod Key, spur & groove reef, 5–9 m; 24.54482°N, 81.40918°W | RB, PS; 2010 | OQ728813 | 12S-V-16S | OQ720928 | OQ725647 |
| *Cayo margarita n. sp.* | FMNH327379 | U.S.A., Florida: Monroe County, Upper Keys; Molasses Reef, spur & groove reef, 4.9–6 m; 25.00917°N, 80.37633°W | RB, PMM; 1999 | OQ728814 | 12S-V-16S | - | OQ725648 |
| *Cayo refulgens n. sp.* | FMNH 327171 | Belize: Stann Creek District, about 24 km off Dangriga; in lagoon east of Carrie Bow Cay, on upper surfaces of dead coral boulders, 0–1 m; 16.80275°N, 88.08163°W | RB, RG, TR, TC, PS; 2011 | OQ728815 | 12S-V-16S | OQ720929 | OQ725649 |
| *Vermetus triquetrus* Bivona Bernardi, 1832 | FMNH 327229 | Spain: Murcia, Cabo de Palos | JT; 1999 | KC583383 | 12S-V-16S | – | KC583419 |
| *Vermetus bieleri* Scuderi, Swinnen, & Templado, 2017 | FMNH 328506 | Azores: São Miguel | AC; 1999 | KC583384 | 12S-V-16S | OQ720930 | KC583420 |
| *Vermetus biperforatus Bieler et al., 2019* | USNM 1480639ex FMNH 327153 | Belize: Stann Creek District, about 34 km S of Dangriga; off Little Cat Cay, 21m; 16.65163°N, 88.20157°W | RB, PS; 2011 | OQ728816 | 12S-V-16S | – | OQ725650 |
| *Dendropoma nebulosum* (Dillwyn, 1817) | FMNH 315538 | U.S.A., Florida: Monroe County, Middle Keys; patch reef off Marathon, Key Vaca, 7 m; 24.659°N, 81.015°W | RB; 1997 | AF338147 | 12S-V-K-P-NAD6-16S | – | KC583388 |
| *Novastoa pholetor Golding et al., 2014* | FMNH 327172 | Belize: Stann Creek District, about 24 km off Dangriga; lagoon E of Carrie Bow Cay, 1 m; 16.803°N, 88.082°W | RB, RG, TR, TC, PS; 2011 | KC583372 | 12S-V-16S | – | KC583403 |
| *Ceraesignum maximum* (G. B. Sowerby I, 1825) | FMNH 318221 | Jordan: Gulf of Aqaba | IK; 1999 | HM174253 | 12S-V-16S | – | KC583405 |

**Notes.**

AC, Ana Costa; IK, Isabella Kappner; JN, Jon Norenberg; JT, José Templado; PMM, Paula M. Mikkelsen; PS, Petra Sierwald; RB, Rüdiger Bieler; RG, Rosemary Golding; SS, Stefano Schiaparelli; TC, Timothy Collins; TR, Timothy Rawlings.

characters. DNA was extracted from pieces of the head/foot using a DNeasy Tissue Kit (Qiagen) and eluted in 50–100 µl AE buffer depending on the amount of starting material. A complete list of non-operculate taxa sampled, including locality details and collector information, is provided in Table 1. We also extracted DNA from foot tissue from the recently described operculate vermetid, *Vermetid biperforatus* (Belize, *Bieler et al., 2019*). In addition, we supplemented our dataset with gene sequences available on GenBank for other operculate vermetid genera (*Ceraesignum, Cupolaconcha, Dendropoma, Eualetes, Novastoa, Petaloconchus, Thylaeodus, Vermetus*) to determine their relationship to focal non-operculate taxa.

DNA sequences were generated through one of two methods: (1) Sanger sequencing of targeted DNA amplification products, or (2) NGS sequencing of vermetid mitochondrial and nuclear genomes using an Ion Torrent PGM with a 318 chip as part of a broader study of vermetid systematics (C Granados-Cifuentes, TA Rawlings, R Bieler, R Golding, P Sharp, TM Collins, 2023, unpublished data). Targeted amplification/Sanger sequencing: Both mitochondrial (mtDNA) and nuclear gene regions were selected for our phylogenetic analyses. For mtDNA, we sequenced a > 1.5 kb fragment spanning domains III–IV of

the small-subunit rRNA (12S), an intervening tRNA valine (trnV), and domains I–IV of the large-subunit rRNA (16S) (*Rawlings, Collins & Bieler, 2001*). The sequence for this gene region was generated from two separate and overlapping amplifications using primer pairs: 12SA/16SA and 12SF/16SBr (*Golding et al., 2014*). Substantial interspecific variation exists at the nucleotide level within this region of the genome, including changes in gene order within and among vermetid genera (*Rawlings, Collins & Bieler, 2001*; *Rawlings et al., 2010*). In addition, we amplified and sequenced a ~1,100 bp portion of the nuclear 28S rRNA gene using primer pairs 28SD1F/28D6R; this region has proved to be informative in discriminating among vermetid lineages previously (*Golding et al., 2014*). Finally, to gauge whether color variation among specimens reflected intraspecific or species-level variation, and to assess species-level distinctiveness of other taxa, we amplified and sequenced the 658 bp barcoding region of the mitochondrial cytochrome oxidase 1 (COI) gene for select taxa (*Hebert et al., 2003*). Methods for DNA amplification, as well as primer sequences for amplification of rRNA genes, have been provided in detail elsewhere (*Golding et al., 2014*). The barcoding region of the COI gene was amplified using universal primers LCO1490 and HC02198 (*Folmer et al., 1994*). DNA sequencing was undertaken at The Centre for Applied Genomics, Sick Kids Hospital, Toronto, Canada. In all cases, PCR products were sequenced on both strands before a final consensus sequence was generated. Sequence editing and assemblage was undertaken using Geneious Prime v2022.2.1. For NGS, genomic DNA was extracted from each sample by proteinase K digestion followed by phenol-chloroform extraction (modified from *Saghai-Maroof et al., 1984*). DNA samples were analyzed for quantity and quality on a Bioanalyzer 2100 (Agilent Technologies Inc., Santa Clara CA). Samples were then sequenced on an Ion Torrent PGM using the 318 chip (ThermoFisher Scientific). *De novo* assembly was conducted in both MIRA (*Chevreux, Wetter & Suhai, 1999*) and the CLC Genomics Workbench (Qiagen, Venlo, The Netherlands). The complete mitochondrial genomes and nuclear ribosomal arrays were recovered by BLAST searches with previously sequenced vermetid mitochondrial genomes and nuclear ribosomal genes with a minimum of 40X coverage for these regions. The relevant regions of these sequences were then extracted for phylogenetic analysis with the PCR-amplified regions.

Data Editing and Phylogenetic Analyses: We used Mitos (http://mitos2.bioinf.uni-leipzig.de/index.py; *Donath et al., 2019*) to scan 12S-16SrRNA sequences to determine the boundaries of trnV gene and to search for other tRNAs and protein-encoding genes associated with potential mtDNA gene order rearrangements within this region, as found in some vermetid species (*Rawlings, Collins & Bieler, 2001*; *Rawlings et al., 2010*). Once our sequences were annotated, we generated a mtDNA dataset for alignment and phylogenetic analysis in Geneious Prime using new 12S-trnV-16S sequences as well as sequences available in GenBank for select taxa. Because *Dendropoma nebulosum* (FMNH 315538) has a derived gene order (12S-V-K-P-NAD6-16S) through this region, we excised the three translocated genes, K-P-NAD6, before adding this sequence to our mtDNA dataset. To facilitate alignment, we separated 12S, trnV and 16S gene sequences into gene specific files, using the trnV gene sequence to define the 3′ end of the 12S and the 5′ end of the 16S gene. *Cupolaconcha guana*, another taxon in our analysis, is also noteworthy here, since it is missing the trnV between 12S and 16S, and so was not represented in the trnV dataset.

Single gene alignments were undertaken in Geneious Prime using MAFFT (v7.450; *Katoh, Rozewicki & Yamada, 2019*), employing the E-INS-i algorithm and a gap opening penalty (GOP) of 1.53 for the 12S and the 16S datasets, and the FFT-NS-i algorithm and GOP of 1.53 for the trnV alignment. The concatenate function in Geneious Prime was then used to reassemble sequences from the three gene regions into a single alignment. Because of extensive length variation among taxa in the mtDNA dataset, we also used Gblocks (*Talavera & Castresana, 2007*) to create a second "conserved" mtDNA dataset in which we eliminated poorly aligned positions and divergent regions within our alignment. To do this, we selected the "relaxed block" settings in Gblocks, which allowed smaller final blocks, gap positions in final blocks, and less strict flanking positions. The 28S dataset was also aligned in Geneious Prime using MAFFT (settings: E-INS-I, GOP=1.53), and then added to the original and conserved mtDNA datasets to generate two combined mtDNA+28S datasets.

Phylogenetic analyses were undertaken on single and combined datasets using Maximum Likelihood (ML) in IQ-TREE (v. 1.6.12) through the IQ-TREE web server (see http://iqtree.cibiv.univie.ac.at/). We estimated the best-fit model for each dataset (mtDNA original; mtDNA conserved, 28S) based on the Bayesian information criterion (BIC) using ModelFinder (*Kalyaanamoorthy et al., 2017*) in IQ-TREE. The best-fit model for each partition was: TVM+F+I+G4 for mtDNA (original), TVM+F+I+G4 for mtDNA (conserved), and TN+F+I+G4 for 28S. For combined analyses, we generated a partition file to apply the appropriate model to each dataset and used the edge-unlinked criterion to allow each partition to have its own set of branch lengths. Branch support for ML analyses in IQ-TREE was determined using the ultrafast bootstrap algorithm and 1,000 ultrafast bootstrap replicates. According to *Minh, Nguyen & Von Haeseler (2013)*, ultrafast bootstrap support values are relatively unbiased compared to the conservative standard bootstrap, with 95% support corresponding roughly to a probability of 95% that a clade is true. We chose to select members of the genera *Dendropoma*, *Novastoa*, and *Ceraesignum* as outgroups for our analyses given the vermetid phylogeny of *Golding et al. (2014)* and our focus on non-operculate vermetids (*Thylacodes*).

We compared phylogenetic inferences using ML to those using Bayesian Inference in MrBayes (v3.2.7; *Ronquist & Huelsenbeck, 2003*; *Ronquist et al., 2012*). Separate model parameters "nst = 6, invgamma" were applied to the mtDNA and 28S dataset, and parameters were allowed to vary independently for each partition. Default priors were used in all analyses. Posterior probabilities of phylogenetic trees were based on running 1,000,000 generations of MCMC, which included two simultaneous runs of four chains (three heated; $h = 0.1$) starting with a random tree and sampling every 500 generations. Preliminary runs of the data output in Tracer (*Rambaut et al., 2018*) were used to confirm the appropriate burn-in for the dataset and ensure stationarity of the dataset. In total, 3,002 trees were used to generate a 50% majority-rule consensus phylogram and estimate Bayesian posterior probabilities.

We also compared genetic distances between select vermetid taxa based on gene sequences of the standard barcoding region of the COI (*Hebert et al., 2003*) and a commonly amplified portion of the 16S gene (*e.g.*, *Lemer et al., 2014*). The 16S gene region was

extracted from our longer 16S sequences and was bounded by universal primer pairs 16Sar and 16Sbr (*Palumbi, 1996*). Gene specific DNA sequences were aligned using MAFFT in Geneious Prime using the settings outlined above, and then exported as FASTA files into MEGA11 (*Tamura, Stecher & Kumar, 2021*). For each gene region, pairwise distances were calculated between taxa using the Kimura-2P algorithm in MEGA11.

The electronic version of this article in Portable Document Format (PDF) will represent a published work according to the International Commission on Zoological Nomenclature (ICZN), and hence the new names contained in the electronic version are effectively published under that Code from the electronic edition alone. This published work and the nomenclatural acts it contains have been registered in ZooBank, the online registration system for the ICZN. The ZooBank LSIDs (Life Science Identifiers) can be resolved and the associated information viewed through any standard web browser by appending the LSID to the prefix http://zoobank.org/. The LSID for this publication is: urn:lsid:zoobank.org:pub:815F8878-6EC4-4EA0-BFB7-3AFEA171E563. The online version of this work is archived and available from the following digital repositories: PeerJ, PubMed Central and CLOCKSS.

## RESULTS

Molecular dataset. We generated a complete mtDNA dataset for all target specimens, except for one taxon, *Cayo brunneimaculatus*. For this species, we were able to amplify and sequence the 12S-trnV region, but not the longer 16S gene region using primers 12SF/16SBr. Consequently, we targeted a shorter (~550 bp) region of the 16S gene using primers 16SAr/16SBr. No gene order changes were uncovered through this region of mtDNA for any of our target specimens compared to the inferred ancestral vermetid gene order (12S-trnV-16S). We were also successful in generating a complete 28S dataset for all targeted specimens. Together, new mtDNA and 28S gene sequences for our targeted specimens were supplemented with genetic data from 11 vermetid species in GenBank to create a final dataset of 22 taxa.

Gene-specific alignments were 553 bp, 78 bp, 1,507 bp and 1,117 bp in length for the 12S, trnV, 16S, and 28S rRNA gene regions, respectively. The concatenated mtDNA dataset was 2,138 bp in length, which was reduced by 44% in length to 1,198 bp when analyzed by Gblocks. The combined mtDNA + 28S datasets resulted in a total alignment of 3255 bp for the original dataset and 2,315 bp for the conserved dataset.

Phylogenetic analyses were congruent in topology across methods (ML, BI) and datasets (original vs conserved alignments), with differences chiefly in the level of support for major clades. For simplicity, therefore, only the ML tree based on the original alignment will be presented herein (Fig. 1), with support values for major clades provided as ultrabootstrap support (ML-UF, above branches) and posterior probabilities (BI-PP, below branches) (Fig. 1). Our ML tree corresponded reasonably well in general architecture to the deeper, more taxon-rich BI phylogeny of *Golding et al. (2014)*, including the separation of *Dendropoma*, *Novastoa*, and *Ceraesignum* from all other vermetid genera (a clade for which the name Dendropomatinae *Bandel & Kowalke, 1997* is available), and the

close phylogenetic relationships of *Petaloconchus*, *Eualetes*, and *Cupolaconcha*. While *Golding et al.*'s (*2014*) analysis included only two non-operculate vermetids in the genus *Thylacodes*, our dataset comprised three described species of *Thylacodes* (*T. decussatus*, *T. squamigerus*, *T. vandyensis*) and five other distinct taxa lacking an operculum. One of these (*Thylacodes bermudensis* n. sp.) grouped within the *Thylacodes* clade, sister to *T. squamigerus*/*T. vandyensis*. Collectively, this robustly-supported monophyletic grouping of *Thylacodes* species was modestly supported (UF/PP: 80%/97%) as sister to the clade of *Petaloconchus*, *Eualetes*, and *Cupolaconcha*. The other four non-operculate taxa (*Cayo brunneimaculatus* n. sp., *C. galbinus* n. sp., *C. margarita* n. sp., *C. refulgens* n. sp.) formed a separate monophyletic grouping, distinct from *Thylacodes*, and strongly supported as sister to a clade consisting of two species of *Vermetus* (*V. bieleri* from northeastern Atlantic archipelagos and *V. triquetrus* from the Mediterranean Sea) both with a small, button-like operculum. Within *Cayo* n. gen., high bootstrap and posterior probability values supported a close sister group relationship between *Cayo galbinus* n. sp. (Belize), with a lime green coloration and *Cayo margarita* n. sp., with a lemon-green coloration (Florida Keys), however, there was no strongly supported hierarchical pattern of relationships between this clade and either *Cayo brunneimaculatus* n. sp. or Cayo *refulgens* n. sp.

Our phylogenetic analysis also included the newly described *Vermetus biperforatus* from Belize (*Bieler et al., 2019*), an operculate species with ability of modifying its shell aperture by covering it with a shell dome with two equal shell openings. Its authors had placed it in *Vermetus* sensu lato for lack of another suitable position. Interestingly, this species did not form a monophyletic grouping with other nominal *Vermetus* species in our analysis. Instead, in our ML tree, *V. biperforatus* was recovered as a sister taxon to the broad grouping of *Vermetus*, *Cayo*, *Cupolaconcha*, *Eualetes*, and *Thylacodes*.

COI genetic distances were high (>20% sequence divergence) for most pairwise comparisons of taxa. The only exceptions were for comparisons between specimens of *Cayo galbinus*, *Thylacodes decussatus*, *and T. bermudensis* (Belize), provisionally labelled as conspecific. For these, sequence divergences were all ≤ 1.1%. Of note were the extremely low levels of genetic differentiation (0 –0.5% differences) between different colour morphs (beige, orange, dark) of *Thylacodes bermudensis*. Our more complete dataset of 16S sequence comparisons showed a very similar pattern, with low levels of sequence divergence between specimens provisionally labelled as conspecifics. This dataset also included comparisons between two specimens of *Cayo margarita*, whose sequence differed by 1%.

Taxonomic descriptions

**Subclass Caenogastropoda Cox, 1960**
**Family Vermetidae Rafinesque, 1815**

Marine caenogastropods with dextrally coiling shells. Postlarval shells cemented to substratum or each other, diverting from standard, helical, gastropod coiling pattern (resembling serpulid polychaete tubes), often with longitudinal and/or axial ribbing; no longer occupied sections of shell often sealed off by concave shell plugs; with or without internal calcareous structures supporting the columellar muscle; muscular foot with or without corneous operculum, with a pair of pedal tentacles; food intake by trapping

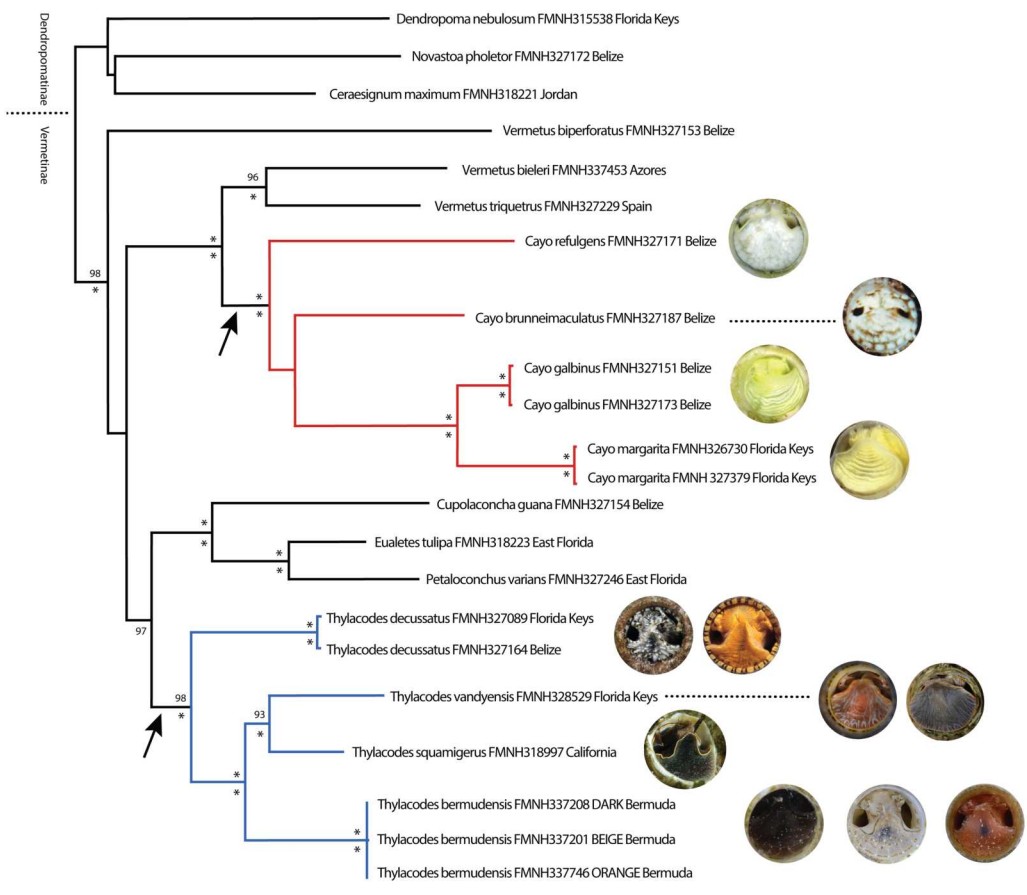

**Figure 1  Maximum Likelihood tree illustrating phylogenetic relationships among select vermetid taxa and focal vermetid genera *Thylacodes* and *Cayo* n. gen.**  Generated in IQ-TREE based on a combined mtDNA (12S, trnV, 16S) and nuclear (28S) dataset. Support values above branches are ML ultrafast bootstrap percentages, and below Bayesian posterior probability values (* representing 100%). For the two non-operculated genera (*Cayo* n. gen in red, *Thylacodes* in blue), the range of head-foot coloration encountered in this study, photographed at the circular shell aperture, is shown at the branch tips. Arrows pointing to operculum loss. (All images by the authors except for *T. squamigerus*, sourced from http://www.inaturalist.org/observations/86868865, CCO 1.0).

plankton and other particles from the water column using the gill and/or a mucous feeding web deployed by the pedal tentacles. Females brooding egg capsules in mantle cavity, either freely or attached to the interior shell wall. Some members with capability of entrenching into calcareous substratum; some with capability of modifying the position, direction, and diameter of the terminal shell section ("feeding tube") by rapid shell growth and cutting action of the robust taenioglossate radula.

### Genus *Thylacodes Guettard, 1770*

Type species by subsequent designation of *Keen* (*1961*: 191): *Serpulorbis polyphragma Sasso, 1827*, an action that made *Thylacodes* an objective senior synonym of *Serpulorbis Sasso, 1827*. For discussions of the complex taxonomic issues involving these genus group names see *Bieler & Petit (2010)* and *Bieler & Petit (2011)*.

Vermetids with large shells (with individual shell masses spanning several centimeters and aperture openings usually considerably larger than 5 mm), often with prominent (in the adult, longitudinal) ribbing, permanently attached to the substratum in the post-larval state, living singly or in clusters. The large muscular pedal disk without an operculum, often polychromatic and strongly colored. Females brood stalked egg capsules (with the stalk at the terminal capsule point) containing multiple eggs, and which are attached to the interior shell through a slit in the anterior mantle. Protoconch pupa-shaped, with rapidly expanding bulbous whorls, forming narrow umbilicus; often with microscopic surface granulations and prominent basal spiral ridge. No distinctive granular sculpture in the columellar region of the interior shell surface.

### *Thylacodes bermudensis* Bieler, Collins, Golding & Rawlings n. sp.
**Figs. 2–4**

*Teredo,* – *Jones, 1864*: 26. *Non Teredo* Linnaeus, 1758 (Bivalvia: Teredinidae).
*Vermetus erectus,* –*Davis, 1904*: 128, pl. 4, fig. 19. Non *Vermetus erectus Dall, 1888*.
*Petaloconchus erectus,* –*Jensen & Pearce, 2009*: 117 (also listing prior Bermudan records of that name).
*Serpulorbis decussatus,* –*Thomas & Cook, 1992*: 137; –*Jensen & Pearce, 2009*: 118 (in part, extant records only; also listing prior Bermudan records of that name). Non *Serpula decussata Gmelin, 1791* (= *Thylacodes decussatus*; see below).

*Type Locality:* Western Atlantic Ocean, Bermuda, Hamilton Parish, SW corner of Harrington Sound, near Flatt's Inlet; immediately north of Bermuda Aquarium, Museum and Zoo; rocky shore with sand patches 1–2.5 m depth; 32°19.436′N, 64°44.296′W (collecting event BER-2013-001; 6 July 2013; R Bieler, T Collins, T Rawlings, P Sierwald).

*Material Examined:* Type specimens: Holotype (largest extent of shell mass 42 mm, tube opening diameter 7.5 mm) and attached paratype (Fig. 2A), BAMZ 2023-326-003 ex FMNH 392795; cemented to shells of *Chama* sp.
Other paratypes: FMNH 337197–337210 (mitochondrial genome analyzed from 337201, DNA data from 337208; internal shell surface in Fig. 2C from 337206), 337198, 337206, 337208, 337425, 337430–337440, 337444, 337839, 337823, 337843, 337846, 337847, BAMZ 2023-326-004 ex FMNH 392796 (all from type locality, BER-2013-001); FMNH 327244 (from type locality, collected in 1988, including SEM specimen Figs. 2D, 2F), 337470–337474, 337683–337684, 337704, 337837 (all from BER-2013-002: N shore, Hamilton Parish, south of Shelly Bay Beach, intertidal rocks and rock pools, 07 July 2013, 32°19.776′N, 64°44.467′W); FMNH 337746 (with DNA data), 337828 (from BER-2013-03, Sandys Parish, N of Grey's bridge, intertidal rocks, 08 July 2013, 32°18.817′N, 64°50.917′W); FMNH 337711, 337838 (from BER-2013-004, Sandys Parish, immediately S of Grey's bridge, intertidal rocks and under rocks in shallow water, 08 July 2013, 32°18.746′N, 64°51.062′W); FMNH 337745–337755, 337845 (from BER-2013-006, ''Cathedral'' dive site, NE side (= ''south'' shore) of island, St. George's Parish, east of St. George's Harbour; subtidal and boiler reefs; SCUBA 1–5 m depth, 09 July 2013, 32°20.535′N; 64°39.412′W);

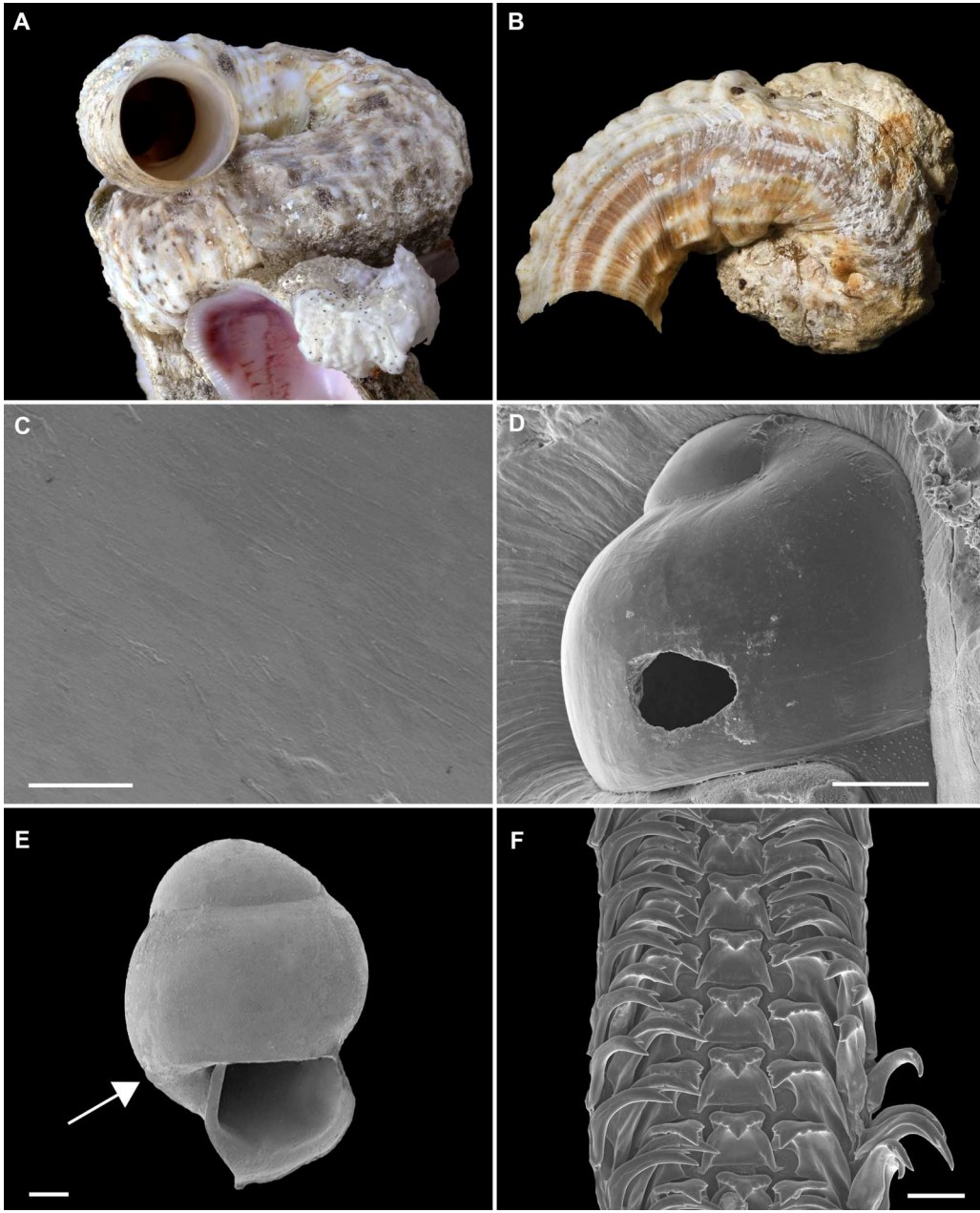

**Figure 2  Shell morphology and radula of *Thylacodes bermudensis* n. sp.** (A) Holotype specimen (diameter of tube opening 7.5 mm; and smaller subadult paratype (white shell, lower right), cemented on purplish shells of the bivalve genus *Chama*; BAMZ 2023-326-003 [ex FMNH 392795]. (B) Paratype [FMNH 337830] with brown interspaces between white longitudinal riblets (largest extent of shell mass 30 mm; diameter of tube opening 8.5 mm); from station BER-2013-007. (C) Internal shell surface in columellar region (SEM); note absence of coarse sculpture [FMNH 337206], scale = 100 μm. (D) Protoconch (SEM; removed from underside of adult shell, body whorl damaged); note overall pupa shape and very fine granules in later part of whorls; FMNH 327244, scale = 100 μm. (E) Apertural aspect of intracapsular larval shell (SEM) [FMNH 337821], note basal keel (arrow) and compare to Figure 1D; scale = 100 μm. (F) Partial aspect of adult radula (SEM) [FMNH 327244; beige morph], scale 100 μm. Images by the authors (R Bieler and R Golding).

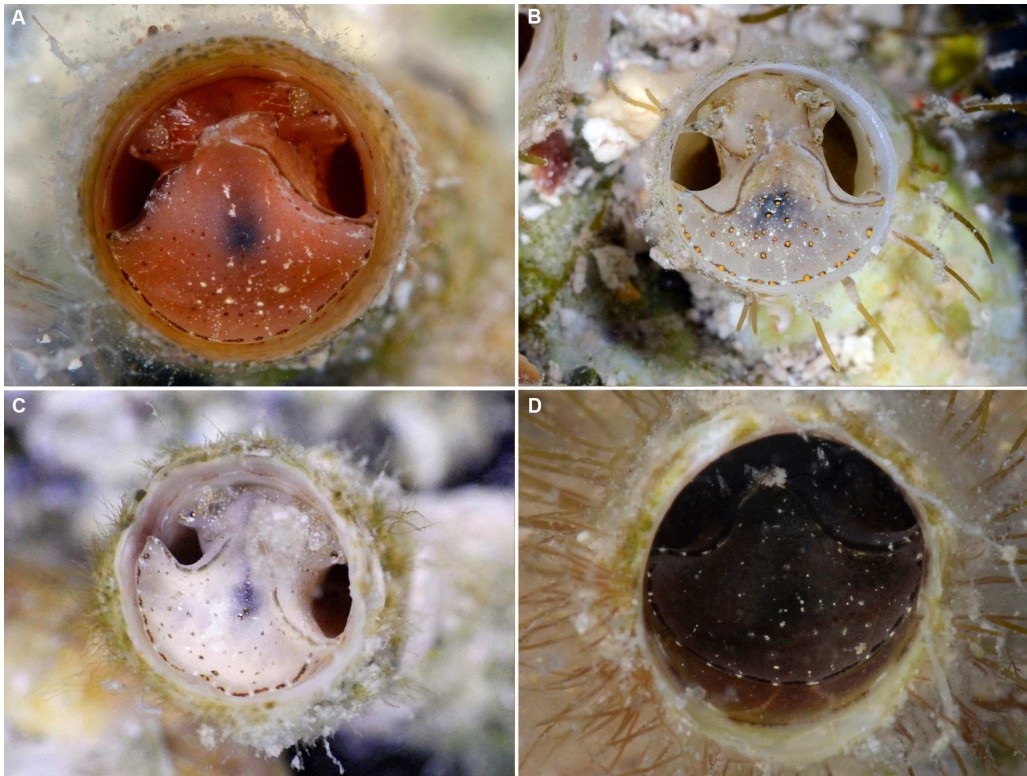

**Figure 3** **External morphology and living coloration of *Thylacodes bermudensis* n. sp.** (A) Orange morph; note central dark spot of pedal disk; pedal tentacles not extended. (B) Beige (or "white") morph with golden-yellow pattern on pedal disk; pedal tentacles not extended. (C) Beige (or "white") morph with yellow pigment limited to head region; note yellow spots near black eyes near the base of cephalic tentacles; pedal tentacles not extended; diameter. (D) Dark morph; pedal tentacles not extended. Specimens in A, B, D from type locality (BER-2013-001), C from station BER-2013-002. Diameter of tube openings 6–7 mm. Images by the authors (R Bieler).

FMNH 337763–337775, 337830 (including striped paratype specimen in Fig. 2B) (all from BER-2013-007; NE side (= "south" shore) of island, St. George's Parish, east of St. David's Island, wreck of the *Rita Zoretta,* SCUBA 3–5 m depth; specimens collected from metal wreckage, 09 July 2013, 32°21.474′N; 64°38.498′W); FMNH 327106 (off Bailey's Bay, Hamilton Parish, 29 June 1983, RB SCUBA, 11 m depth). The large number of assigned catalog numbers reflects field sorting into the three recognized color morphs and various preservatives (*e.g.*, ethanol, RNA*later*, buffered glutaraldehyde).

Additional material: FMNH 327131, 327215, 337821 (Fig. 2E), 337822, 337824, 337825, 337840, 337841 (mostly mixed-species clusters with other vermetid and/or turritellid taxa).

*Description:*

*Teleoconch* (Figs. 2A, 2B): Largest length of attached individual adult shell mass usually 20–40 mm; length of standing portion of adult tube above attached shell mass usually 5–20 mm (rarely exceeding 35 mm); largest diameter of attached shell whorl about 8–10 mm, often developing knobby flange in attached part; diameter at apertural opening of feeding tube in adults about 5–8.5 mm. Not entrenching into the substratum. Early whorls

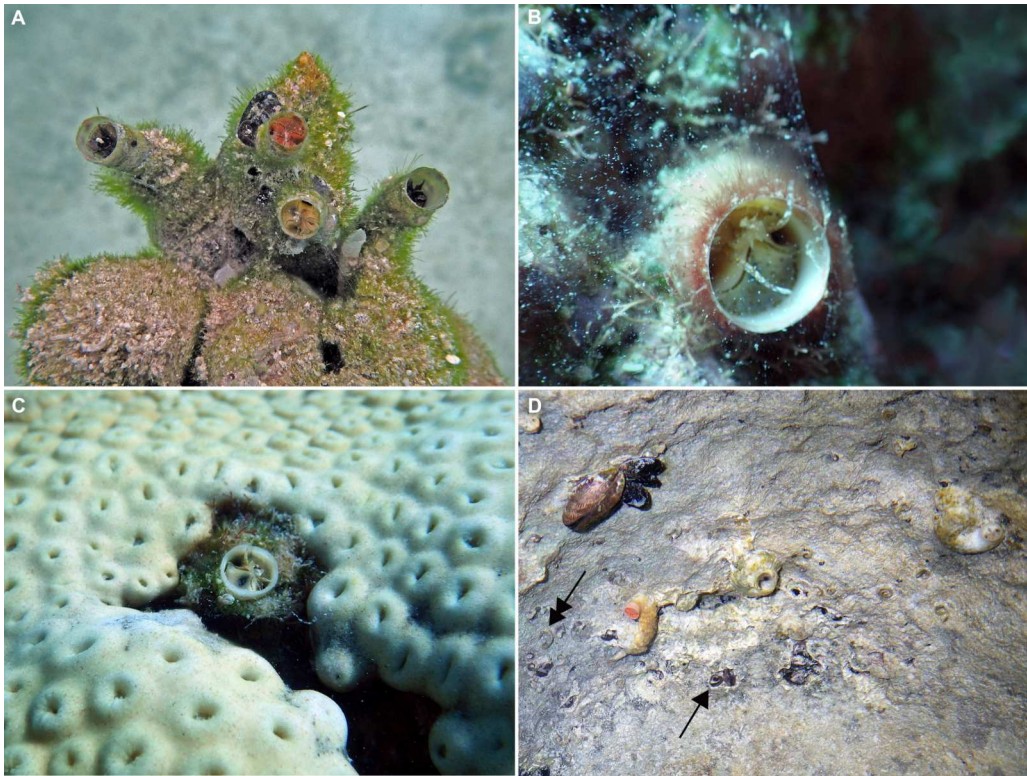

**Figure 4** **Habitat photographs and mucus-net feeding of _Thylacodes bermudensis_ n. sp.** (A) Cluster of specimens from type locality, with dark morphs (left and right), beige morph (lower center, with feeding mucous thread), and orange morph (upper center); the center specimens are surrounded by the dark brown shells of an unrelated worm-snail, the turritellid _Vermicularia spirata_ (Philippi, 1836) [FMNH 337823]; diameter of orange-morph tube opening 7.0 mm. Underwater photograph at type locality (station BER-2013-001), 1.5 m depth. (B) Solitary beige morph with large mucus web; note the long and extended pedal tentacles involved in mucous web production. Underwater photograph at boiler reefs E of St. George's Harbour (station BER-2013-006, 5 m depth; diameter of tube opening 7.5 mm. (C) Solitary beige morph within zoantharian (_Palythoa_ sp.) colony; note the retarded growth of surrounding polyps; underwater photograph at boiler reefs E of St. George's Harbour (station BER-2013-006) in about 5 m depth; diameter of tube opening 7.0 mm. (D) Orange and beige morphs at the bottom wall of a surf-exposed rock pool in intertidal rocks just south of Shelly Bay Beach on Bermuda's north shore, sharing the habitat with the operculate vermetids _Petaloconchus varians_ (simple arrow) and _Dendropoma_ cf. _corrodens_ (double-tipped arrow), as well as mytilid bivalves (station BER-2013-002); diameter of tube opening of orange-morph specimen ca. 7 mm. Images by the authors (R Bieler).

frequently sealed off by concave septa. Sculpture of irregular growth lines, often coarse and knobby (especially at periphery of whorls and in tight turns of the attached shell part), and with 6–10 narrow and longitudinal ridges or threads (and often finer additional striae between them) that are not extending onto the erect feeding tube, if present. Inside wall of shell tube macroscopically smooth, without columellar laminae (Fig. 2C). Color porcellaneous white, often with fine light brown markings, especially at transverse growth marks; occasionally with area between longitudinal threads solid brown, resulting in a longitudinally striped pattern (Fig. 2B). Inside of shell often olive brown, imparting a purplish hue to outside shell. Living specimens usually fouled by algae and other organisms.

*Protoconch* (Figs. 2D, 2E): Pupa-shaped, much taller than wide (height/width ratio ca. 1.5). Height 670–700 µm. About two rapidly expanding bulbous whorls, forming narrow umbilicus; lower part of whorl with strongly (Fig. 2E) developed spiral rib; initial embryonic whorl smooth, then surface with densely spaced, microscopic, individual pustules arranged in spiral striae; light tan.

*Operculum* absent.

*Radula* (Fig. 2F): Length of adult radular ribbon about 1.8–2.6 mm; up to 30 rows. Taenioglossate; rachidian with 3 and lateral teeth with 3–4 cusps on either side of strong central cusp; inner marginal tooth with elongate and strong central cusp, 2–3 small cusps on outer side, and one larger cusp on inner side; slender outer marginal tooth with elongate tip and 1 cusp on inner side. No difference noted between different color morphs.

*External morphology and soft-body coloration* (Figs. 3, 4): Body stout, head-foot massive and fleshy; columellar muscle short. Reproductive females with anterior mantle slit to accommodate stalk of egg capsules. Color in life: Polychromatic: Orange morph (Fig. 3A): milky orange base color; pedal disk with dark zone in center visible through the surface tissue; pedal disk, lateral pedal ridges, and ridges behind cephalic tentacles with fine, interrupted, dark brown to black pigment band with white surface pigment in the interspaces; pedal disk and sides of foot with some speckles of white surface pigment; both sets of tentacles and around the mouth with small yellow pigment clusters usually surrounded by dark brown or black. Mantle margin patterned alternating white and brownish black by very fine pigment granules. Beige (or "white") morph (Figs. 3B, 3C): as above, but with off-white base color, lacking orange pigment; yellow pigment clusters usually (Fig. 3B): but not always (Fig. 3C) more strongly developed than in other color morphs. Dark (or "brown") morph (Fig. 3D): as above, but with dark tan to greyish black base color (depending on amount of orange coloration combined with black pigment); patterning as in the other form, but occasionally solid brown without yellow spotting, and with base body coloration so dark that the center spot of pedal disk does not stand out; head and edge of pedal disk with much black surface pigment. Ethanol-preserved specimens fade to yellowish tan. No formal quantitative assessment of color morph frequency was made in the field, but the beige/white morph appears to be more common than the others (sampled clusters from the type locality analyzed in the lab contained a total of 16 beige, six orange, and five dark morphs; for all specimens with color notes collected in 2013, the ratio was 37 beige, 15 orange, and seven dark individuals).

*Reproductive anatomy*: Reproductive females with anterior mantle slit to accommodate stalk of egg capsules.

*Development*: Several (up to 18), stalked, ovoid eggs capsules brooded in mantle cavity (usually arranged in two rows), maximum length ca. 4.0 mm; attached to female inner shell wall; stalk short and nearly centered at terminal end; eggs and embryos greenish (sometimes bright lime-green to turquoise in color); different developmental stages in single female, ranging from undifferentiated yolk masses to fully developed crawl-away larvae with vela completely resorbed, with about 40 eggs per capsule.

*Sperm ultrastructure:* Not known.
[1] The worm-snail mainly responsible for the buildup of coralline algal-vermetid (cup or "boiler") reefs off the Bermudan south shore is usually referred to as "*Dendropoma annulatus*" in the regional literature (*e.g.*, *Thomas, 2006*; *Thomas, 2007*). The name *Spiroglyphus annulatus* Daudin, 1800 was based on a polychaete worm and was suppressed as invalid by *ICZN (1987)* Opinion 1425. The current name for the common *Dendropoma* species in Bermuda is *D.* cf. *corrodens* (d'Orbigny, 1841) following *Golding et al. (2014)*.

*Habitat and ecology* (Fig. 4): Lower intertidal or subtidal, ranging from sheltered sand and rubble habitat, to walls of deeper intertidal rock pools and sheltered portions of wave-exposed reefs (limestone and dead coral rock); greatest depth observed: 11 m. Often in clusters of different color morphs. Feeding by mucus nets that often are shared among neighboring animals and apparently can retard growth of surrounding cnidarian colonies (Fig. 4C).

*Density*: Ranging from single individuals to dense clusters. *Thomas* (*2006*: 142) noted that "a few reefs off Spanish Point (a prominent headland northwest of Bermuda's capital, Hamilton) are made up entirely of Worm Shells (vermetid snails). However, these [...] are constructed by the Large Tube Shell (*Serpulorbis decussatus*), a different species[1] from the one responsible for building the boilers."

*Molecular data*: Sequence data for three individuals from Bermuda (FMNH 337201; FMNH 337208; FMNH 327746) spanning three gene regions (mtDNA: 12S-trnV-16S, COI; nuclear: 28S) have been deposited on GenBank. See Table 1 for GenBank accession numbers. Low levels of genetic variation in the COI barcoding gene (<0.05%) supports our interpretation that these individuals are conspecifics (Table 2).

*Distribution:* Currently only known from the archipelago of Bermuda.

*Etymology: bermudensis, -e*: "Bermudan" (used as adjective).

*Comparative and Taxonomic Remarks:* The presence of a comparatively large-shelled species of *Thylacodes* in Bermuda has been documented previously. The earliest record appears to be that of *Jones* (*1864*: 26), who mistook the species found "on the reefs and rocks under water; Harrington Sound; tubes standing upright above the rocks" as a member of the wood-boring bivalve genus *Teredo*. *Davis (1904)* and some subsequent authors applied the name *Vermetus erectus* (*Dall, 1888*) to it (which is an operculate, deeper-water western Atlantic species not known from Bermuda). More recently, references to this common Bermudan species have been to "*Serpulorbis decussatus,*" which apparently originated with reports of that taxon in Pleistocene Bermudan deposits by *Richards, Abbott & Skymer (1969)*. The name was subsequently adopted also for the extant species in that island group (*e.g.*, *Thomas & Cook, 1992*), who noticed that the species occurred "in both red and white forms," p. 137). *Muhs, Simmons & Steinke (2002)*: 1369) excluded *T. decussatus* from the modern Bermudan fauna, but that name remained in use for the species here newly named (*e.g.*, *Thomas, 2006*; *Thomas, 2007*). *Thylacodes decussatus* was not found in Bermuda waters during our surveys in 1983, 1988, and 2011, and no Bermudan shells referable to that species were encountered by us in relevant museum collections. All images of living animals identified as *T. decussatus* in Bermuda on the iNaturalist.org site (as of 5 April 2023) are in fact of *T. bermudensis* n. sp. A re-description of *T. decussatus*, which appears to have a wide range from Florida to the Caribbean, is provided for comparative purposes below. *T. bermudensis* differs from *T. decussatus* in lacking the regular and finely scaly sculpture of the teleoconch and the V-shaped black markings on the mantle margin that are characteristic of that species. *T. vandyensis Bieler et al. (2017a)*, another polychromatic species with orange and dark (grey) morphs described from Florida, differs by having

**Table 2  COI and 16SrRNA Kimura 2-P distances for pairwise comparisons.** (A) Between *Cayo* species and *Vermetus bieleri*, and (B) between *Thylacodes species*. COI distances, calculated based on complete alignments of genes, are shown beneath the diagonal and 16S distances above the diagonal. Empty light grey cells in the COI matrix represent cells with no data because COI sequences were not available for one or both taxa. Cells with bold numbers indicate distances at or below 1.1% sequence difference.

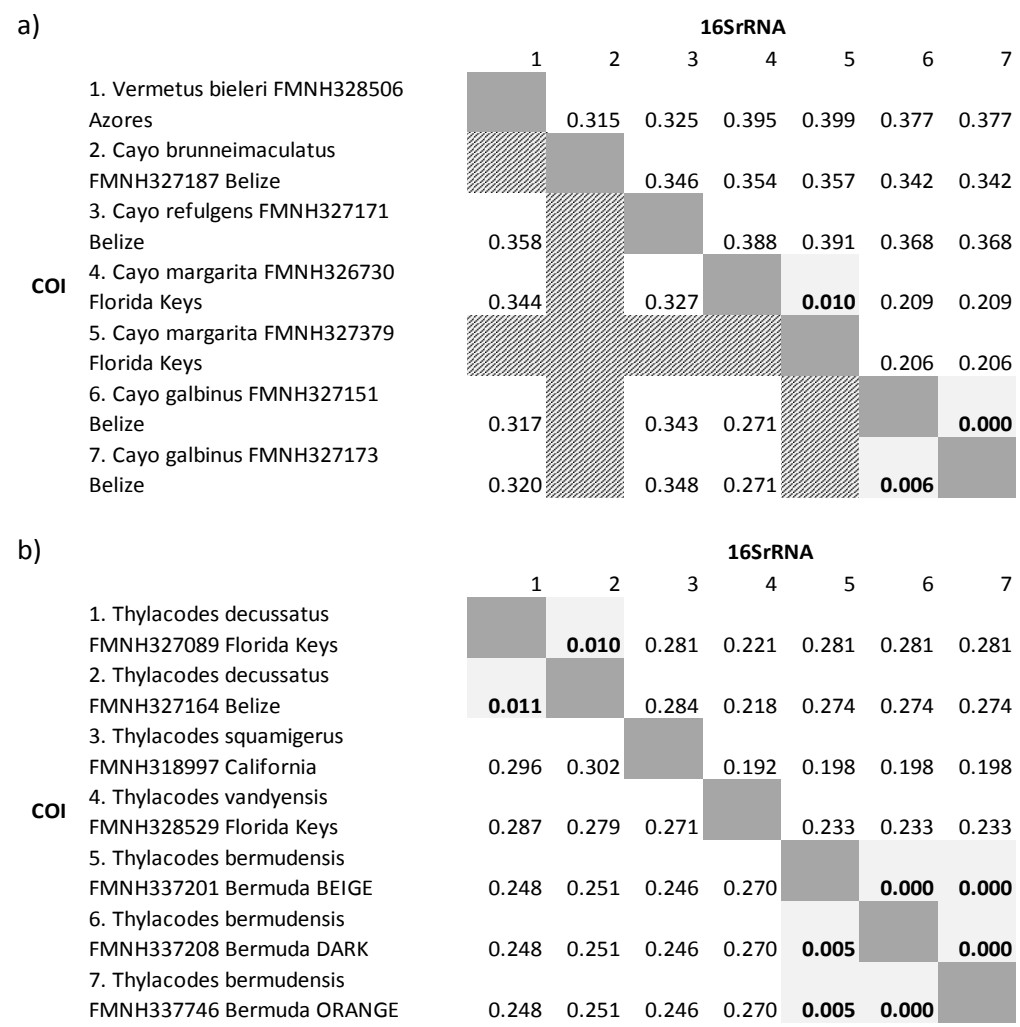

a)

|  | | 16SrRNA | | | | | |
|---|---|---|---|---|---|---|---|
| COI | 1 | 2 | 3 | 4 | 5 | 6 | 7 |
| 1. Vermetus bieleri FMNH328506 Azores | | 0.315 | 0.325 | 0.395 | 0.399 | 0.377 | 0.377 |
| 2. Cayo brunneimaculatus FMNH327187 Belize | | | 0.346 | 0.354 | 0.357 | 0.342 | 0.342 |
| 3. Cayo refulgens FMNH327171 Belize | 0.358 | | | 0.388 | 0.391 | 0.368 | 0.368 |
| 4. Cayo margarita FMNH326730 Florida Keys | 0.344 | | 0.327 | | **0.010** | 0.209 | 0.209 |
| 5. Cayo margarita FMNH327379 Florida Keys | | | | | | 0.206 | 0.206 |
| 6. Cayo galbinus FMNH327151 Belize | 0.317 | | 0.343 | 0.271 | | | **0.000** |
| 7. Cayo galbinus FMNH327173 Belize | 0.320 | | 0.348 | 0.271 | | **0.006** | |

b)

|  | | 16SrRNA | | | | | |
|---|---|---|---|---|---|---|---|
| COI | 1 | 2 | 3 | 4 | 5 | 6 | 7 |
| 1. Thylacodes decussatus FMNH327089 Florida Keys | | **0.010** | 0.281 | 0.221 | 0.281 | 0.281 | 0.281 |
| 2. Thylacodes decussatus FMNH327164 Belize | **0.011** | | 0.284 | 0.218 | 0.274 | 0.274 | 0.274 |
| 3. Thylacodes squamigerus FMNH318997 California | 0.296 | 0.302 | | 0.192 | 0.198 | 0.198 | 0.198 |
| 4. Thylacodes vandyensis FMNH328529 Florida Keys | 0.287 | 0.279 | 0.271 | | 0.233 | 0.233 | 0.233 |
| 5. Thylacodes bermudensis FMNH337201 Bermuda BEIGE | 0.248 | 0.251 | 0.246 | 0.270 | | **0.000** | **0.000** |
| 6. Thylacodes bermudensis FMNH337208 Bermuda DARK | 0.248 | 0.251 | 0.246 | 0.270 | **0.005** | | **0.000** |
| 7. Thylacodes bermudensis FMNH337746 Bermuda ORANGE | 0.248 | 0.251 | 0.246 | 0.270 | **0.005** | **0.000** | |

white streaks in its pedal-disk coloration. *Thylacodes riisei Mörch, 1862*, a species originally described from the Virgin Islands (St. Thomas), has an adult shell sculpture dominated by larger longitudinal threads with multiple weaker threads interspersed. This "*riisei* pattern" of stronger primary and smaller secondary longitudinal shell cords is present also in the nominal species *Thylacodes squamolineatus* (*Petuch, 2002*), a poorly defined taxon that was based on a single incomplete and empty shell collected in deeper water off the Bahamas and originally claimed to be "closest to *S. decussatus*" by its author (2002: 63). Our molecular-based phylogenetic analyses clearly discriminate *Thylacodes bermudensis* from other western Atlantic *Thylacodes* examined herein (*Thylacodes decussatus*; *T. vandyensis*) (Fig. 1).

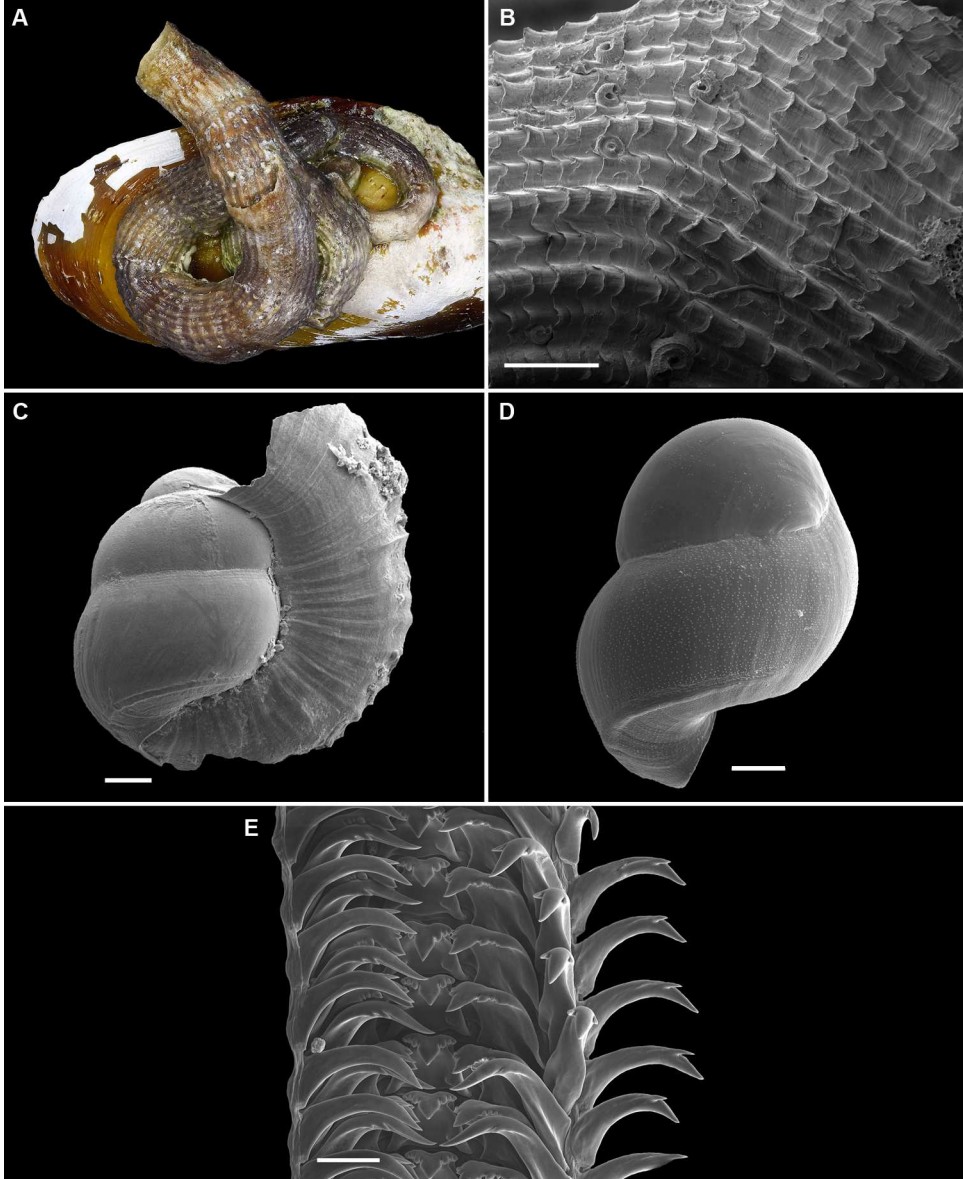

**Figure 5** **Shell morphology and radula of *Thylacodes  decussatus*.** (A) Shell of adult specimen on valve of mytilid bivalve *Modiolus* (FMNH 289840; Key West, Florida Keys); diameter at shell aperture, 5.0 mm. (B) Detail of surface sculpture (SEM) (FMNH 326937, Carrie Bow Cay, Belize); note fluted scales and alternating pattern of stronger and weaker longitudinal ribs/cords; here with attached serpulid (polychaete) shells; scale = 1 mm. (C) Newly settled juvenile with early portion of teleoconch wrapping over apex of protoconch (SEM); FMNH 327009, Carrie Bow Cay, Belize; scale = 100 μm. (D) Intracapsular larval shell, showing finely pustulate sculpture and basal keel (SEM) [FMNH 327143, Belize]; compare to fully developed protoconch in 5C; scale = 100 μm. (E) Partial aspect of adult radula (SEM) [FMNH 326927, Carrie Bow Cay, Belize]; scale = 100 μm. Images by the authors (R Bieler and R Golding).

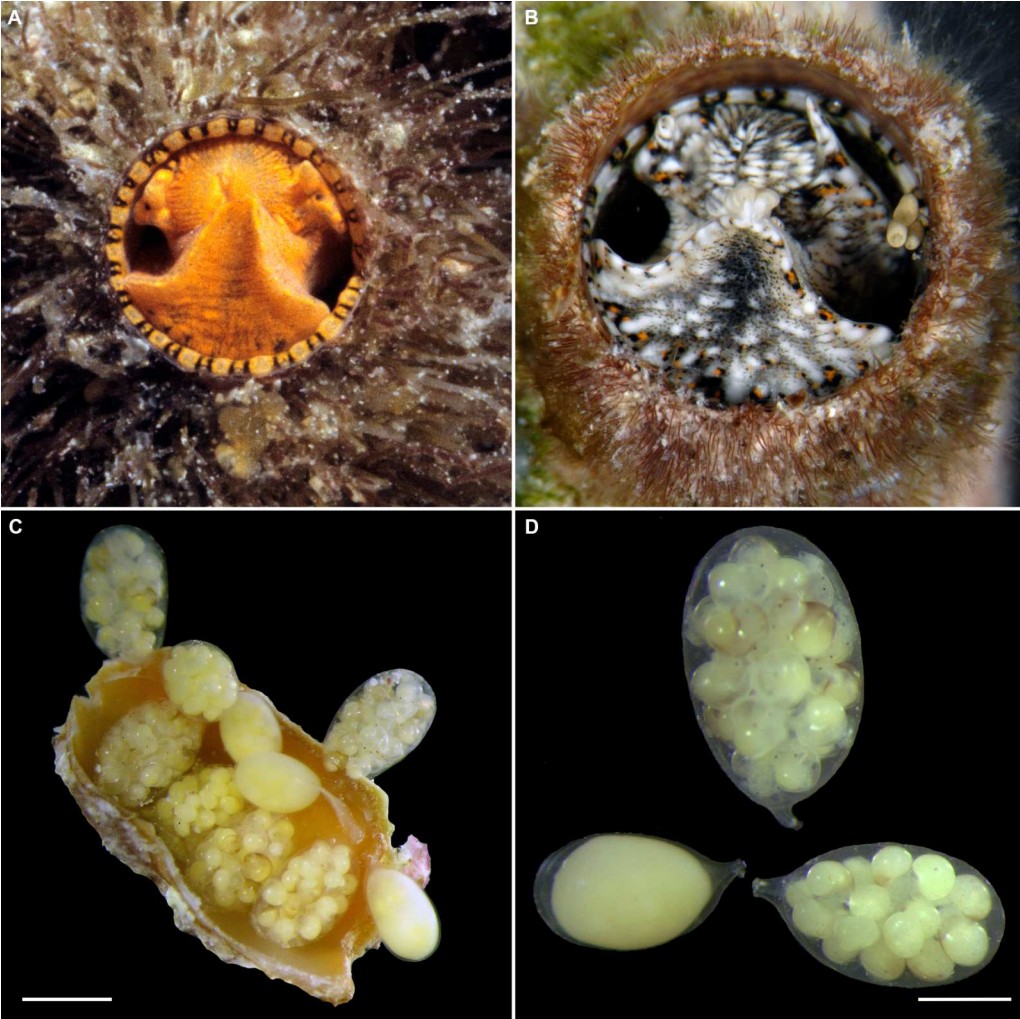

**Figure 6  External morphology, living coloration, and egg capsules of *Thylacodes decussatus.*** (A) Orange morph, with shell covered by algal turf; note black eyes near base of cephalic tentacles and mantle edge pattern of dark V-shaped markings lined by vivid orange coloration; diameter of tube opening ca. 5.0 mm (FMNH 326937, Carrie Bow Cay, Belize, 1986). (B) Grey morph, with shell opening surrounded by filamentous algae; note overall head-foot coloration mottled in white and black and sparse but vivid orange markings on pedal disk, the base of the cephalic tentacles, and lining the V-shaped markings of the mantle edge; pedal tentacles withdrawn; fecal pellets (light tan oval structures) on the right; interior diameter of tube opening 5.0 mm (FMNH 328277, off Big Pine Key, Florida Keys station FK-973). (C) Egg capsules attached to inside female shell (broken open), showing embryos in various stages of development (FMNH 327143, Belize station BEL2001-001); scale 2 mm. (D) From same sample, showing range of developmental stages of stalked egg capsules from a single female; note black eye spots and darkening shells in upper capsule; scale 1 mm. Images by the authors (R Bieler).

**Re-description of *Thylacodes decussatus* (*Gmelin, 1791*)**
Figs. 5–8

—— Lister, 1688 in *Lister (1685–1695)*: pl. 547, fig. 4.
"Das röthliche gestreifte Seewurmgehäuse" *Martini, 1769*: p. 53–54, pl. 2, fig. 17 [after Lister].
*Serpula decussata Gmelin, 1791*: 3745; –*Pfeiffer, 1840*: 2; *Bieler & Petit, 2011*: 33 (catalog).
*Vermetus (Dofania) decussatus,* –*Mörch [as Moerch], 1860*: 34–35.
*Thylacodes decussatus,* –(*Mörch, 1862*): 74-75; –Clessin, 1903 in *Clessin (1901–1912)*: 99–100 (in part, excluding named varieties); –*Bieler et al., 2017a*: 6/27 (with molecular COI data), –*Bieler et al., 2019*: 8. –*Ponder, Lindberg & Ponder, 2020*: 371, fig. 19.6(h) (living animal, stated as from Florida but actually from Carrie Bow Cay, Belize; R. Bieler photograph, 1986)
*Serpulorbis decussatus,* –*Warmke & Abbott, 1961*: 66, pl. 12, fig. D; –*Abbott, 1974*: 101, fig. 966; –*Redfern, 2013*: 102.
*Serpulorbis* cf. *decussatus,* –*Rawlings, Collins & Bieler, 2001*: 1605 (with tRNA-Val data, GenBank: KY338154.1).

*Material examined* (selection, focusing on live-collected material with observations of head-foot coloration):
Florida Keys (Monroe County): **Upper Keys:** FMNH 290118, Carysfort Reef, 25°13.25′N, 80°12.78′W, (collecting event FK-205, 10-IV-99, coral rubble, sand patches, coral heads, 1.8–4.6 m depth, by SCUBA; R Bieler & PM Mikkelsen). FMNH 290119, 08-IV-99, Molasses Reef, 25°00.55′N, 80°22.58′W (collecting event FK-202, 08-IV-99, spur & groove reef, 7.3 m depth, by SCUBA; R. Bieler & P.M. Mikkelsen). **Middle Keys:** FMNH 327085, off Key Vaca, ocean side, "outer patches" S of Hawk Channel, 24°39.30′N, 81°01.30′W [collecting event FK-131, 07-VIII-97, 6.4 m depth, by SCUBA, rubble/gorgonians/sponges, R Bieler and R Cipriani. FMNH 327089; "The Slabs," patch reef between "outer patches" and "coral humps" off Key Vaca, 24°39.53′N, 81°00.90′W (specimen in Fig. 8B; and used for DNA data) [collecting event FK-119, 20-VII-97, 7.0 m depth, by SCUBA; R Bieler & PM Mikkelsen. FMNH 344666 (ex 327090), FK-121, locality as in 327089 (specimen in Fig. 8A) (collecting event FK-121, 21-VII-97, 7.0 m depth, by SCUBA; R. Bieler & P.M. Mikkelsen). **Lower Keys:** FMNH 289840, Key West, E end of Smathers Beach, 24°33.15′N, 81°45.93′W [collecting event FK-281, 07-IV-00, wrack line, freshly dead, by hand, R. Bieler & P.M. Mikkelsen] (specimen in Fig. 5A). FMNH 315949 off west shore of Pigeon Key (bayside of Tavernier), 25°03.299′N, 80°30.712′W (collecting event FK-368, 08-X-00, 0.3–0.9 m, by snorkeling, *Thalassia*/some *Syringodium*; R. Bieler & P.M. Mikkelsen). FMNH 327080, Western Dry Rocks, off Key West, 24°26.63′N, 81°55.61′W (collecting event RB-1598, 26-VI-93, reef and patch reef rubble, gorgonians and fire coral, 6 m depth, by SCUBA, R Bieler & PM Mikkelsen). FMNH 327084, American Shoals, 24°31.56′N, 81°31.10′W (collecting event FK-359, 10-VII-00, 3.3–3.6 m depth, by SCUBA, on *Thalassia/Syringodium* seagrass with rubble and rocks; R. Bieler, P.M. Mikkelsen, R. Bieler, et al.). FMNH 327086, American Shoals, 24°31.41′N, 81°31.14′W (collecting event FK-358, 10-VII-00, 5.5 m depth, by SCUBA, patch reef with gorgonians, rubble, rocks; R Bieler, PM Mikkelsen, et

al.). FMNH 327243, Looe Key Reef, fore reef (collecting event PMM-LK-087, 14-VIII-86, just N of W fore reef, coral rubble field, 1.5−2.4 m depth, by SCUBA, P.M. Mikkelsen). FMNH 328277 (specimen in Fig. 6B), unnamed patch reef due S of Hopkins Island (between Big Munson Island and Cook Island, 24° 37.111′N, 81°22.641′W (collecting event FK-973, 28-V-11, 2.7 m depth, by SCUBA; R Bieler & P Sierwald). FMNH 343312, Looe Key Reef, fore reef to rubble zone, 24°32.87′N, 81°24.41′W (collecting event FK-276, 21-VIII-99, 0.3–2.4 m depth by snorkeling; R Bieler, PM Mikkelsen, P Sierwald & A Bieler).

*Belize* (Western Atlantic, Meso-American Barrier Reef, Belize, Stann Creek District, about 24 km off Dangriga; Carrie Bow Cay and vicinity): FMNH 326924, 326927 (specimen in Fig. 5E), 326937 (specimens in Figs. 5B, 6A), 327009 (specimen in Fig. 5C) (all March 1986; R Bieler & P Sierwald, by SCUBA). The following observed and collected in April 2011 by RB, RG, TC, TR & P. Sierwald: FMNH 327143 (specimens in Figs. 5D, 6C, 6D, 7; also, specimen examined for sperm ultrastructure) (collecting event BEL2011-001, 16°48.131′N, 88°04.951′W, 0–2 m depth, by snorkeling). FMNH 327150 (collecting event BEL2011-003, 16°48.131′N, 88°04.951′W, 0–1 m depth). FMNH 327156 (collecting event BEL2011-008, 16°54.184′N, 88°03.613W, 0–1 m depth. FMNH 327164 (COI data in *Bieler et al., 2017a*; *Bieler et al., 2017b*; other DNA data herein) (collecting event BEL2011-011, 16°48.390′N, 88°04.980′W, 1–3 m depth). FMNH 327179 (collecting event BEL2011-016, Tobacco Reef, 16°49.417′N, 88°04.873′W, 0–1.5 m depth). FMNH 327180 (collecting event BEL2011-019, 16°48.384′N, 88°04.986′W, 0–2 m depth).

*Teleoconch* (Figs. 5A, 5B): Thin-shelled for size. Largest length of attached individual adult shell mass usually 20–40 mm; length of standing portion of adult tube above attached shell mass usually 5–10 mm (rarely exceeding 20 mm); largest diameter of attached shell whorl about 8 mm; diameter at apertural opening of feeding tube in adults about 5 mm. Not entrenching into the substratum. Early whorls occasionally sealed off by concave septa. Sculpture of distinct, rounded spiral cords crossed by weaker growth marks beset with lamella-like scales, the latter particularly closely spaced on feeding tube; frequently with intercalated single, weaker, spiral thread between cords. Inside wall of shell tube macroscopically smooth, without columellar laminae. Color purplish brown; interior surface of shell pale fawn with paired dark brown bands corresponding to patterning on mantle margin.

*Protoconch* (Figs. 5C, 5D): Pupa-shaped, much taller than wide (height/width ratio ca. 1.4). Height 680–820 $\mu$m. About 2.5 rapidly expanding bulbous whorls, forming narrow umbilicus; lower part of whorl with well-developed spiral rib; initial embryonic whorl smooth, then surface with densely spaced, microscopic, individual pustules arranged in spiral striae; light tan.

*Operculum* absent.

*Radula* (Fig. 5E): Length of adult radular ribbon about 2.8 mm; up to 35 rows. Taenioglossate; rachidian with 3 and lateral teeth with 3–6, sometimes ill-defined cusps on either side of strong central cusp; inner marginal tooth with elongate and strong central cusp, 1–2 small cusps on outer side, and one larger cusp on inner side; slender outer marginal tooth with elongate tip and 1 cusp on inner side. No difference noted between different color morphs.

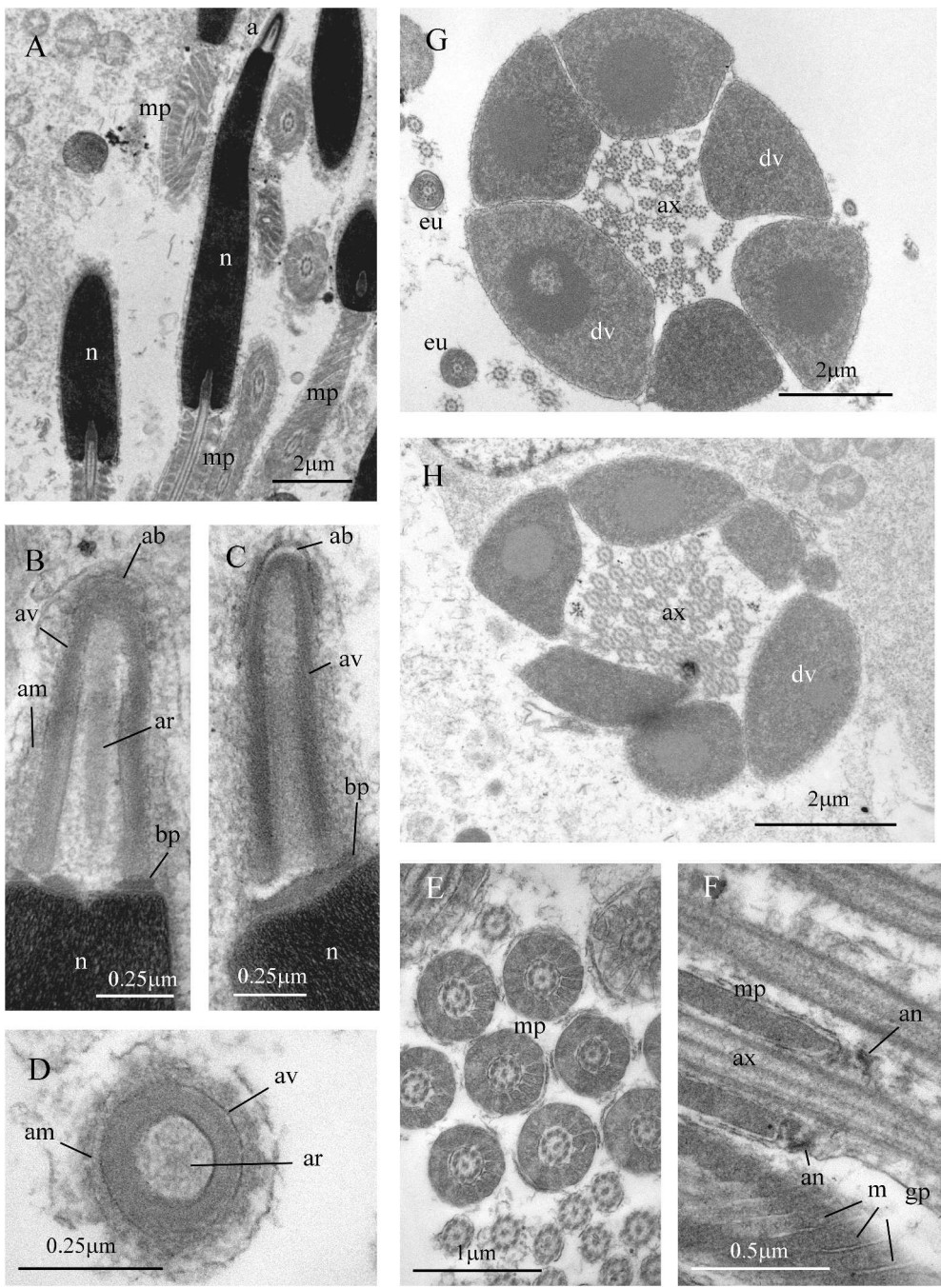

**Figure 7 Sperm ultrastructure of *Thylacodes decussatus*.** (A–E) Euspermatozoa. (A) Longitudinal section (LS) through acrosomal complex (a), nucleus (n) and proximal portion of midpiece (mp). (B, C) LS acrosomal complex (acrosomal vesicle (av), axial rod component of subacrosomal material (ar); apical bleb (ab); accessory membrane (am); basal plate component of subacrosomal material (bp)) situated on nuclear (n) apex. (D) Transverse section (TS) acrosomal vesicle, accessory membrane (am), and subacrosomal material (including axial rod (ar)). Note circular profile of vesicle. 

**Figure 7 (…continued)**
(E) TS midpiece. (F) LS junction of midpiece (mp) and glycogen piece (gp). Note axoneme (ax), annulus (an) and elongate helical mitochondria (m). (G, H) Paraspermatozoa. (G) TS main body of cell showing 39 axonemes (ax), and occasional mitochondria, sheathed by large dense vesicles (dv). Dense vesicle contents clearly show spherical zones of differentiation. Euspermatozoa also visible (eu). (H) TS showing detail of axonemes and dense vesicles. All from FMNH 327143, Belize. Images by the authors (J Healy).

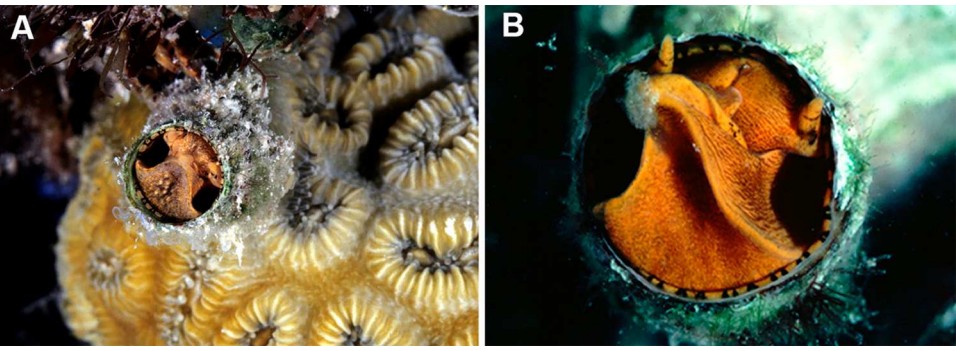

**Figure 8** **Habitat photographs and mucus-net feeding of *Thylacodes decussatus* (Florida Keys).** (A) Solitary animal (orange morph) in Elliptical Star Coral, *Dichocoenia stokesii* Milne Edwards & Haime, 1848, photographed in patch reef off Key Vaca [FMNH 344666, FK-121]; diameter of aperture, 5.5 mm. (B) Orange morph in the process of retrieving mucous web; note mucus bolus at upper tip of very flexible, muscular pedal disk, and the only slightly extended pedal tentacles between head and foot areas (FMNH 327089, patch reef off Key Vaca, FK-119; this specimen was sampled for tRNA-Val data, GenBank: KY338154, by *Rawlings, Collins & Bieler (2001)*). Images by the authors (R Bieler).

*External morphology and soft-body coloration* (Figs. 6A, 6B): Body stout, head-foot massive and fleshy; columellar muscle short. Foot held in a folded triangular shape, with the narrowest part below the mouth. Reproductive females with anterior mantle slit to accommodate stalk of egg capsules. Presence of mantle slit sometimes reflected as longitudinal scar in the female shell tube. Color in life: polychromatic: Orange morph (Fig. 6A; observed in Belize, St. Thomas, Florida Keys, and the Bahamas): head- foot, tentacles, and mantle edge opaque orange; pedal disk and "snout" area with fine black surface pigment; fine dark brown pigment spots roughly arranged into transverse lines on exposed surface of foot and longitudinal lines on sides of foot and neck and radiating out from around mouth, running along the snout and side of head; outermost rim of mantle edge always without black pigment; narrow black pigment band immediately posterior of outer mantle edge; mantle edge with pattern of dual black lines forming open "V" shapes, with area around dual black lines lined with brighter orange pigment; posterior dorsal surface of mantle mottled with opaque black, white, and orange pigment spots over a pale brown background. Orange base coloration turning red in formalin and alcohol. Grey morph (Fig. 5B; observed in Belize, St. Croix, and Florida Keys): foot mottled off-white with fine olive-grey dots and dotted lines between pure-white longitudinal patches; mantle edge with black "open V" pattern as above; in addition, black circles at the edge of concave lateral zones of foot; orange coloration limited to mantle edge (spaces within and next to dual black markings) and small clusters of orange pigment around mouth, on tentacles, and

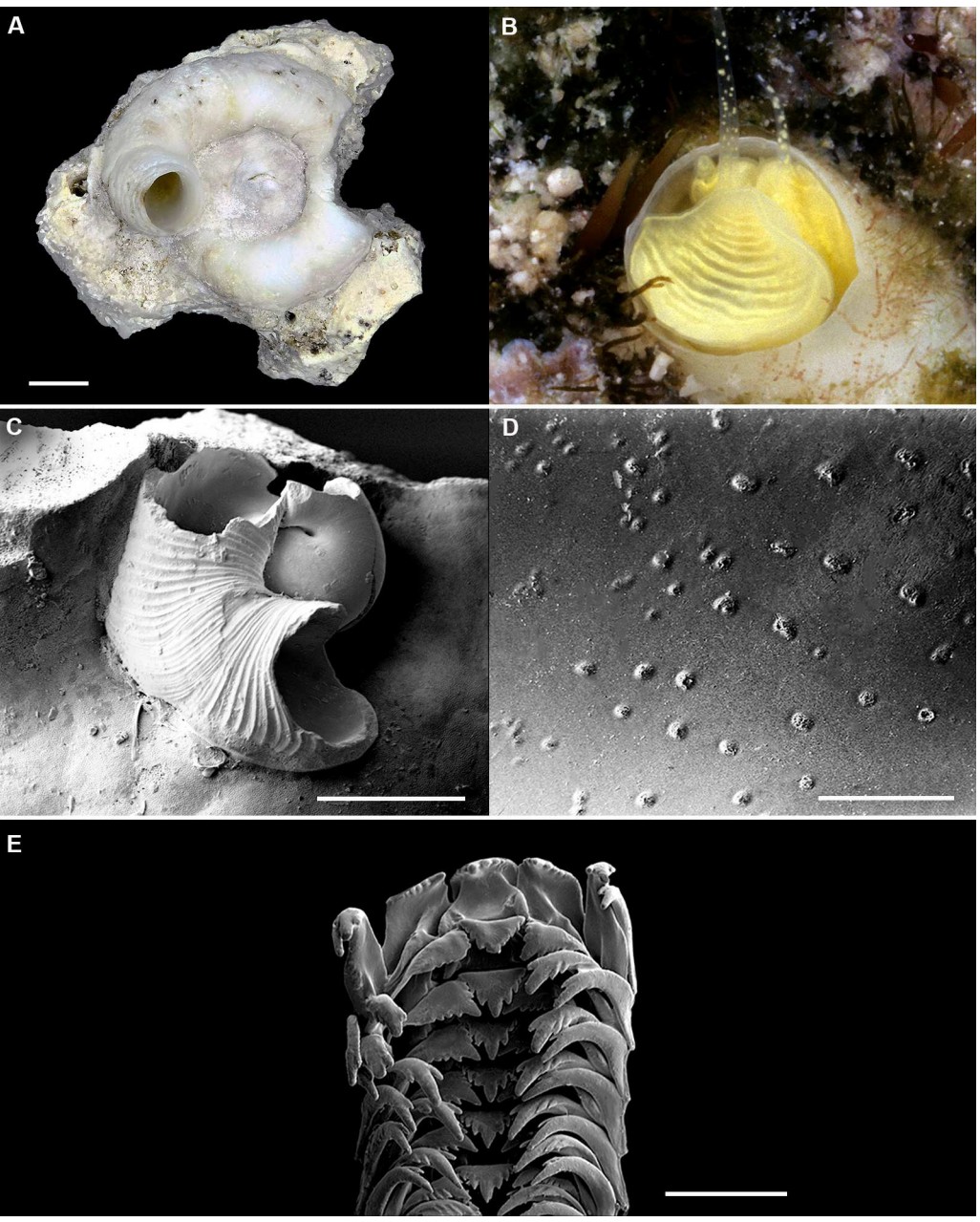

**Figure 9** **Shell morphology, living coloration, and radula morphology and radula of *Cayo margarita* n. sp.** (A) Holotype [FMNH 326730, Florida Keys]; largest diameter 9.3 mm, body whorl damaged on right; scale = 2 mm. (B) Living animal; note pattern of "stacked crescents" on otherwise translucent pedal disk and unequally extended pedal tentacles [FMNH 327380; Florida Keys FK-202]; diameter of tube opening 3.5 mm. (C) Detail of attached protoconch and early teleoconch sculpture (SEM) [FMNH 315561, paratype], scale = 1 mm. (D) Internal shell sculpture in columellar region (SEM) [FMNH 344631, paratype], scale = 100 μm. (E) Partial aspect of adult radula (SEM) [FMNH 315561], scale = 100 μm. Images by the authors (R Bieler).

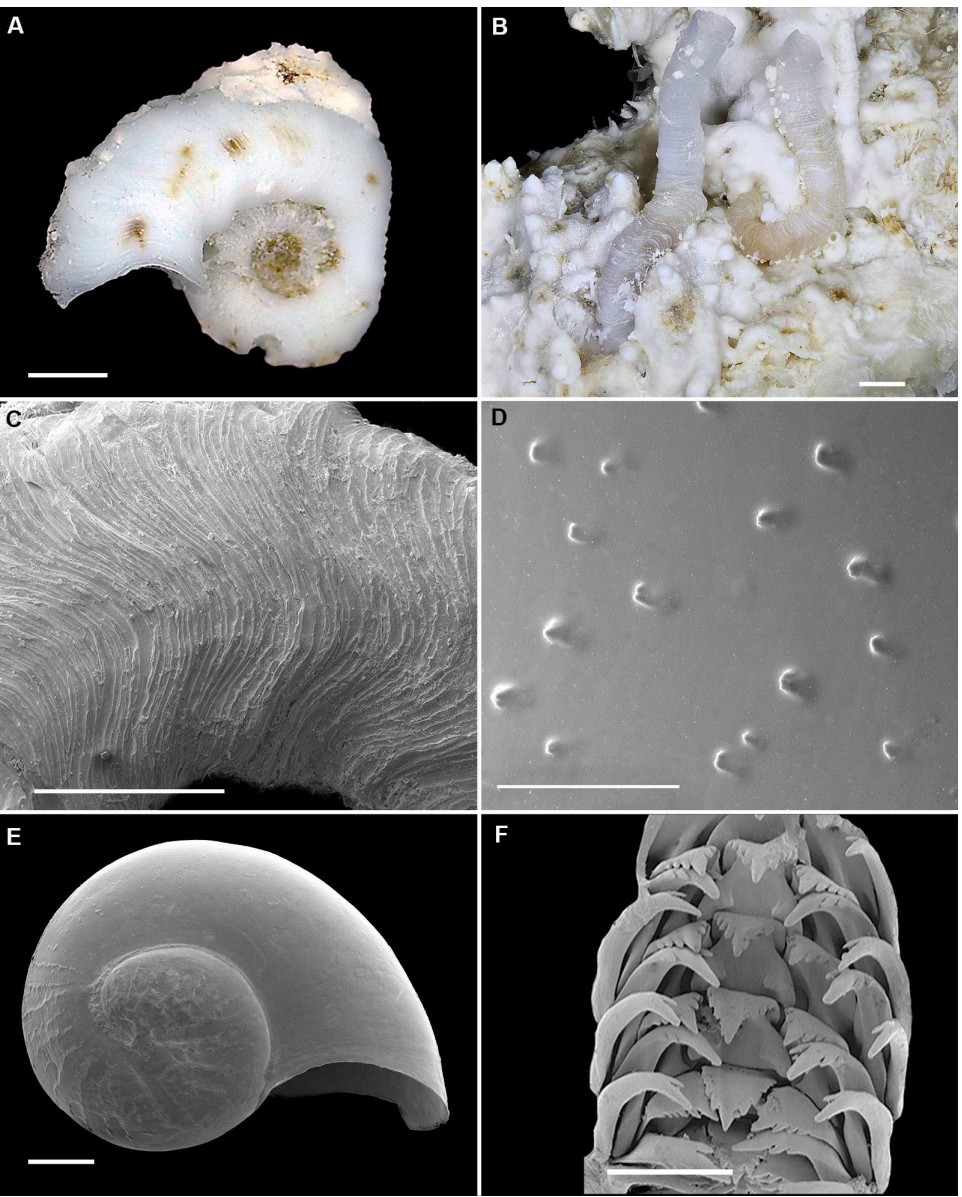

**Figure 10  Shell morphology and radula of *Cayo galbinus* n. sp.**  (A) Holotype, with brown fleck pattern (USNM 1688730 ex FMNH 392794, Belize); largest diameter 4.6 mm, body whorl damaged on right; scale = 1mm. (B) Two paratypes from protected area on underside of coral rock; note long and erect, non-entrenched, tubes [FMNH 327188, Belize]; scale = 2 mm. (C) Detail of teleoconch sculpture (SEM) [FMNH 327144, Belize], scale = 1 mm. (D) Internal shell sculpture in columellar region (SEM) [FMNH 327173, Belize], scale = 100 µm. (E) Apical aspect of intracapsular larval shell (SEM); compare to Fig. 5D in *Thylacodes decussatus* [FMNH 327144, Belize], scale = 100 µm. (F) Partial aspect of adult radula (SEM) [FMNH 327183, Belize] scale = 100 µm. Images by the authors (R Bieler and R Golding).

sometimes lining the pedal disk; circles in concave lateral zone of head-foot often filled with yellow. Juvenile animals with more prominent black pigment covering head.

*Reproductive anatomy*: Reproductive females with anterior mantle slit to accommodate stalk of egg capsules. Presence of mantle slit sometimes reflected as longitudinal scar in the female shell tube.

*Development* (Figs. 6C, 6D): Many (up to 23 per female), stalked, ovoid eggs capsules brooded in mantle cavity (usually arranged in two rows), with maximum length of about 4.0 mm, attached to female inner shell wall; stalk short and nearly centered at terminal end; different developmental stages in single female, ranging from undifferentiated yolk masses to fully developed crawl-away larvae with vela completely resorbed; capsules in latest stages still containing about 20 embryos each.

*Sperm ultrastructure* (Fig. 7): Euspermatozoa (Figs. 7A–7F): filiform and uniaxonemal, with conical acrosomal vesicle (length 1.1 μm, with apical bleb), subacrosomal material organized as axial rod within vesicle invagination and as basal plate at nuclear apex, vesicle invagination running almost full length of vesicle, accessory membrane surrounding base of acrosomal vesicle; nucleus rod-shaped (length 7.8 μm) but flexible with short basal invagination (housing centriolar complex); axoneme surrounded anteriorly by multiple (9–10) helical mitochondria with irregular cristae (midpiece), then putative glycogen granules (glycogen piece) and lastly by plasma membrane only (end piece). Paraspermatozoa (Figs. 7G, 7H): vermiform and multiaxonemal with very large (3–4 μm maximum diameter), block-shaped dense vesicles (contents exhibiting spherical zones of differing electron-density) sheathing 34–48 axonemes and scattered mitochondria.

*Habitat and ecology* (Fig. 8): Usually solitary in crevices among coral rock, at times completely surrounded by living tissue of hard corals (Fig. 8A) or zoanthids; from continuously submerged reef crest at about 1 m depth to 30 m. Specimens in Belize often occurring in shallow water at snorkeling depth, whereas specimens in the Florida Keys were near-exclusively encountered while SCUBA diving.

*Molecular data*: Sequence data for two individuals (FMNH 327089, Florida Keys; FMNH 327164, Belize) spanning mtDNA and nuclear gene regions (mtDNA: 12S-trnV-16S, COI; nuclear: 28S) are available on GenBank. See Table 1 for GenBank accession numbers. Low levels of genetic variation (<1.1%) between these two specimens in both COI and 16S genes supports our interpretation that these individuals are conspecifics (Table 2).

*Distribution*: Known from South Florida, the Bahamas, and the Caribbean. Living specimens of both color morphs observed during this study along the entire Florida Keys archipelago (Carysfort Reef to Dry Tortugas), the Bahamas, the U.S. Virgin Islands, and Belize. Verified shells seen from many other Caribbean locales, including St. Lucia, Dominica, and Venezuela. Records from North Carolina (*Abbott, 1974*) and from Brazil (see taxonomic remarks, below) need verification. Not in Bermuda (see *Thylacodes bermudensis* n. sp., above). Historic Indo-Pacific records for this species (*e.g.*, by *Sowerby, 1892* for South Africa) are in error.

*Taxonomic remarks:* Linking living populations of polymorphic vermetids to historic type material (which usually consists of parts of the adult shell only) and/or the usually brief original descriptions is difficult. *Serpula decussata* was introduced by *Gmelin* (*1791*: 3745) with a short textual description, without a type locality but with reference to the

pre-Linnean text and illustration by Lister (1688 in 1685–1695: pl. 547, fig. 4), whose German vernacular name translates to "reddish striated ocean worm-shell." Based on Lister's given locality (Barbados) and additional material now in the Zoological Museum of Copenhagen (A.H. Riise collection, *vidi*) from St. Thomas (now U.S. Virgin Islands), Mörch (1860, 1862) applied the name *Thylacodes decussatus* to a common Caribbean species. Mörch provided the first comprehensive taxonomic review of the family (*Bieler, 1996*) and his concept of this species was followed by many (*e.g.*, *Warmke & Abbott, 1961*; *Abbott, 1974*; *Redfern, 2013*), but not all subsequent authors. Our live-observed material, which includes specimens from the U.S. Virgin Islands, the Bahamas, Belize, and Florida, matches Mörch's understanding of this nominal species and has previously been cited as *Serpulorbis* cf. *decussatus* in molecular studies (*Rawlings, Collins & Bieler, 2001*). This widespread Caribbean taxon does not seem to reach Bermuda (see *T. bermudensis* n. sp., above) and does not appear to be conspecific with "*Serpulorbis decussatus*" of Brazilian authors (*e.g.*, *Sportorno, 2009*: 119; *Spotorno, Tâmega & Bemvenuti, 2012*). *Simone (2001)*: 183 ff.) provided an extensive morphological description of Brazilian specimens. His description of the shell ("sculptured by irregular longitudinal and transversal low ridges, both predominating, producing a reticulation") does not match our concept of this prominently longitudinally ribbed species, nor does the described head-foot coloration ("scarcely pigmented by brown spots, most pale cream"; *Simone, 2001*: 183). It appears that, parallel to the situation found in Bivalvia (*Simone, Mikkelsen & Bieler, 2015*), assumed conspecific vermetids in Florida, the Caribbean, and in Brazil are in need of additional comparative investigation. Our molecular-based phylogenetic analyses clearly discriminate *Thylacodes decussatus* from *T. bermudensis* n. sp. and *T. vandyensis* (Fig. 1).

### *Cayo* n. gen.

*Type species: Cayo margarita* n. sp., as described below.
*Other species originally included: Cayo galbinus* n. sp., *Cayo refulgens* n. sp., and *C. brunneimaculatus* n. sp., as described below.
*Etymology: Cayo* (male noun), the Spanish term for a small low island in the Caribbean and surrounding regions, equivalent to "key" in Florida, "cay" in the Bahamas, and "caye" or "cay" in Belize. Here referring to the type localities of the four currently known species of this genus, Looe Key in the barrier reef of the Florida Keys and Carrie Bow Cay in the Belizean reef.

*Diagnosis:*
Vermetids with small to medium-sized shells that are sculptured with densely spaced undulating growth marks and, in some species longitudinal cords on the adult whorls; white, often with brown markings on the outside and inside of shell. Attached to the substratum in the post-larval stage, often deeply entrenching into the substratum, living singly (not clustering), frequently forming very regularly expanding spirals in a Flemish flake pattern, resulting in near-circular attachment, occasionally with erect feeding tubes. The large pedal disk without an operculum, in some species with luminous yellow or yellowish green coloration lining edges or covering the entire head-foot. Females brooding stalked egg capsules (with the stalk attaching near but not at the terminal end) containing multiple
eggs, which are attached to the interior shell through a shallow slit in the anterior mantle. Larval shell paucispiral and depressed. Fine granular sculpture in the columellar muscle region of the interior shell. Eusperm, where known, with laterally flattened acrosome.

*Comparative and taxonomic remarks:*

*Cayo* n. gen. differs from *Thylacodes* –the only other known vermetid clade also lacking an operculum –in its entrenching behavior (not present in *T*.), its overall smaller teleoconch size, its regular coiling pattern, its protoconch morphology (pupa-shaped in *T*., paucispiral and depressed helical in *C*.), its often strongly uneven positioning of the pedal tentacles when alive (equal in *T*.), and in some species a luminous, "neon-like" yellow or yellowish green quality of the head-foot coloration (not so in *T*.). Based on available data, *Cayo* females brood fewer egg capsules with fewer hatchlings than those of *T*., and their egg capsule stalks are attached off-center (near-terminal in *T*.). The laterally flattened sperm acrosome in two known species of *Cayo*, and a spiral keel on the acrosome in one species, are currently unique within the family. Most *Thylacodes* species are known for extreme soft-body polychromatism whereas this has not been recorded in the investigated *Cayo* species.

No earlier name for this group has been found. At least seven genus-group names have been introduced for non-operculate vermetids, all of which here interpreted as members of the *Thylacodes* clade. These include: *Aletes* Carpenter, 1857 (with the Californian type species *Aletes squamigerus* Carpenter, 1857), *Cladopa* Gray, 1850 (with the Australian type species *Vermetus grandis* Gray, 1842), *Tetranemia* Mörch [as Moerch] (1859) (with the Australian type species *Vermetus dentiferus* Mörch [as Moerch] (1859)), and *Hatina* Gray, 1842 (with the Red Sea type species *Vermetus inopertus* Leuckart, 1828). For the aforementioned, detailed anatomical data exist for *Aletes* (now *Thylacodes*) *squamigerus* (e.g., Hadfield, 1970). *Tulaxodus* (Guettard, 1770) and *Tulaxoda* Blainville, 1828 are objective synonyms of *Thylacodes*, sharing the same Mediterranean type species, *Serpulorbis polyphragma* (Sasso, 1827) = *Serpula arenaria* Linnaeus, 1758. An additional subjective synonym of *Thylacodes* (if interpreted as a gastropod and not as a polychaete worm) is *Lemintina* Risso, 1826 (see (Bieler & Petit, 2010; Bieler & Petit, 2011); the latter work also providing full references for the nominal genera and type species here discussed).

#### *Cayo margarita* Bieler, Collins, Golding & Rawlings n. sp. Fig. 9

*Type Locality:* Western Atlantic Ocean, U.S.A., Monroe County, lower Florida Keys; Looe Key reef, about 9 km north off the island of Ramrod Key, spur & groove reef, 24°32.689′N, 81°24.551′W, 5–9 m depth (collecting event FK-956, 9 September 2010, by SCUBA; R. Bieler & P. Sierwald).

*Material Examined:*

Type specimens: Holotype: FMNH 326730) (Fig. 9A) with paratype FMNH 327387 from same lot. Holotype (source of mt-DNA data and sperm data herein) voucher shell now kept dry; paratype voucher shell now 2 small fragments with attached tissue remains.

Other paratypes: Florida Keys (Monroe County), all from Upper Keys:

FMNH 290118, 314533, 344624, Carysfort Reef, 25°13.25′N, 80°12.78′W (collecting event FK-205, 10-IV-99, coral rubble, sand patches, coral heads, 1.8–4.6 m depth, by SCUBA; R Bieler & PM Mikkelsen). FMNH 344632, Lake Surprise (SW quadrant), Key Largo, 25°10.5′N, 80°22.8′W [collecting event FK-28, 07-III-96, upper subtidal, by hand; R Bieler & PM Mikkelsen. FMNH 344631 (SEM images of internal shell surface in Fig. 9 from this), Molasses Reef, 25°00.653′N, 80° 22.377′W (collecting event FK-694, 09-VI-03, spur & groove reef, to 6.4 m depth, by SCUBA, R. Bieler & P.M. Mikkelsen). FMNH 327379 Molasses Reef, 25° 00.55′N, 80°22.58′W [collecting event FK-200, 07-IV-99, spur & groove reef, 4.9–6 m depth, SCUBA, R. Bieler & P.M. Mikkelsen] (DNA data from this). FMNH 315561 (SEM images of early shell and radula in Fig. 9 from this), Sand Island reef (near Molasses Reef), 25°01.116′N, 80°22.046′W (collecting event FK-696, 10-VI-03, patch reef with rubble, to 6.7 m depth, by SCUBA, R Bieler & PM Mikkelsen).

Additional material studied:

FMNH 327380 (Specimen in live photo, Fig. 9B; currently missing), Molasses Reef, 25°00.55′N, 80° 22.58′W (collecting event FK-202, 08-IV-99, spur & groove reef, to 7.3 m depth, by SCUBA, R Bieler & PM Mikkelsen). Numerous empty shells in FMNH collections, spanning the entire archipelago, from the Upper Keys to the Dry Tortugas) are likely of this species.

*Description*

*Teleoconch* (Figs. 9A–9E): Largest length of attached individual adult shell mass 9–13 mm, with most about 10.0 mm; either near-fully covered by coralline algal growth (with often only the aperture visible on the surface) or forming a very fairly regular open spiral, either slightly piling up and or attached in one plane in a Flemish flake pattern (with only some overlap between individual whorls), surface-attached shells somewhat entrenched into the substratum (with shallow basin excavated around aperture in specimens without upright feeding tube), with shell floor sometimes thin; often with no or very short (2–3.5 mm) upright feeding tubes (Figs. 9A–9B); largest attached whorl diameter ca. 4 mm, with inner apertural diameter at that size tapering to ca. 2.5 mm. Sculpture of narrow undulating axial (transverse) growth lines on juvenile whorls, later often with fairly coarse, irregular growth marks; without spiral (longitudinal) components (Fig. 9C); feeding tubes smooth except for growth marks. Interior shell wall in region of columellar muscle with minute, irregularly spaced pustules (Fig. 9D). Shell color porcellaneous white, occasionally with some (usually faint) brown staining on parts of the inside shell wall.

*Protoconch* (Fig. 9C): Broadly helical shape, much wider than tall (height/width ratio ca. 0.7). Diameter >600 $\mu$m (broken in all specimens observed). About 1.5 rapidly expanding whorls, with final whorl smoothly rounded, forming umbilicus, and broadly D-shaped aperture; surface without noticeable pustules or spiral striae; off white.

*Operculum* absent.

*Radula* (Fig. 9E): Length of adult radular ribbon about 1.5 mm; up to 30 rows. Taenioglossate; rachidian and lateral teeth with 3–4 cusps on either side of strong central cusp; inner marginal tooth with long and strong main cusp, 2 smaller cusps on outer side,

and single longer cusp on inner side; slender outer marginal tooth with single cusp on inner side.

*External morphology and soft-body coloration* (Fig. 9B): Body stout, head-foot massive and fleshy; columellar muscle short. Cephalic tentacles short, triangular, with a tapered tip and a tiny black eye at the outer base. Both pedal tentacles reaching the same length and width (relatively thick and about 5–6 times longer than cephalic tentacles), but right tentacle usually much more active, further extended (length may exceed the diameter of the pedal disk) and appearing longer and wider than left tentacle; the exposed pedal disk with stacked, transverse, crescent-shaped ridges opaque whitish-yellow pigment in a translucent darker yellow matrix; mantle edge without pattern. Reproductive females with shallow anterior mantle slit to accommodate stalk of egg capsules. Color in life: with luminous lemon-yellow coloration.

*Development*: Females with stalked egg capsules.

*Sperm ultrastructure*: Not known.

*Habitat and ecology*: On the surface of dead areas of coral boulders, such as *Orbicella faveolata* (Ellis & Solander, 1786) and *Acropora palmata* (Lamarck, 1816), and on the underside of coral rocks that do not directly rest on sand. Sometimes found together with *Dendropoma cf. corrodens* (d'Orbigny, 1841). Live observations patch reefs and outer barrier reefs in about 2–7 m depth.

*Density*: Usually encountered singly.

*Molecular data*: Sequence data for two individuals (FMNH 326730, Lower Florida Keys; FMNH 327379, Upper Florida Keys) spanning mtDNA and nuclear gene regions (mtDNA: 12S-trnV-16S, nuclear: 28S) are available on GenBank. COI barcoding sequence data are also available for one specimen, FMNH 326730. See Table 1 for GenBank accession numbers. Low levels of genetic variation between these two specimens across the 16S (0.1% sequence difference) supports our interpretation that these individuals are conspecifics (Table 2).

*Distribution:* To date, living specimens are recorded only from the Florida Keys.

*Etymology: margarita*: Alluding to the vividly lemon-yellow coloration that this species shares with the citrus-juice-based cocktail drink (noun in apposition).

*Comparative and taxonomic remarks:*

*Cayo margarita* n. sp. shares its overall luminous "neon-like" appearance of the entire head-foot region and the "stacked crescent" pattern of the pedal disk with *C. galbinus* n. sp. (described below), which has a greenish instead of yellow coloration. *C. refulgens* n. sp. (see below) differs in having a mottled-white head-foot with thin yellow lines, and a shell with longitudinal ribbing. *C. brunneimaculatus* n. sp. (below) has a brown-and white body color pattern and also longitudinal ribbing on its adult teleoconch whorls. Our phylogenetic analyses based on a combined mtDNA and nuclear dataset support *C. margarita* n. sp. and *C. galbinus* n. sp. as sister taxa (Fig. 1).

***Cayo galbinus*** Bieler, Collins, Golding & Rawlings n. sp.
Figs. 10–13

*Type Locality:* Western Atlantic Ocean, Meso-American Barrier Reef, Belize, Stann Creek District, about 24 km off Dangriga; SW from boat launch at Carrie Bow Cay, 16°48.131′N, 088°4.951′W (collecting event BEL2011-003, 16 April 2011, on upper surface sides of coral boulder in shallow intertidal zone, 0–1 m depth, R Bieler, T Collins, R Golding, T Rawlings, P Sierwald).

*Material Examined:*

Type specimens: Holotype USNM 1688730 (ex FMNH 392794), largest diameter 4.6 mm (Fig. 10A).

Paratypes: Belize, Stann Creek District (about 24 km off Dangriga):

   FMNH 327151, type locality (used for DNA). FMNH 339499, lagoon and surrounding patch reefs at Carrie Bow Cay, 16° 48.131′N, 088°4.951′W [March 1986, 0–6 m depth, by snorkeling and SCUBA. R. Bieler & P. Sierwald] (live photo in Fig. 13B). FMNH 327144, SW from boat launch at Carrie Bow Cay, 16°48.131′N, 088°4.951′W (collecting event BEL2011-001, 14 April 2011, sides of coral boulder in shallow intertidal zone, 0–1 m depth, R Bieler, T Collins, R Golding, T Rawlings, P Sierwald) (incl. specimens in Fig. 10C (teleoconch sculpture), 10E (intracapsular larval shell), 11A, 13A (live photographs); also used for DNA. FMNH 327173, in lagoon east of Carrie Bow Cay, 16°48.165′N, 088°04.898′W (collecting event BEL2011-015, 21 April 2011, on upper surfaces of dead coral boulders, 0–1 m depth, R Bieler, T Collins, R Golding, T Rawlings, P Sierwald) (incl. specimen in Fig. 10D (internal shell sculpture) and used for DNA data). FMNH 327175, Tobacco Reef, 16°49.417′N, 088°04.873′W (collecting event BEL2011-022, 22 April 2011, coral rubble and boulders, 0–1.5 m depth, R. Bieler, T. Collins, R. Golding, T. Rawlings, P. Sierwald). FMNH 327183, in lagoon east of Carrie Bow Cay, 16°48.165′N, 088°04.898′W (collecting event BEL2011-022, on coral rubble and dead coral boulders, 24 April 2011; 0–1 m depth; R Bieler, T Collins, R Golding, T Rawlings, P Sierwald) (incl. specimens in Fig. 10F (radula), 11B (egg capsules), and Fig. 12 (sperm), and used for DNA). FMNH 327188, patch reef SSW of Carrie Bow Cay, 16°48′4.17″N, 88°4′56.23″W (collecting event BEL2011-023, 24 April 2011; SCUBA, 6 m depth; R. Bieler & P. Sierwald) (specimens in Fig. 10B). FMNH 327190, E of Carrie Bow Cay, 16°48.148′N, 088°04.885′W (collecting event BEL2011-025, on underside of coral rubble in upper intertidal zone, 25 April 2011; 0–1 m depth, R Bieler, T Collins, R Golding, T Rawlings, P Sierwald).

Other material studied (Belize, Stann Creek District, about 24 km off Dangriga):

   FMNH 327009, Tobacco Reef, 16°49.370′N, 088°04.765′W, entrenched on standing (in situ) branches of dead *Acropora palmata* coral, 31 March 1986; SCUBA 6 m depth, R Bieler & P Sierwald. Multiple specimens of *Cayo galbinus* n. sp., together with *Cayo refulgens* n. sp., *C. brunneimaculatus* n. sp. (see below), *Thylacodes decussatus*, *Dendropoma* cf. *corrodens*, and *Dendropoma nebulosum* (this material was used to count densities of several vermetid

species while still alive; the deeply entrenched specimens cannot be positively separated to species level in their preserved (retracted and leached) state without destructive sampling).

*Description*

*Teleoconch* (Figs. 10A–10E): Largest length of attached individual adult shell mass 8.5–13 mm, with most about 10.0 mm; either near-fully covered by coralline algal or coral growth (with often only the aperture visible on the surface) or forming a very regular spiral attached in one plane (in a Flemish flake pattern, with only some overlap between individual whorls), surface-attached shells somewhat entrenched into the substratum (with shallow basin excavated around aperture in specimens without upright feeding tube), with shell floor often very thin; usually with no or very short (2–3 mm) upright feeding tubes (Figs. 10A–10B); in cavities on underside of rocks occasionally developing long and tapering feeding tubes reaching up to 8 mm in length (Fig. 10C); largest attached whorl diameter 3.1–3.4 mm, with inner apertural diameter at that size tapering to 1.9–2.5 mm. Sculpture of narrow undulating axial (transverse) growth lines, without spiral (longitudinal) components (Fig. 10D); feeding tubes smooth except for growth marks. Interior shell wall in region of columellar muscle with minute, irregularly spaced pustules (Fig. 10E). Shell color general white, occasionally with some (usually faint) brown staining on parts of the inside shell wall; some specimens with small brown flecks on teleoconch surface (Fig. 10B).

*Protoconch* (Fig. 10F): Broadly helical shape, much wider than tall (height/width ratio ca. 0.7). Diameter 730–760 µm. About 1.5 rapidly expanding whorls, with final whorl smoothly rounded, forming umbilicus, and broadly D-shaped aperture; surface without noticeable pustules or spiral striae; off white.

*Operculum* absent.

*Radula* (Fig. 10G): Length of adult radular ribbon about 1.4 mm; up to 28 rows. Taenioglossate; rachidian and lateral teeth with 3–4 cusps on either side of strong central cusp; inner marginal tooth with long and strong main cusp, 2 smaller cusps on outer side, and single longer cusp on inner side; slender outer marginal tooth with single cusp on inner side.

*External morphology and soft-body coloration* (Fig. 11A): Body stout, head-foot massive and fleshy; columellar muscle short. Cephalic tentacles short, triangular, with a tapered tip and a tiny black eye at the outer base. Both pedal tentacles same length and width (relatively thick and about 5–6 times longer than cephalic tentacles), but right tentacle usually much more extended and appearing longer and wider than left tentacle. Reproductive females with shallow anterior mantle slit to accommodate stalk of egg capsules. Color in life: vivid lime-green (Fig. 11A): head-foot including tentacles and anterior mantle region are translucent luminous lime-green with delicate spots of opaque whitish-green pigment on pedal tentacles and forming a rim around the perimeter of the exposed foot surface; the exposed pedal disk with stacked, transverse, crescent-shaped ridges opaque whitish-green pigment in a translucent darker greenish matrix; mantle edge without pattern; juveniles often with orange colored concave regions flanking the foot. Female pallial slit rimmed with jet black pigment, gill entirely jet black. Juveniles often with orange colored concave lateral regions of the head-foot.
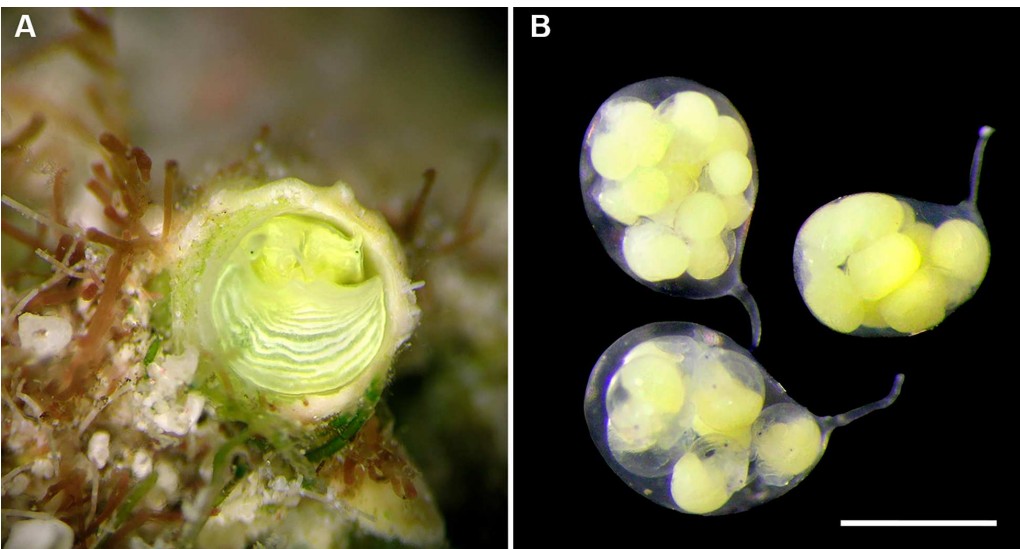

**Figure 11** **External morphology, living coloration, and egg capsules of *Cayo galbinus* n. sp.** (A) Living animal; note small black eyes on cephalic tentacles; pedal tentacles not extended (FMNH 327144, paratype, Belize station BEL2011-001); diameter of tube opening 2.1 mm. (B) Egg capsules from single female, showing range of developmental stages of stalked egg capsules; note the considerably off-center attachment of capsule stalk [FMNH 327183, Belize station BEL2011-022], scale = 1mm. Images by the authors (R Bieler).

*Development* (Fig. 11B): Multiple, stalked, ovoid to bean-shaped, egg capsules brooded in mantle cavity, with maximum length of 1.2 mm (without stalk), attached to female inner shell wall; stalk fairly long (about 1/3 of capsule length) and attached off-center; different developmental stage in single female, with early capsule stages containing up to 18 greenish yellow eggs, the latest stage with 10 crawl-away larvae with the vela completely resorbed; larval bodies completely white except for eye spots.

*Sperm ultrastructure* (Fig. 12): Euspermatozoa (Figs. 12A–12E): filiform and uniaxonemal, with flattened, elongate-conical acrosomal vesicle (length 1.4 μm with apical bleb), subacrosomal material organized as an axial rod (flattened in transverse profile) within vesicle invagination and as basal plate seated at apex of nucleus, vesicle invagination about half total length of vesicle, accessory membrane surrounding base of acrosomal vesicle; nucleus short and stocky (length 4.8 μm), helically-keeled, with short basal invagination (housing centriolar complex); axoneme surrounded anteriorly by multiple (7–8) helical mitochondria with irregular cristae (midpiece), then by putative glycogen granules (glycogen piece) and lastly by plasma membrane only (end piece). Paraspermatozoa (Figs. 12F, 12G): vermiform and multiaxonemal with large (2 μm maximum diameter), block-shaped dense vesicles (contents moderately and uniformly heterogeneous) sheathing four axonemes and numerous scattered mitochondria.

*Habitat and ecology* (Fig. 13): Entrenched in dead coral such as *in situ* branches of *Acropora palmata* (Lamarck, 1816), on the surface of dead areas of coral boulders, and

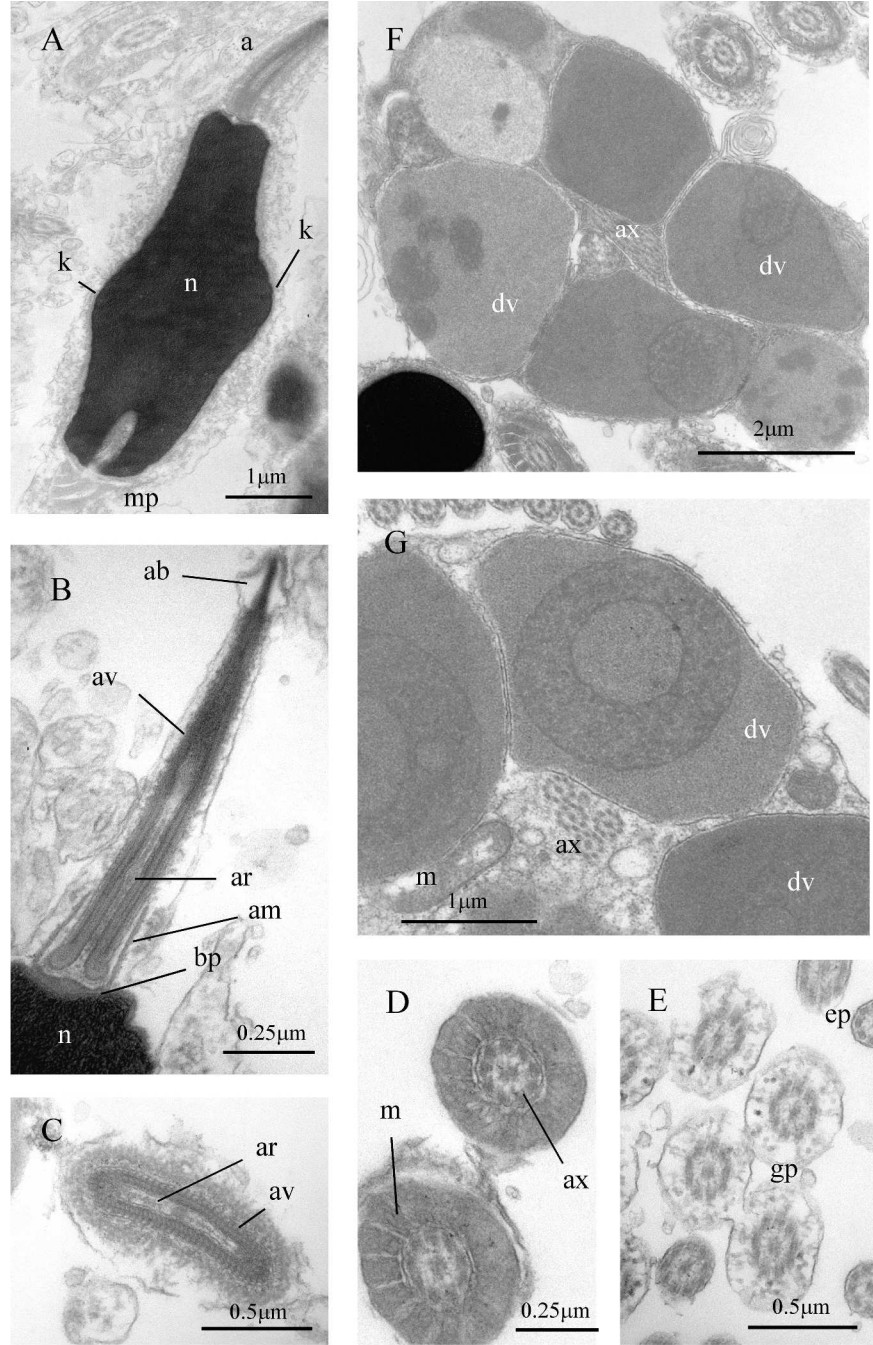

**Figure 12** **Sperm ultrastructure of *Cayo galbinus*. n. sp.** (A–E) Euspermatozoa. (A) Longitudinal section (LS) through acrosomal complex (a), nucleus (n) and proximal portion of midpiece (mp). Note helical keel (k) on nucleus. (B) LS acrosomal complex (acrosomal vesicle (av), axial rod component of subacrosomal material (ar); apical bleb (ab); basal plate component of subacrosomal material (bp)) situated on nuclear (n) apex. (C) Transverse section (TS) (continued on next page...)

Figure 12 (…continued)
acrosomal vesicle and subacrosomal material. Note flattened profile of vesicle. (D) TS and oblique TS midpiece, showing helical mitochondria. (E) TS glycogen piece (gp) and end piece (ep). (F–G) Paraspermatozoa. (F) TS main body of cell showing four axonemes (ax) enclosed by large dense vesicles (dv). Dense vesicles show spherical differentiation zones and random dark inclusions. (G) TS showing detail of axonemes (ax), dense vesicles (dv) and mitochondrion (m). All from paratype lot FMNH 327183, Belize. Images by the authors (J Healy).

on the underside of dead coral rocks that do not directly rest on sand. When exposed on surface, frequently forming a perfect circle. Often, with shallow basin excavated around aperture (also mentioned above). In Belize (Carrie Bow Cay and Tobacco Reef) co-occurring on the same piece of dead coral with other vermetids including C. *refulgens* n. sp. and *C. brunneimaculatus* n. sp. (see below), *Cupolaconcha guana* (*Golding et al., 2014*), *Dendropoma cf. corrodens* (d'Orbigny, 1841), and *Petaloconchus* sp.  Live observations from reef crests, patch reefs, and outer barrier reefs in 2–7 m depth.

  *Density*: Usually encountered singly or in loosely spaced aggregations of 2–10 per boulder or large *Acropora* branch.

  *Molecular data*: Sequence data for two individuals (FMNH 327151, Belize; FMNH 327173, Belize) spanning mtDNA and nuclear gene regions (mtDNA: 12S-trnV-16S, COI; nuclear: 28S) are available on GenBank. See Table 1 for GenBank accession numbers. Low levels of genetic variation (<0.6%) between these two specimens across COI and 16S gene regions supports our interpretation that these individuals are conspecifics (Table 2).

*Distribution:* To date, living specimens are recorded only from the Belizean barrier reef.

*Etymology: galbinus, -a, -um*: Latin for greenish-yellow, referring to the vivid head-foot coloration (adjective).

*Comparative and taxonomic remarks:*
*Cayo galbinus* n. sp. shares its overall luminous "neon-like" appearance of the entire head-foot region and its pedal disk pattern of whitish stacked crescents with *C. margarita* n. sp. The latter has a head-foot coloration in vivid lemon color, whereas *C. galbinus* is more greenish lime-colored. Shells of the two species are very similar, with *C. galbinus* occasionally displaying a brown fleck pattern on the dorsum. The spiral keel on the laterally flattened sperm nucleus is unique to date (no comparative sperm data from *C. margarita* are available). Compare to *C. refulgens* n. sp. (see below), which has a mottled-white head-foot with thin yellow lines, a shell with longitudinal ribbing, and different sperm ultrastructure. *C. brunneimaculatus* n. sp. (below), which has a brown-and white body color pattern, likewise has longitudinal ribbing on its adult teleoconch whorls. Corresponding with the morphological observations described above, our molecular phylogenetic analyses based on a combined mtDNA and nuclear dataset support a sister group relationship between *C. galbinus* n. sp. and *C. margarita* n. sp. (Fig. 1).

***Cayo refulgens*** Bieler, Collins, Golding & Rawlings n. sp.
Figs. 14–16

*Type Locality :* Western Atlantic, Meso-American Barrier Reef, Belize, Stann Creek District, about 24 km off Dangriga; W of Carrie Bow Cay, far off boat launch, 16°48.390′N, 088°04.980′W, 1–1.5 m depth (collecting event BEL2011-026; 26 April 2011; R. Bieler, T. Collins, R. Golding, T. Rawlings, P. Sierwald).

*Material Examined:*

Holotype:

   USNM 1688731 ex FMNH 327193 (Fig. 14A); type locality as above; best-preserved shell with teleoconch covered by coral growth (early whorls missing); body of male animal in 70% ethanol.

Paratypes (Belize):

   FMNH 327171, Stann Creek District, about 24 km off Dangriga; in lagoon east of Carrie Bow Cay, on upper surfaces of dead coral boulders, 16°48.165′N, 088° 4.898′W [collecting event BEL2011-015; 0–1 m depth, 21 April 2011, R. Bieler, T. Collins, R. Golding, T. Rawlings, P. Sierwald] (specimen in Fig. 14C (inside sculpture), 15A, B (live photos), 16 (sperm); also source of DNA data). FMNH 326929 lagoon and surrounding patch reefs at Carrie Bow Cay, 16°48.131′N, 088°4.951′W (March 1986, 6 m depth, by SCUBA. Bieler & P. Sierwald) (specimens in Figs. 14B, 14D, 14E (teleoconch fragments, protoconch, radula).

Other material studied:

   FMNH 327009, Belize, Stann Creek District, about 24 km off Dangriga; Tobacco Reef, entrenched on standing (in situ) branches of dead *Acropora palmata* coral, 16° 49.370′N, 088°04.765′W, 31 March 1986; 6 m depth; multiple specimens of *Cayo refulgens* n. sp., together with *Cayo galbinus* and *C. brunneimaculatus* n. spp., *Thylacodes decussatus, Dendropoma* cf. *corrodens*, and *Dendropoma nebulosum* (this material was used to count densities of several vermetid species while still alive; the deeply entrenched specimens cannot be positively separated to species level in their preserved (retracted and leached) state without destructive sampling). FMNH 335073, Bahamas, reefs NE of Great Abaco Island, outside of barrier reef off Walker's Key ''Grouper alley''; 26°38.457′N, 077°02.245′W [collecting event BAH2012-008; by SCUBA, 9.1–12.2 m depth, 08-VIII-2012, R. Bieler, P. Sierwald]. This material from the Bahamas appears conspecific based on live animal coloration and overall shell morphology. Additional material/sequence data are needed.

*Description*

*Teleoconch* (Figs. 14A–14C): Largest length of attached individual adult shell mass 9–13 mm; fully covered by coralline algal or coral growth (with often only the aperture visible on the surface; Fig. 14A) or forming a very regular spiral attached in one plane (in a Flemish flake pattern, with only some overlap between individual whorls), surface-attached shells somewhat entrenched into the substratum; with no or very short (1–2 mm) upright feeding tubes; largest attached whorl diameter 3.0–3.4 mm, with inner apertural diameter at that size tapering to 1.8–2.3 mm; early sculpture of narrow undulating axial (transverse) growth

lines, with ca. five dominant spiral (longitudinal) cords on adult whorls (Fig. 14B). Fully coral-embedded parts of shell often very thin. Shell white with partial dark brown coloration both on outside and inside, white surface areas occasionally with small brown flecks; often appearing greenish due to algal growth. Interior shell wall in region of columellar muscle with minute, irregularly spaced pustules (Fig. 14C).

*Protoconch* (Fig. 14D): Broadly helical shape, wider than tall. Diameter (from very limited material) about 700 µm. Nearly two rapidly expanding whorls, with final whorl smoothly rounded, forming umbilicus, and broadly D-shaped aperture; surface without noticeable pustules or spiral ribs; off white.

*Operculum* absent.

*Radula* (Fig. 14E): Length of adult radular ribbon about 1.5–2.0 mm; up to 29 rows. Taenioglossate; rachidian with 3 and lateral teeth with 3–5 cusps on either side of strong central cusp; inner marginal tooth with long and strong main cusp, 1–2 small cusps on outer side, and single longer cusp on inner side; slender outer marginal tooth with single cusp on inner side.

*External morphology and soft-body coloration* (Figs. 15A, 15B): Body stout, head-foot massive and fleshy; columellar muscle short. Reproductive females with shallow anterior mantle slit to accommodate stalk of egg capsules, which are brooded in the mantle cavity and attached to the inside wall shell tube. Color in life: Anterior parts of head-foot mottled white with large white pigment spots; dorsal head surface with faint mottling of ginger-brown pigment. Mouth, perimeter of pedal disk, and small black eyes on cephalic tentacles outlined by a thin line of luminous, neon yellow pigment. Tentacles translucent white, with only pedal tentacles having distinct small pigment granules; peripheral regions of foot, posterior head, and especially concave zones, with orange spots; interior surface of anterior mantle margin with alternating pattern of white and ginger-brown pigment followed by speckles of pale yellow and ginger-brown; area below posterior foot, posterior head, and area of inner mantle posterior of mantle edge almost solid white; dorsal mantle surface with dense superficial white pigment except where the intestine forms a sinuous outline.

*Development*: Multiple, stalked, ovoid, egg capsules brooded in mantle cavity, with maximum length of 1.2 mm, attached to female inner shell wall; stalk short and attached off-center; early capsule stages not observed, the latest stage with 10 whitish crawl-away larvae with the vela completely resorbed; larval bodies completely white except for eye spots.

*Sperm ultrastructure* (Fig. 16): Euspermatozoa (Figs. 16A-16E): filiform and uniaxonemal, with flattened, elongate-conical acrosomal vesicle (length 1.7 µm, with apical bleb), subacrosomal material organized as axial rod (flattened in transverse profile) within vesicle invagination and as basal plate seated at nuclear apex, vesicle invagination about half total length of vesicle, accessory membrane surrounding base of acrosomal vesicle; nucleus slender, rod-shaped (length 8.0 µm), with short basal invagination (housing centriolar complex); axoneme surrounded anteriorly by multiple (7–8) helical mitochondria with irregular cristae (midpiece), then by putative glycogen granules (glycogen piece) and lastly by plasma membrane only (end piece). Paraspermatozoa (Figs. 16F, 16G): vermiform

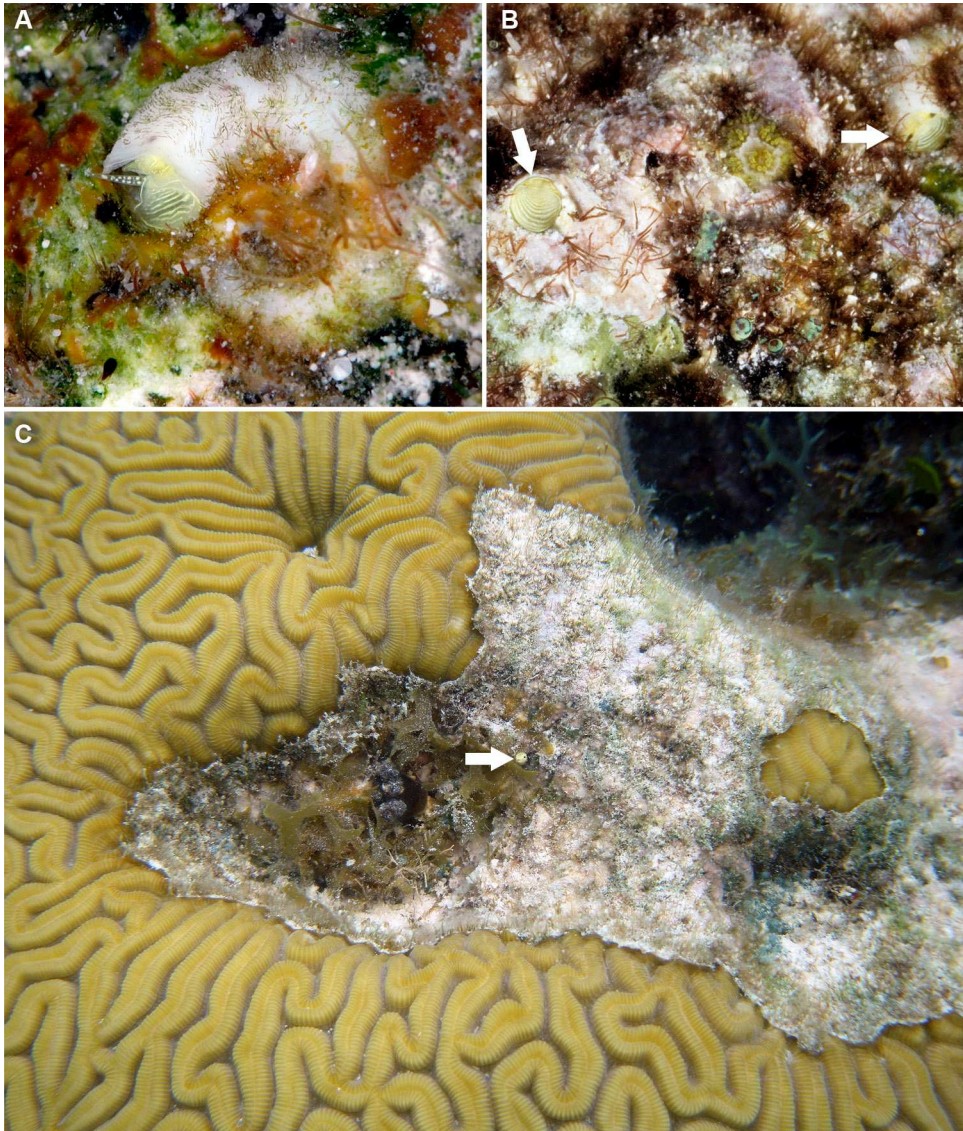

**Figure 13 Habitat photographs of *Cayo galbinus.* n. sp.** (A) Solitary animal with upper side of white shell exposed; note dorsal keel on very regular shell coil and the single extended pedal tentacle filled with white pigment granules (FMNH 327144; Belize station BEL2001-001); diameter of tube opening ca. 2.1 mm. (B) Two specimens (arrows) deeply entrenched on dead branch of branching elkhorn coral *Acropora palmata* (Lamarck, 1816), Carrie Bow Cay, Belize, 1986 [FMNH 327009]; diameter of tube opening ca. 2.0 mm. (C) Solitary animal (arrow) in typical habitat, dead hard-substratum areas within and between hard corals (here the grooved brain coral, *Diploria labyrinthiformis* (Linnaeus, 1758)); Belize, Stann Creek District, about 24 km off Dangriga; shallow patch reefs, 16°48.390′N, 088°4.980′W (station BEL2011-012), underwater photograph in 2 m; 19 April 2011; specimen not collected; diameter of tube opening ca. 2.0 mm. Images by the authors (R Bieler).

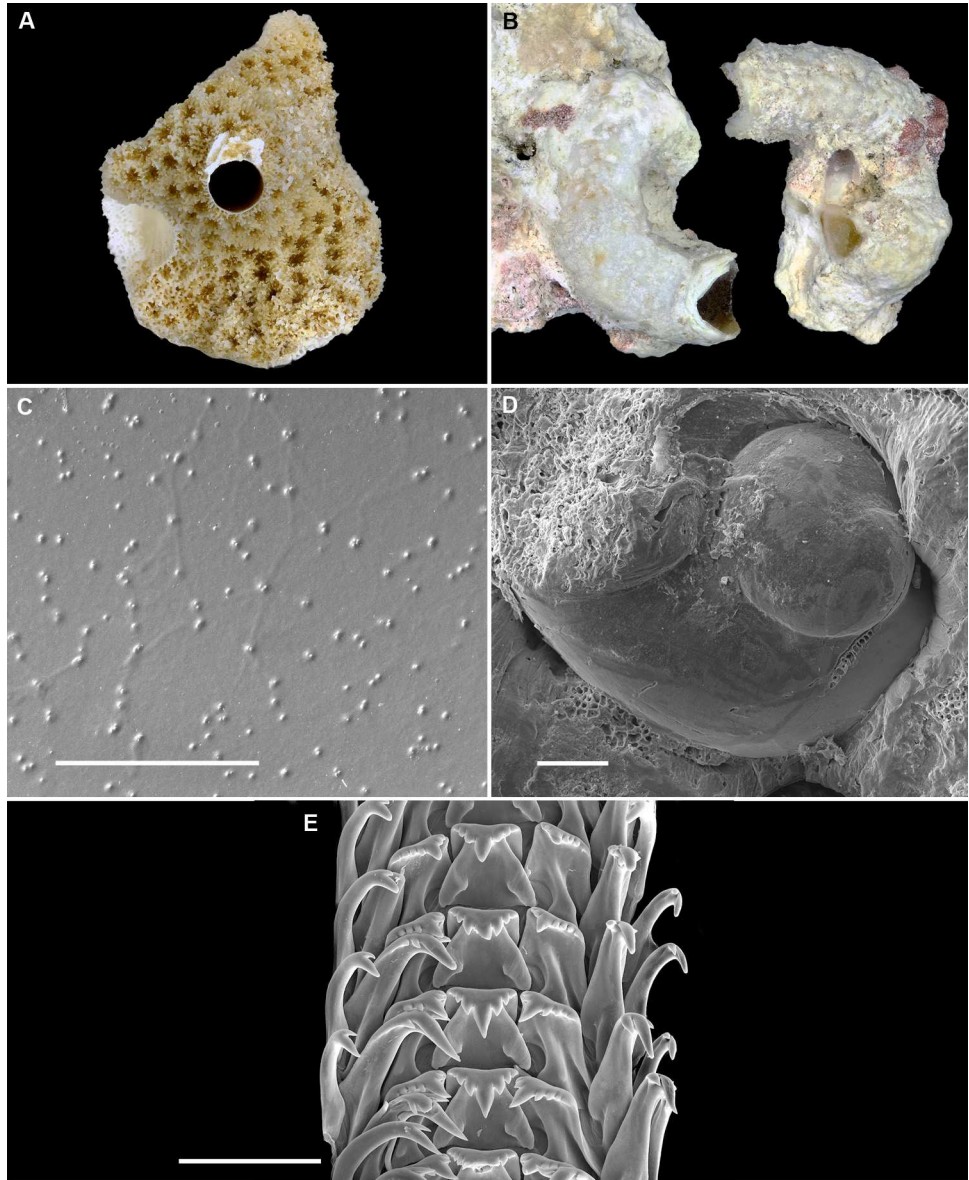

**Figure 14** **Shell morphology and radula of _Cayo refulgens_ n. sp. (Belize).** (A) Holotype [USNM 1688731 ex FMNH 327193]; apertural view of shell deeply embedded in and overgrown by coral, likely _Porites_ sp.; diameter of opening, 2.3 mm. Scale 2 mm. (B) Detail of teleoconch sculpture [FMNH 326929, 2 fragments of same paratype], scale 2 mm. (C) Internal shell sculpture in columellar region (SEM) [FMNH 327171], scale = 100 µm. (D) Shell fragment with protoconch exposed (SEM) [FMNH 326929], scale = 100 µm. (E) Partial aspect of adult radula (SEM) [FMNH 326929, Belize], scale = 100 µm. Images by the authors (R Bieler and R Golding).

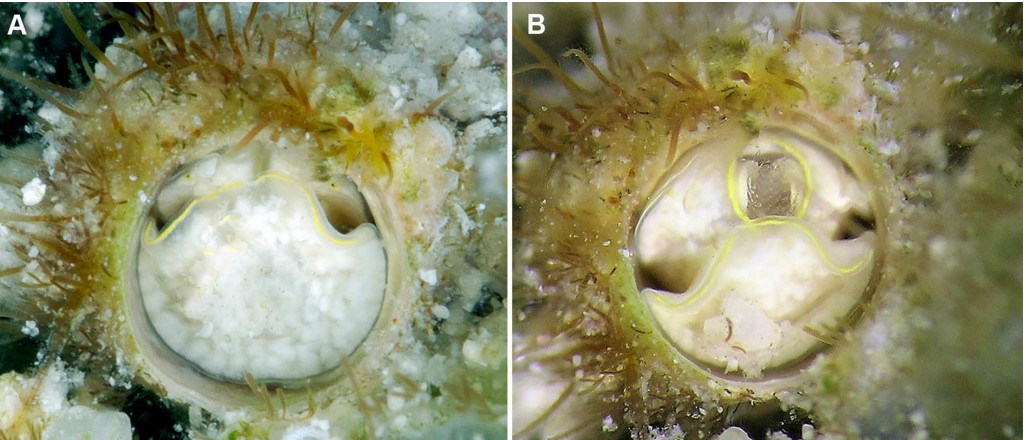

**Figure 15** **External morphology and living coloration of *Cayo refulgens* n. sp.** (A) Living animal showing yellow lining of pedal disk and black eye spots surrounded by yellow pigment on cephalic tentacles; pedal tentacles withdrawn (paratype FMNH 327171, station BEL2011-015, Belize); diameter of tube opening ca. 2.1 mm. (B) Same specimen, exposing yellow-lined mouth region with extended radula in the process of retrieving mucus net strands. Images by the authors (R Bieler).

and multiaxonemal with large (1.5 µm maximum diameter), block-shaped dense vesicles (contents exhibiting spherical zones of differing electron-density) sheathing four axonemes and scattered mitochondria.

*Habitat and ecology* (Fig. 14A)*:* Deeply entrenched in dead coral such as *in situ* branches of *Acropora palmata*, on the surface of dead areas of coral boulders (often covered by algal turf). In the Belizean reef co-occurring on the same piece of dead coral with C. *galbinus* n. sp. (see above). Live observations from the shallow reef crest to about 12 m.

*Density*: Usually encountered singly or in loosely spaced aggregations of 2–5 per boulder or large *Acropora* branch.

*Molecular data*: Sequence data for one individual (FMNH 327171, Belize) spanning mtDNA and nuclear gene regions (mtDNA: 12S-trnV-16S, COI; nuclear: 28S) is available on GenBank. See Table 1 for GenBank accession numbers. High levels of sequence difference between this species and other *Cayo* species described herein based on COI and 16S gene regions supports our species-level designations of these taxa (Table 2).

*Distribution:* Living specimens known from the Belizean barrier reef (off Dangriga) and the Bahamas (Great Abaco Island, no DNA data). Empty shells likely belonging to this species are known from the Florida Keys (*e.g.*, FMNH 344629, Dry Tortugas), and this species is probably widely distributed in the Caribbean realm.

*Etymology: refulgens*: "shining back" or, figuratively, "standing out" (used as adjective), here referring to the luminous yellow lining surrounding the pedal disk and mouth region.

*Comparative remarks:*

*Cayo refulgens* n. sp. is similar to *C. margarita* and *C. galbinus* n. spp. in overall size and appearance in the reef, especially when deeply entrenched or overgrown. Exposed shells

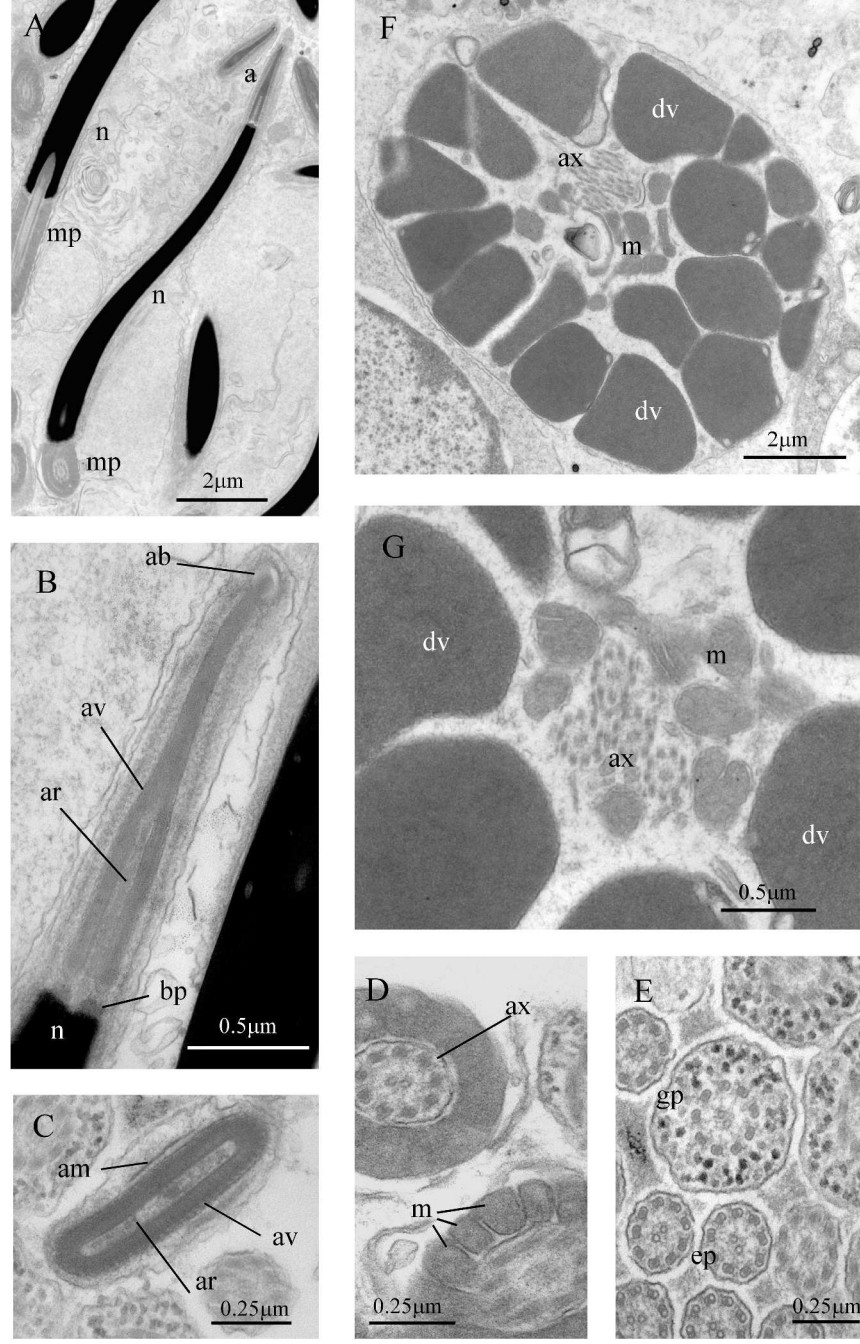

**Figure 16** **Sperm ultrastructure of *Cayo refulgens.* n. sp.** (A–E) Euspermatozoa. (A) Longitudinal section (LS) through acrosomal complex (a), nucleus (n) and proximal portion of midpiece (mp). (B) LS acrosomal complex (acrosomal vesicle (av), axial rod component of subacrosomal material (ar); apical bleb (ab); basal plate component of subacrosomal material (bp)) 

differ in ornamentation (longitudinal ribbing in adult *C. refulgens*, undulating transverse growth lines in *T. margarita*). Living animals can readily be distinguished by the yellow coloration that is limited to a narrow lining of the otherwise mottled-white pedal disk and mouth region (in contrast to the overall luminous yellow or green body color in *C. margarita*). *C. brunneimaculatus* n. sp. (below) lacks yellow pigmentation in the head-foot area. Eusperm of *C. refulgens* lacks a spiral keel on the nucleus that is present in *C. galbinus* (see section on sperm ultrastructure, below). Molecular-based phylogenetic analyses strongly support *Cayo* n. gen. as a monophyletic grouping, with *C. refulgens* n. sp. as sister to a clade comprising *C. brunneimaculatus n. sp.*, *C. galbinus n. sp.*, and *C. margarita n. sp.* (Fig. 1).

### *Cayo brunneimaculatus* Bieler, Collins, Golding & Rawlings n. sp.
### Fig. 17

*Type Locality:* Western Atlantic Ocean, Meso-American Barrier Reef, Belize, Stann Creek District, about 24 km off Dangriga; patch reef SSW of Carrie Bow Cay, 16° 48′4.17″N, 88°4′56.23″W, in 6 m depth (collecting event BEL2011-023, 24 April 2011; by SCUBA, R Bieler & P Sierwald).

*Material Examined* (all from Belize)*:*
Holotype
    USNM 1688732 ex FMNH 327187, holotype shell (Fig. 17A); molecular data from this female specimen.
Paratypes:
    FMNH 326933 Belize, Stann Creek District, about 24 km off Dangriga; Tobacco Reef, entrenched on standing (in situ) branches of dead *Acropora palmata* coral, 16° 49.370′N, 088°04.765′W, 31 March 1986; by SCUBA, 6 m depth, R. Bieler & P. Sierwald] (2 specimens; animal in live photo (Fig. 17B), radula (Fig. 17D), intracapsular larval shells (Fig. 17C), egg capsules (Fig. 17E)).
Other material studied:
    FMNH 327009 (same data as FMNH 326933), with multiple specimens of *Cayo margarita* and *C. refulgens* n. spp., together with *Thylacodes decussatus*, *Dendropoma* cf. *corrodens*, *Dendropoma nebulosum* (this material was used to count densities of several vermetid species while still alive; the deeply entrenched specimens cannot be positively

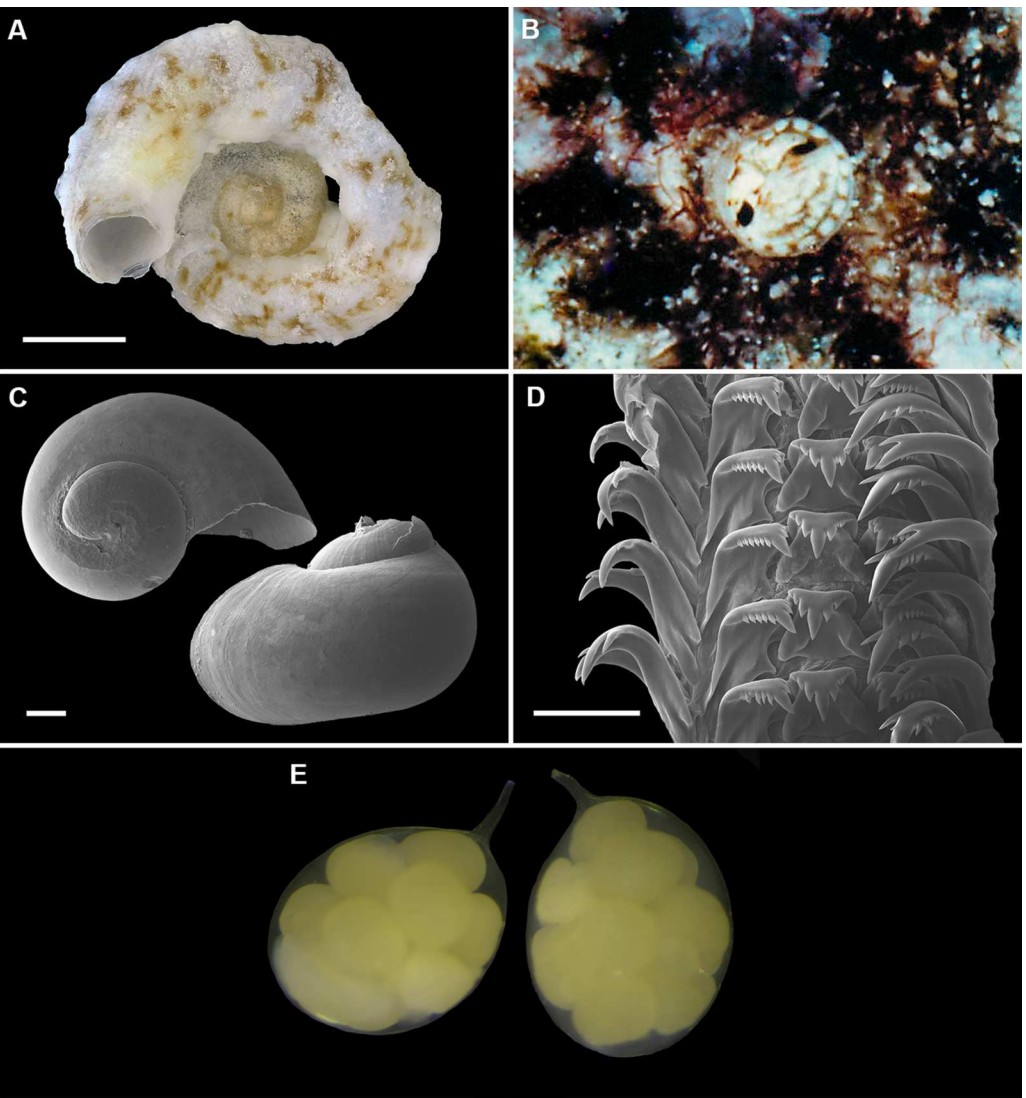

**Figure 17** **Shell morphology, living coloration, radula, and egg capsules of *Cayo brunneimaculatus* n. sp. (Belize).** (A) Holotype specimen [USNM 1688732 ex FMNH 327187]; largest dimension 8.0 mm; diameter of tube opening 1.7 mm. Scale = 2 mm. (B) Living specimen in situ, entrenching on dead *Acropora palmata* branch, Tobacco Reef, Belize, diameter of tube opening ca. 1.9 mm. (C) Very thin-shelled intra-capsular larval shells, slightly damaged (SEM); scale = 100 μm. (D) Partial aspect of adult radula (SEM); scale 100 μm. (E) Egg capsules from single female, both with eggs in early developmental stages; length of right capsule, without attachment stalk, 1.2 mm. (B–E) from paratype lot FMNH 326933, Belize. Images by the authors (R Bieler and R Golding).

separated to species level in their preserved (retracted and leached) state without destructive sampling)

*Description*

*Teleoconch* (Figs. 17A, 17B): Largest length of attached individual adult shell mass up to 8 mm (7.8 in holotype); fully covered by coralline algal or coral growth (with only the

aperture visible on the surface) or forming a very regular spiral attached in one plane (in a Flemish flake pattern, with only some overlap between individual whorls; Fig. 17A); with no or very short (1 mm) upright feeding tube (Fig. 17B); largest attached whorl diameter 2.5 mm, with inner apertural diameter at that size tapering to 1.6 mm (holotype); sculpture of irregular axial (transverse) growth marks and 5–6 dominant but low spiral (longitudinal) cords on dorsum of whorls. Interior shell wall in region of columellar muscle with minute, irregularly spaced pustules. Shell white, irregularly flecked with brown; outside fleck pattern reflected on inside shell walls.

*Protoconch* (Fig. 17C): Broadly helical shape, much wider than tall (height/width ratio ca. 0.7). Diameter (from very limited material) ca. 760 μm. About two rapidly expanding whorls, with final whorl smoothly rounded, forming umbilicus, and broadly D-shaped aperture; surface without noticeable pustules or distinct spiral ribs; yellowish white.

*Operculum* absent.

*Radula* (Fig. 17D): Length of adult radular ribbon 1.7 mm; 25 rows (single animal studied). Taenioglossate; rachidian teeth with three sturdy cusps on either side of strong central cusp (some teeth with smaller fourth cusp interspersed); lateral teeth with five cusps on the outer side and 3–4 on inner side of strong central cusp; inner marginal tooth with long and strong main cusp, three cusps on outer side, and single longer cusp on inner side; slender outer marginal tooth with single cusp on inner side.

*External morphology and soft-body coloration* (Fig. 17B): Body stout, head-foot massive and fleshy; columellar muscle short. Reproductive females with shallow anterior mantle slit to accommodate stalk of egg capsules, which are brooded in the mantle cavity and attached to the inside wall of the shell tube. Living animal often with pedal disk oriented toward interior shell wall. Color in life: Base color of head and foot somewhat translucent white, with pedal disk (and especially its lower edge) mottled with patches of brighter and more solid white pigment; base of head tentacles and various areas of foot with chestnut brown markings, occasionally with two brown dots in the concave area above the head tentacles; pedal disk occasionally appearing light orange-brown due to fine brown surface pigment and white and slightly orange deeper body color shining through; 2–5 fine brown lines running toward mouth opening; pedal tentacles very strong and long (4 times the length of head tentacles), glossy with many white (at base with few orange) pigment granules; mantle edge with alternate white/chestnut brown pattern; lateral area of head, posterior part of foot, and interior mantle white with orange tint; exterior mantle yellowish posterior to mantle edge. Brooding females with long mantle slit, bordered by orange coloration.

*Development* (Fig. 17E): Multiple, stalked, ovoid, egg capsules brooded in mantle cavity, with maximum length of 1.4 mm, attached to female inner shell wall; stalk short and attached off-center; different developmental stage in single female, with early capsule stages containing 12–16 yellow eggs, the latest stage 5 crawl-away larvae with the vela completely resorbed; larval bodies completely white except for eye spots.

*Sperm ultrastructure*: Not known.

*Habitat and ecology:*

Entrenched in dead coral such as *in situ* branches of *Acropora palmata* (Lamarck, 1816). In Belize (Carrie Bow Cay and Tobacco Reef) co-occurring on the same piece of dead coral
with other vermetids including *C. margarita* n. sp., C. *refulgens* n. sp., *Cupolaconcha guana*, and *Dendropoma cf. corrodens.* Live observations from 6 m depth.

    *Density*: Rarely encountered and occurring singly.

*Molecular data*: Sequence data for one individual (FMNH 327187, Belize) spanning mtDNA and nuclear gene regions (mtDNA: 12S-trnV, 16S; nuclear: 28S) is available on GenBank. See Table 1 for GenBank accession numbers. High levels of sequence difference between this species and other *Cayo* species described herein based on a region of 16S supports our species-level designations of these taxa (Table 2).

*Distribution:* Only known from Belizean barrier reef.

*Etymology: brunneimaculatus, -a, -um*: from Latin brunneus (brown) and maculatus (spotted, blotched) (used as an adjective). Referring to the brown pattern in both the soft body and teleoconch coloration in this species.

*Comparative and taxonomic remarks: C. brunneimaculatus* n. sp. shares spiral (longitudinal) sculpture of the adult shell with *C. refulgens* n. sp. The brown-and-white coloration of both headfoot and mantle margin separates it from the three known congeners. In molecular phylogenetic analyses based on a combined mtDNA and nuclear dataset, *C. brunneimaculatus* n. sp. is strongly supported as a member of the *Cayo* n. gen. lineage, and sister to a clade comprising *C. galbinus n. sp.* and *C. margarita n. sp.* (Fig. 1).

## DISCUSSION

### Relationships/Phylogenetic position

Our molecular results have helped to confirm the species-level status of several new vermetid specimens examined herein from the western Atlantic and to clarify their relationships to members of other major vermetid lineages. A major outcome of these phylogenetic analyses has been the discovery of a new lineage of non-operculate vermetids –*Cayo* n. gen –and its sister group relationship to a lineage of vermetids with a small button-like operculum, *Vermetus*. Given that the presence of an operculum across all life-history stages (larva, juvenile, adult) is the inferred ancestral condition within the Vermetidae (*Morton, 1965*), the most parsimonious explanation based on our ML tree topology is that the operculum has been lost twice independently within this family, once along the lineage leading to *Cayo* and separately along a lineage leading to *Thylacodes* (Fig. 1 with arrows showing inferred state change). Clearly, other interesting dynamics with respect to changes in the operculum size, shape, and mineral content have occurred during the radiation of vermetids (*Morton, 1965*). More detailed explorations of evolutionary changes in the operculum and other relevant morphological/anatomical features await completion of our comprehensive sampling of vermetid species worldwide and the generation of a robust phylogenetic hypothesis for the entire family. Intraspecific genetic distances were modest relative to interspecific differences within *Cayo* and *Thylacodes* clades, with ≤ 1.1% sequence divergence within species for which more than one specimen was sampled (Table 2). In contrast, interspecific differences were substantial within both *Cayo* and *Thylacodes* clades, reaching up to 34.8% divergence among *Cayo* species and 30.2% divergence among

*Thylacodes* for COI, supporting the considerable morphological/anatomical variation observed within each clade. Our genetic distance comparisons among taxa have also been important in illustrating the extent of foot/mantle color variation within the newly described species, *Thylacodes bermudensis*. Like several other *Thylacodes* species, including our recently described *Thylacodes vandyensis* (*Bieler et al., 2017a*; *Bieler et al., 2017b*), *T. bermudensis* can be recognized as polychromatic, with population-level variation in foot and mantle pigmentation patterns. Our field observations did not uncover any obvious association between color morph and habitat, however, with different color morphs living side by side on the same coral or other hard substratum (Fig. 3A). Interestingly, polychromatism does not appear to be a feature of *Cayo* species: color variation was limited within each newly described species. Finally, as with our previous investigations of vermetid snails (*e.g.*, *Golding et al., 2014*; *Bieler et al., 2017a*), detailed morphological/molecular examinations of focal vermetid groups continue to uncover unrecognized diversity with the circumscribed geographic regions that we have explored (see *Unrecognized vermetid diversity within the Caribbean*, below).

## Loss of operculum

Operculum loss in *Thylacodes* occurs at the transition from a mobile larva form to a sessile attached juvenile stage, with the operculum jettisoned at metamorphosis (*Morton, 1951*; *Hughes, 1978*; *Calvo & Templado, 2005*). And, although we now have evidence to indicate that this loss has happened independently in a second lineage leading to *Cayo*, little is known about the selective advantages, if any, associated with a reduction or loss of the operculum in juvenile/adult vermetids. *Yonge & Iles (1939)* suggested that operculum reduction/loss evolved in association with the elaboration of mucus-net feeding in vermetids, since the edge of the operculum can impede the distribution of mucus threads. In support of this, mucus-net feeding is generally considered to be more fully developed and widely employed in *Vermetus* and *Thylacodes,* two genera with reductions/loss of the operculum, compared to other vermetid genera (*Morton, 1965*). Somewhat counterintuitively, operculum reduction and loss also have the potential to facilitate predator evasion, since rapid, deep retreat into the shell tube is only possible through the compressibility of the foot and the absence of a large, attached structure, the operculum, which can act as a physical impediment to withdrawal (*Morton, 1965*). In fact, *Morton (1965)* noted that deep retreat into the shell tube by *Thylacodes* could be "effected with lightning rapidity when the animal is disturbed". Likewise, *Calvo & Templado (2005)* determined that camouflage, through epizoic overgrowth, and rapid and deep retreat into the shell tube are likely effective protective mechanisms against predation in the solitary vermetid *Thylacodes arenarius* (Linnaeus, 1758). The effect of retraction may also depend on how deep the shell is embedded in the substratum, as some fish (*e.g.*, parrotfish, Bieler, 1983–2023, pers. obs.) and crab (*e.g.*, *Shlesinger, Akkaynak & Loya, 2021*) can readily break vermetid shells. The absence of an operculum does have some obvious disadvantages, however. Non-operculate juvenile and adults lack physical protection of soft tissues and the ability to seal off the aperture of their tube-like shells, thus rendering their exposed head/foot vulnerable to

predators and temporary physical stresses, such as excessive siltation or (especially in shallow-water settings) salinity changes.

Predation pressure is likely to be considerable within the shallow-water reef habitats where most vermetids are found (*Jones, Ferrell & Sale, 1991*), with known vermetid predators including fish, sea stars, crabs, octopus and neogastropods (*e.g., Menge et al., 1986*; *Osman, 1987*; *Calvo & Templado, 2005*; *Ramírez et al., 2013*; *Shima, Phillips & Osenberg, 2016*; *Brown et al., 2014*; *Shlesinger, Akkaynak & Loya, 2021*). Even if not directly targeted as a prey item, sessile vermetids are also likely to be constantly exposed to grazing activities by parrotfish and others. In such a high-predation environment, how can they afford to give up the additional safety that a close-fitting apertural seal can provide? In contrast to other soft-bodied species in the same habitat (*e.g.*, sponges, tunicates, sea anemones, nudibranch slugs), vermetids lack nematocysts or known chemical defenses in their exposed tissues. In fact, once a shell is broken open, vermetids are consumed quickly by surrounding fish (Bieler, 1983–2023, pers. obs. during SCUBA diving). Instead, their primary defense might be the presence of (at least) unpalatable mucus used in mucus-net feeding, which is often widely spread around the shell aperture during feeding activities (Fig. 4B). Although laden with trapped plankton and detritus and itself consisting of mucopolysaccharides and mucins (glycosylated proteins), fish appear to avoid this mucus, even those known to be planktivores and/or mucivores (see *Coles & Strathmann, 1973*; *Klöppel et al., 2013*; *Bieler et al., 2017a*). Mucus liberated during mucus-net feeding by some species also has a deleterious effect on surrounding coral growth (*e.g.*, *Shima, Phillips & Osenberg, 2013*; *Hoeksema et al., 2022*), resulting visually in a "halo effect" surrounding shell apertures not only affecting scleractinian corals but also zoantharians (see *Thylacodes decussatus* on *Palythoa* sp.: Fig. 4C). These effects may be attributable to vermetid-derived bioactive metabolites documented in the mucus of the large-bodied operculate *Ceraesignum maximum* (*Sowerby, 1825*), the only species examined for such bioactive properties to date (*Klöppel et al., 2013*). Given, therefore, that *Thylacodes* species typically rely on mucus-net feeding as their primary source of nutrition, the production of unpalatable mucus has the potential to provide an effective means of defense. Likewise, if visual predators such as fish link this mucus to the strong head/foot coloration of *Thylacodes* and *Cayo* species, foot color can evolve as an aposematic signal in these vermetids (see *Head-foot coloration*, below).

Operculum loss has occurred repeatedly during the evolution of another conspicuous family of sessile benthic marine invertebrates—the serpulid worms. *Kupriyanova, Ten Hove & Rouse (2023)* estimate that the operculum has been lost four times independently within the Serpulidae and regained once. Selective pressures associated with operculum loss in this group are unclear, but ten Hove and Kupriyanova (2009) have speculated that it could be associated with the development of close relationships with "protective" associates, such as sponges and corals, well endowed with chemical and mechanical defenses. Other hypotheses include morphological innovations inside the worm tube, such as the "grab-footholds" of *Spiraserpula,* which could facilitate more rapid retreat from predators. Operculum loss in serpulids, however, does not appear to be associated with a heightened coloration of exposed body regions—at least not to the human eye: serpulids are brightly

colored, in general, with considerable intra and interspecific variation in color patterns (*Ten Hove & Kupriyanova, 2009*). Among vermetids, there is little evidence to suggest that non-operculate species have stronger affiliations with protective associates compared to operculate species, except perhaps for *Thylacodes squamigerus* (P. P. Carpenter, 1857). *Osman (1987)* determined that *Thylacodes squamigerus* had a commensal, albeit facultative, relationship with encrusting cheilostome bryozoans, with newly settled juvenile vermetids deriving a survival advantage from their association with these bryozoans relative to larvae that settled on other substrates. The direct cause of increased survival was not determined but could have resulted from the benefits of greater camouflage, decreased competition for space and food, or through fortuitous protection by the avicularia (protective zooids) of the bryozoans. These protective benefits were relatively short-lived, however, with no influence of substratum type on mortality rates once vermetids were >3 months of age. Given the close proximity of vermetids to other sessile benthic organisms with well-established chemical and mechanical defensives, however, the potential exists for vermetids to be deriving some protective benefit from their neighbors. It remains to be determined, however, whether non-operculate vermetids, lacking the physical protection of the operculum, could be exploiting their benthic neighbors in different ways from their operculate congeners.

## Head-foot coloration

What is the target audience of the polychromatism and the often vivid, potentially aposematic, coloration in the non-operculate vermetids, *Cayo* and *Thylacodes*? Vermetids use gill filtering and mucous-net suspension feeding from the water current, so their body color is unlikely to be involved in targeting food items. As sessile organisms that broadcast spermatophores (*Hadfield & Hopper, 1980*), such coloration would have no role in their mating even if they were able to see colors (no published information on vermetid or other caenogastropod color vision was found). The most likely roles of this elaborate body coloration therefore are predator avoidance and/or deterrence, at least during daytime and at shallow depth. Fish are the main predators and are known to have color vision (*e.g.*, *Siebeck, Wallis & Litherland, 2008*). It appears that multiple mechanisms might be involved: (1) crypsis, with for instance the mottled patterning in the grey morph of *T. decussatus* (Fig. 6B) breaking up the head-foot outline and blending into the surroundings (*Stevens & Merilaita, 2009*); (2) predator confusion, as in the case of *T. bermudensis* n. sp. with multiple colors mixed across the population (Fig. 4A), and (3) aposematism, with vivid red, orange, and yellow colors coupled with "aggressive" movements, as in the case of the orange morph of *T. decussatus* (Figs. 6A, 8). It should be noted that the vivid coloration of vermetids color may, depending on surroundings, be part of camouflage (the bright orange body in some morphs of *T. vandyensis,* for instance provide background matching among the likewise vivid coloration of other fouling organisms such as red encrusting sponges (*Bieler et al., 2017a*; *Bieler et al., 2017b*), and its detectability might change with the blue-shifted light at greater water depth. In shallow-water habitats, against the pale base coloration of coral tissue and sand patches, however, such patterns stand out and appear to invite visual predators to notice. Species such as *Thylacodes decussatus*, with a vivid orange-red head-foot surrounded by black markings, clearly advertise their presence

and, when approached by fish, often do not withdraw into the shell like their operculate cousins but instead rapidly extend their head-foot in an aggressive manner to deter such fish (Florida Keys and Belize, Bieler, 1983–2023, pers. obs.). It appears that at least in these cases of extreme coloration that do not blend into the surroundings, the vermetid head-foot coloration is aposematic. The neon-like effect of brightly colored *Cayo margarita* and *C. galbinus* n. spp. might fall into the same category. The appearance of these vermetids in the ultraviolet light spectrum (which many reef fish can see; *Siebeck, Wallis & Litherland, 2008*) has yet to be explored. Overall, non-operculate vermetids provide a rich study system in which to explore the evolution of warning coloration, akin to the better-explored situation of nudibranch slugs for which more extensive observations and experimental data for defensive traits exist (*e.g.*, *Tullrot & Sundberg, 1991*; *Tullrot, 1994*; *Avila, Núñez Pons & Moles, 2018*).

## Sperm ultrastructure

Like the majority of other caenogastropods, the Vermetidae produce two morphologically distinct lines of spermatozoa—one for egg fertilization (euspermatozoa) and one of likely multiple, though non-fertilizing, functions (paraspermatozoa) (see reviews of *Melone, Lora Lamia Donin & Cotelli, 1980*; *Healy & Jamieson, 1981*; *Giusti & Selmi, 1982*; *Healy, 1988a*; *Hodgson, 1997*; *Buckland-Nicks, 1998*). Light microscopic accounts had demonstrated that paraspermatozoa of vermetids were worm-shaped and composed of a central bundle of axonemes surrounded by large vesicles (*Kuschakewitsch, 1913*; *Nishiwaki, 1964*; *Tochimoto, 1967*). At the ultrastructural level relatively few vermetid species and genera have previously been examined (*Hadfield, 1966*; *Melone, Lora Lamia Donin & Cotelli, 1980*; *Healy, 1988b*; *Buckland-Nicks & Hadfield, 2005*). However, those results, in conjunction with new data presented herein, have proven of systematic and phylogenetic importance (*Healy, 1988a*; *Healy, 1996*; *Buckland-Nicks & Hadfield, 2005*; *Ponder et al., 2008*). Based on details of eusperm and parasperm ultrastructure, *Healy (1988a)* and *Healy (1988b)* excluded the Vermetidae from the Cerithioidea and suggested that their closest affinities lay with that broad assemblage of Caenogastropoda outside of the Architaenioglossa+Cyclophoroidea+Cerithioidea, most likely with littorinimorph superfamilies such as the Cypraeoidea, Stromboidea (now including Xenophoridae) and Tonnoidea, a view further supported by more recent vermetid sperm work (*Buckland-Nicks & Hadfield, 2005*) and phylogenetic studies (*Ponder et al., 2008*; *Strong et al., 2011*).

## Comparison between *Thylacodes* and *Cayo* sperm ultrastructure

Available eusperm ultrastructural data for *Thylacodes* (*T. arenaria*–(*Melone, Lora Lamia Donin & Cotelli, 1980*); *T.* sp.–(*Healy, 1988b*); *T. squamigerus*–(*Hadfield, 1966*); Buckland-Nicks & Hadfield, 2005; *T. decussatus*–present study) and *Cayo* (*C. galbinus*, *C. refulgens*–present study) suggest that the two genera can be distinguished from each other on basis of the acrosomal vesicle profile (circular in *Thylacodes*, flattened in *Cayo*). In fact, the flattened acrosomal condition observed in *Cayo* spp. has not previously been observed in the Vermetidae. Unfortunately, available parasperm data do not appear to be as helpful in separating the two genera: *Thylacodes* showing much variation between species in the

maximum axonemal number (from 3 to 48) (*Hadfield, 1966*; *Melone, Lora Lamia Donin & Cotelli, 1980*; *Healy, 1988a*; present study and JM Healy & R Bieler, unpublished data, 2023) and both species of *Cayo* (present study) showing four axonemes. However, within *Thylacodes*, parasperm axonemal number may prove very useful in assessing species clusters (*e.g.*, *T. decussatus* (present study) and *T. squamigerus* (*Hadfield, 1966*) both showing large numbers of axonemes; most other examined species showing only four axonemes–JM Healy & R Bieler, unpublished data, 2023). In *Cayo*, the striking difference between *C. galbinus* and *C. refulgens* in eusperm nuclear morphology (short and helically-keeled in *C. galbinus*, thin, longer and rod-like in *C. refulgens*) (present study) not only help to define these two new species but also show that eusperm morphology in the Vermetidae is potentially much more diverse than the results of previous studies have suggested. It is also worth noting that the helically-keeled nuclear morphology of *C. galbinus* has not to date been observed in any other vermetid species. Elsewhere among the Caenogastropoda, helical or helically-keeled eusperm nuclei are routinely observed only in the Ampullarioidea (*e.g.*, *Yasuzumi & Tanaka, 1958*; *Kohnert & Storch, 1984*; *Koike, 1985*; *Catalán, Schlick de Santolaya & Winik, 1997*), and sporadically in the Truncatelloidea (*Giusti & Mazzini, 1973*; *Claveria & Etges, 1988*; JM Healy, unpublished, 2023), Littorinoidea and Velutinoidea (JM Healy, unpublished, 2023). In contrast, helical or helically-keeled sperm nuclei are widespread among heterobranch gastropods (for examples, figures, and full literature see reviews of *Thompson, 1973*; *Healy, 1988b*; *Healy, 1996*). *Thompson (1973)* suggested that helical keels in the sperm nucleus and midpiece of heterobranchs may perhaps be associated with a need for improved mobility through fluids of increased viscosity within the receiving 'female' reproductive tract. With this in mind, it would be interesting to compare in closer detail the female tracts of *C. galbinus* with *C. refulgens* (and other vermetids) to determine if there is a functional correlation with the observed sperm nuclear difference between the two species. Finally, in relation to parasperm morphology, the two examined species of *Cayo* show a marked difference in the contents of the dense vesicles (differentiated contents in *C. galbinus* versus homogeneous contents in *C. refulgens*)–the differentiated condition being seen in most other vermetids and therefore presumably the ancestral condition.

## Unrecognized vermetid diversity within the Caribbean.

Our ongoing study of vermetids in the western Atlantic continues to uncover new morphological forms which molecular, anatomical, behavioral, and sperm characters are helping to confirm as genetically discrete lineages (*Golding et al., 2014*; *Bieler et al., 2017a*; *Bieler et al., 2017b*; *Bieler et al., 2019*). The present study is particularly significant in revealing an entirely new clade of non-operculate vermetids which, upon first appearance, could easily be confused with members of the genus, *Thylacodes*. The application of these same tools is also helping to establish species-level status in some vermetid taxa, particularly those with marked plasticity in shell growth, or exhibiting distinct color variants, such as here for *Thylacodes bermudensis*. Distinguishing previously unrecognized native diversity from recent introductions of non-indigenous species, however, is a much less straightforward task in this historically understudied group. Vermetids are known to attach to ship hulls and other floating objects allowing long-distance dispersal and their

larvae might be dispersed by ballast waters. Several species of this family have been tagged as potentially invasive or with a high potential to be invasive (*e.g.*, Bieler in *Carlton, 1999*; *Coles & Eldredge, 2002*; *Strathmann & Strathmann, 2006*; *Bieler et al., 2017a*; *Bieler et al., 2017b*; *Patoka et al., 2020*). Recently, *Patoka et al. (2020)* identified a *Thylacodes* sp. along with 16 other non-target invertebrate species in the holding tanks of a marine pet distributor in the Czech Republic, illustrating the potential for biofouling species to hitchhike along with the transport of targeted marine species. The COI barcode sequence of this *Thylacodes* specimen was not a species-level match to any taxon in our database but appeared most closely related to *Thylacodes variabilis* (Hadfield & Kay, 1972 in *Hadfield et al., 1972*) from Hawaii, illustrating the potential for specimen transport across ocean basins. In the case of the newly described *Cayo* clade, without any other global records to date, we assume that these comparatively small-bodied forms represent previously unrecognized native populations, akin to the recently recognized radiation in another small-bodied vermetid group, *Cupolaconcha* (*Golding et al., 2014*). We also interpret *Thylacodes bermudensis* as a native species, with the earliest published reference of its presence in that archipelago stemming from 1864 (then misinterpreted as a tube-building bivalve) and its apparent restriction to Bermuda.

## CONCLUSIONS

There has been an independent loss of an operculum at least twice in the history of Vermetidae. The two clades, *Thylacodes* and *Cayo* n. gen, differ in many morphological and behavioral features (including larval shell and eusperm morphology, substrate-entrenching behavior, overall shell size, and expression of polychromatism) and belong to different branches of the vermetid tree. These results add to the accumulating evidence that convergent evolution is more common than has been widely appreciated (*Blount, Lenski & Losos, 2018*). In this case, the development of a robust phylogenetic hypothesis supported by morpho-anatomical, ultrastructural, behavioral, developmental, and molecular data enabled the recognition of convergence. The severe constraints of a sessile filter feeding lifestyle, with limited options for antipredatory responses, may also limit the range of adaptive solutions, leading to similar outcomes. Another example of adaptive constraints leading to similar outcomes in the mollusks is convergence in shell shape in swimming bivalves (*Serb et al., 2017*). Further comparative studies based on robust phylogenies will undoubtedly turn up additional examples, allowing us to arrive at an estimate of the frequency of, and optimal conditions for, convergent evolution.

## ACKNOWLEDGEMENTS

Over the many years that led to this study, numerous friends and colleagues helped us with obtaining vermetid specimens above and below the water line. For help with material for this particular portion of the study we especially thank Michael Hadfield (joint field work in the U.S. Virgin Islands), and Paula Mikkelsen (joint field work in Florida and Bahamas). Formal research station visits, field workshops, and research expeditions were supported by the Bermuda Biological Station and the Bermuda Aquarium, Museum & Zoo (Wolfgang Sterrer

and Struan R. Smith); the West Indies Laboratory of the Fairleigh Dickinson University, St. Croix (John and Nancy Ogden); locations in Abaco, Bahamas (Craig Layman and Friends of the Environment); and the Smithsonian Marine Station on Carrie Bow Cay, Belize (Klaus Ruetzler, Zachary Foltz, Scott Jones; Caribbean Coral Reef Ecosystems Program). Help with loans of type and other museum material and assistance during museum visits was provided by numerous individuals, including Ole Tendal and Antonia Vedelsby (ZMUC), and Ellen Strong (USNM). Stephen Cairns (USNM, SI) kindly advised on coral identifications. Lisa Kanellos (FMNH) assisted with figure arrangements, Janeen Jones and Kalina Griffin-Jakymec (FMNH) with specimen and data management, and Betty Strack and Stephanie Ware (FMNH) with SEM support. EM technician Erica Lovas (Queensland Museum) carried out tissue preparation, sectioning, and photography of sections at the Centre for Microscopy and Microanalysis (an Australian Microscopy and Microanalysis Research Facility) at the University of Queensland (the staff of which is here thanked for access to facilities and technical assistance). We thank two anonymous reviewers and Marta Calvo for their constructive input. This is Contribution 297, Bermuda Biodiversity Project (BBP), Bermuda Aquarium, Museum and Zoo, Department of Environment & Natural Resources.

### Funding

Vermetid research was funded under US National Science Foundation awards DBI-0841760/0841777. Coral reef-related studies in the Florida Keys were supported by the Paul M. Angell Family Foundation (to Rüdiger Bieler). Other fieldwork and specimen acquisition were partly supported by The Negaunee Foundation and The Grainger Foundation (to Rüdiger Bieler). The funders had no role in study design, data collection and analysis, decision to publish, or preparation of the manuscript.

### Grant Disclosures

The following grant information was disclosed by the authors:
US National Science Foundation: DBI-0841760/0841777.
Coral reef-related studies in the Florida Keys were supported by the Paul M. Angell Family Foundation.
Negaunee Foundation and The Grainger Foundation.

### Competing Interests

Rüdiger Bieler, Petra Sierwald and Timothy Collins are Academic Editors for PeerJ.

### Author Contributions

- Rüdiger Bieler conceived and designed the experiments, performed the experiments, analyzed the data, prepared figures and/or tables, authored or reviewed drafts of the article, and approved the final draft.

- Timothy M. Collins conceived and designed the experiments, performed the experiments, analyzed the data, authored or reviewed drafts of the article, and approved the final draft.
- Rosemary Golding conceived and designed the experiments, performed the experiments, analyzed the data, prepared figures and/or tables, authored or reviewed drafts of the article, and approved the final draft.
- Camila Granados-Cifuentes performed the experiments, authored or reviewed drafts of the article, and approved the final draft.
- John M. Healy conceived and designed the experiments, performed the experiments, analyzed the data, prepared figures and/or tables, authored or reviewed drafts of the article, and approved the final draft.
- Timothy A. Rawlings conceived and designed the experiments, performed the experiments, analyzed the data, prepared figures and/or tables, authored or reviewed drafts of the article, and approved the final draft.
- Petra Sierwald performed the experiments, authored or reviewed drafts of the article, and approved the final draft.

## Field Study Permissions

The following information was supplied relating to field study approvals (i.e., approving body and any reference numbers):

Specimen collecting in the protected waters of the Florida Keys was conducted under Florida Keys National Marine Sanctuary Research Permit FKNMS-2017-069 and U.S. Fish and Wildlife Service Special Use Permit 41580-FWS-17-1 (and earlier issues of these permits) for work in the National Wildlife Refuges. Collecting in Bermuda in 1983 was conducted under the auspices of the Bermuda Biological Station, in 2013 under a Department of Environmental Protection Special Permit from the Bermuda Government. Collecting in the Bahamas was conducted a Marine Scientific Research Permit from the Department of Fisheries/Marine Resources of the Bahamian Government. Field work in Belize was conducted under the auspices of Smithsonian's Caribbean Coral Reef Ecosystems Program. Vermetid surveys on the Caribbean island of Saba (now part of The Netherlands) were supported by the Saba Conservation Foundation and permitted by the Executive Council of the Public Entity Saba.

## DNA Deposition

The following information was supplied regarding the deposition of DNA sequences:

The mtDNA sequences are available at GenBank: OQ728805–OQ728816, OQ732731, OQ720922–OQ720930.

Nuclear sequences are also available at GenBank: OQ725639–OQ725650. For details see Table 1.

## Data Deposition

The studied vermetid specimens are stored in the permanent collections of the Field Museum of Natural History in Chicago (FMNH), with certain type specimens having

been transferred to the Bermuda Aquarium, Museum and Zoo (BAMZ), and the National Museum of Natural History in Washington, D.C. (USNM). The individual registration numbers are available in the Supplementary File.

Access to taxonomic and collection-event data for the mollusks mentioned in this study are available through FMNH's institutional Invertebrate Zoology collections database at http://collections-zoology.fieldmuseum.org/. The collection links to the 141 cited specimen lots are available in the Supplementary File.

### New Species Registration

The following information was supplied regarding the registration of a newly described species:

Publication LSID:

urn:lsid:zoobank.org:pub:815F8878-6EC4-4EA0-BFB7-3AFEA171E563

Genus LSID:

urn:lsid:zoobank.org:act:105AE8A1-2C20-4B5C-BF56-01E21E77D25B

Species LSIDs:

- Cayo margarita: urn:lsid:zoobank.org:act:9D7D1DEB-6FA9-42FF-8FF6-89354D701D50

- Cayo brunneimaculatus LSID: urn:lsid:zoobank.org:act:E27FF0F3-E122-481E-9F92-CB0ACA31BFC5

- Cayo galbinus: urn:lsid:zoobank.org:act:55427624-5822-4FDA-8798-ABE1B8DCFB45

- Cayo refulgens LSID: urn:lsid:zoobank.org:act:803D13C6-B9FA-41BB-AEBF-A9842D05CCE3

- Thylacodes bermudensis LSID: urn:lsid:zoobank.org:act:B8584F9A-72DA-4DAC-B951-1137DB991EA5.

### Supplemental Information

Supplemental information for this article can be found online at http://dx.doi.org/10.7717/peerj.15854#supplemental-information.

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
