# Peer review of "Replacing mechanical protection with colorful faces–twice: parallel evolution of the non-operculate marine worm-snail genera Thylacodes (Guettard, 1770) and Cayo n. gen. (Gastropoda: Vermetidae)"

_PeerJ, doi:10.7717/peerj.15854_

## Round 0.1 · original submission · Minor Revisions

I agree with the reviewers that this manuscript is very well written, and the framework and scope are valid and extensive. Pending minor revisions, I think this manuscript will be a great addition to the understanding of the taxonomy and systematics of the non-operculate marine worm-snails.

Reviewer 1 ·

Basic reporting

The ms by Bieler et al. deals with antipredatory protection by various species of Caribbean / West Atlantic vermetid worm snails consisting of aposematic coloration hypothesized to warn predators of toxins that the animals secrete. This bright coloration is not seen in all species. The species are cryptic because of their small maximum size, which is why four of them could be described as new to science, along with a new genus. The new taxa are not only distinguished because of their distinctive morphological characters but also because of their separate positions in a phylogeny reconstruction. Judging by the photographs, some of them are found embedded in coral and appear to be harmful to their living hosts or competitors for space (scleractinian corals and zoantharians) owing to their toxicity, which is of ecological relevance.

Experimental design

Experimental design is clear and extensive, following standard protocols, rigorously executed

Validity of the findings

The paper is original and a valid contribution to its field. The text is clear and extensively written with many relevant details. All criteria have been met. The authors are clearly experts in their field. The ms reads very well. I found it very informative because it deals with species that are commonly overlooked. It is also well-illustrated. The in-situ illustrations have ecological information on the habitat (substrate choice and aggressive behavior) that could perhaps get more emphasis in the discussion

Additional comments

Minor comments.

Abstract, first line. There is no need for a hyphen in “worm-snails”. The hyphen in the title (worm-snail genera) is correct.
Line 5. When the authors say “not unlike that of tube-building serpulid worms”, could they perhaps be more specific by giving examples of such worm behavior (recently published examples of Caribbean Spirobranchus come to mind) and explain if the tubes of the vermetids are found likewise on dead substrate or on living hosts, such as corals? In case they are living on corals, are the tubes (partly) embedded inside the host or do they only attach to the host’s surface. A recent study of Caribbean Petaloconchus shows that the shells can be embedded, which cause them to be partly concealed. This raises the question whether such endolithic animals are protected by their substrate.
Line 17. FYI. A recent example of (nocturnal) predation of Red Sea vermetids by crabs was published in 2021: https://doi.org/10.1002/ecy.3420
Line 29. Red is not distinguishable at depths over 10 m. Does this have implications for the depth distribution of the snails? We can only assume that this coloration has a function against predators that are active in day-time and in shallow water.
Line 37. Orange does not appear so at depths over 10 m. See remarks above.
Lines 67-68. Netherlands Antilles have been renamed Dutch Caribbean on 10-10-2010. https://en.wikipedia.org/wiki/Caribbean_Netherlands
Lines 324. The paper follows the rules of the ICZN by the registration of the article in Zoobank. As an addition, I suggest that you add a Zoobank registration number for the new taxon. For examples, see species descriptions in ZooKeys.
Line 332. 1–2.5 m -> 1–2.5 m depth
Line 407. The walls of rock pools consist of dead coral rock, limestone, or anything else? What is the substrate at greater depths? Dead coral?
Line 423. large shelled -> large-shelled
Line 441. pedal disk coloration -> pedal-disk coloration
Lines 467-502. The authors could add the word “depth” where depth ranges are mentioned. This also concerns the “Material examined” sections for the other species.
Line. 593. Zoobank registration number for the new genus name?
Line. 640. Zoobank registration number for the new species name?
Line 642. Here you say “5.9 m depth”. I suggest that you do this elsewhere when relevant. Just for the record.
Line 689. Hyphen should be replaced by n-dash in “5-6 times”
Line 697. Do you mean “live coral boulders”?
Line 697. Spelling. Orbicella foveolata -> Orbicella faveolata
Line. 725. Zoobank registration number for the new species name?
Line 812. C. refulgens and C. brunneimaculatus n. spp. -> C. refulgens sp. and C. brunneimaculatus n. sp.
Line. 841. Zoobank registration number for the new species name?
Line. 950. Zoobank registration number for the new species name?
Lines 1064 and onward. The effect of retraction also depends on how deep the shell is embedded in the coral matrix. Some crabs have no problem with breaking vermetid shells (see reference above).
Line 1090. A recent example has been published from the Caribbean in which coral growth was affected by Pateloconchus. Here we do not only see effect on scleractinian corals, but also in the zoantharian Palythoa (Fig. 4C)
Lin 1129. Only effective in day-time and shallow depths (l. 1138)
Line 1190. Thompson, 1973; Healy, 1988b, 1996). -> Thompson (1973) and Healy (1988b, 1996).
Line 1197. Delete “versus”
Line 1203. Bieler et al., 2017a; 2017b; Bieler et al., 2019 -> Bieler et al., 2017a; 2017b; 2019
Figure 14A. This is not Favia fragum because the polyps are not separated from each other and too small in comparison to the snail tube. More likely Porites sp. because of the small polyp size and polyp morphology.

·

Basic reporting

The article uses clear, technically correct text and conforms to an acceptable format of ‘standard sections’.

The introduction and background properly fit the subject into a broader field of knowledge.

All the sections of the manuscript (introduction., material and methods, results with taxonomic descriptions, and discussion) are correctly and rigorously treated, supported by a nice set of figures and an exhaustive bibliographical review. Therefore, I only provide bellow few comments on small details.

Experimental design

It is mainly a descriptive or observational research. Further, the methodology for the molecular and phylogenetic analyses is correct.

The research has been conducted in conformity with the prevailing ethical standards in the field.

Methods are described in great detail with sufficient information to be reproducible by other researchers.

DNA data is correctly provided.

Validity of the findings

The findings provided are valid, very interesting, and constitute a relevant advance in the topic of study.

Additional comments

General comment

Historically Vermetidae is one of the most understudied group of gastropods. Its taxonomy and systematics is currently in a state of disarray and many species remain undescribed because of the difficulty in finding diagnostic phenotypic characters. The first author (RB) is a maximum authority in the study of these peculiar gastropods and he is putting order within this taxonomic group in successive research pieces. Thus, the present study deals with two clades characterized by the loss of the operculum and inquires about the evolution of these gastropods. As a result, a new genus and several new species are described through integrative taxonomy (using characters of the shell, protoconch, soft-body, sperm ultratrastructure, molecular data, and data on development and habitat and ecology). All of this represents a considerable advance in the knowledge of the diverse vermetid fauna in the Caribbean area. At the same time, broadly, new perspectives are provided on the evolution of these gastropods regarding the loss of the operculum and the acquisition of alternative defensive strategies.

This research piece constitutes a new and relevant advance in the study of the fauna of vermetid gastropods of the Caribbean area, which follows two papers previously published in this same journal:

Bieler R, Collins TM, Golding RE, Rawlings TA. 2019. A novel and enigmatic two-holed shell aperture in a new species of suspension-feeding worm-snail (Vermetidae). PeerJ 7:e6569. doi:10.7717/peerj.6569

Bieler R, Granados-Cifuentes C, Rawlings TR, Sierwald P, Collins TM. 2017a. Non-native molluscan colonizers on deliberately placed shipwrecks in the Florida Keys, with description of a new species of potentially invasive worm-snail (Gastropoda: Vermetidae). PeerJ 5:e3158. doi:10.7717/peerj.3158

Minor comments

- Are there other known potential predators on these species apart from fish?
In the Mediterranean, the muricid Stramonita haemastoma and the buccinid Euthria cornea have been observed preying on Thylacodes arenarius by drilling the shell. In this species the epibiotic overgrowth contributes to its defense against these predators.

Abstract
- Instead of “… allied with operculate species Vermetus triquetrus and V. bieleri” I suggest: “ … allied with the species with reduced operculum Vermetus triquetrus and V. bieleri”.
- Instead of “… show characteristics similar to nudibranch slugs …” I suggest: “… show characteristics approaching to nudibranch slugs …”

Introduction
- Page 38: I suggest “… there is a some similarity to …” instead of “… there is a strong similarity to …”.

Material and methods
- The material and methods is very detailed. Perhaps some routine procedures could be summarized.

Results

- Page 270. Specify “Vermetus bieleri from the NE Atlantic archipelagos and V. triquetrus from the Mediterranean”.

- Page 338: I suggest “…Chama sp.” instead of “…Chama.”

To which morph of those described in the lines 384-391 does the holotype correspond?

- Page 400: Instead of “several” specify how many (range or “up to …”). Up to 23 per female is specify in S. decussatus. Unify the descriptions of the two species.

- Page 419: Delete “island”. I suggest “Currently only known from the archipelago of Bermuda”.

- Page 447: “… from other W Atlantic Thylacodes” instead of “… from other Atlantic Thylacodes”.

- Pages 504- : Provide data on size of adult shell mass and tube diameter, as in T. bermudensis. Unify the descriptions of the two species. Diameter of the tube opening of T. decussatus is provided in figures 6 and 8.

- Page 551: Move to the external morphology “Foot held in a folded triangular shape, with the narrowest part below the mouth” (does not refer to habitat).

- Pages 615- “Comparative and taxonomic remarks”: Include among the differentiating characters of Thylacodes and Cayo, the smaller size of the species of the new genus.

- Page 623: “Most Thylacodes species …” instead of “Thylacodes species”, since the Mediterranean T. arenarius always shows the same color pattern.

- Page 770: Add “mm” after “1.9-2.5”.

- Page 866: “26°38.457’N, 077°02.245’W” instead of “N 26°38.457’, W 077°02.245’”.

Discussion

- Page 1070: delete comma before Thylacodes arenarius.

- Page: 1213: “sp” no italics.

- It should be commented that the genus Vermetus may be considered an intermediate step in the evolution towards the loss of the operculum.

Conclusion

- Page 1228: add “size” among the differentiating characters.

- Page 1232: specify what type of data instead of “several type of data”.

- Something should be commented within discussion on inter- and intraspecific sequence difference proposed or considered.

Reviewer 3 ·

Basic reporting

The manuscript is clearly written and easy to follow with very informative and relevant tables and figures. The provided background information is logically structured showing how this specific study fits in a broader context. Raw data are provided.

Experimental design

The research goal is well defined and relevant. The research to address the question is original, extensive, of high quality and described in detail.

Validity of the findings

All results are provided and sound. Conclusions are clear, supported by extensive data discussed and connect to the original research goal discussed in the introduction.

Additional comments

This manuscript is a very comprehensive study of a very interesting but often overlooked group of molluscs common on any hard surfaces in temperate and tropical shallow marine habitats. Parallels to nudibranchs are exciting and though the study is not necessarily conclusive to the benefits of coloration, it is a start. The presented data is discussed in detail while also giving some historic context of the groupʻs taxonomy while adding new taxa. It is an important contribution to the still limited knowledge of diversification in marine invertebrates.

---

## Round 0.2 · accepted · Accept

Thank you for following through on the suggestions and comments suggested by the reviewers. The current version of the manuscript is suitable for publication.